# LANGUAGE MODELS ARE OPEN KNOWLEDGE GRAPHS

## ABSTRACT

This paper shows how to construct knowledge graphs (KGs) from pre-trained language models (e.g., BERT, GPT-2/3), without human supervision. Popular KGs (e.g, Wikidata, NELL) are built in either a supervised or semi-supervised manner, requiring humans to create knowledge. Recent deep language models automatically acquire knowledge from large-scale corpora via pre-training. The stored knowledge has enabled the language models to improve downstream NLP tasks, e.g., answering questions, and writing code and articles. In this paper, we propose an unsupervised method to cast the knowledge contained within language models into KGs. We show that KGs are constructed with a single forward pass of the pre-trained language models (without fine-tuning) over the corpora. We demonstrate the quality of the constructed KGs by comparing to two KGs (Wikidata, TAC KBP) created by humans. Our KGs also provide open factual knowledge that is new in the existing KGs. Our code and KGs will be made publicly available.

## 1 INTRODUCTION

Knowledge graphs (KGs) are an important resource for both humans and machines. Factual knowledge in KGs is injected into AI applications to imitate important skills possessed by humans, e.g., reasoning and understanding. KG construction is mainly supervised, requiring humans to handwrite every fact, such as Freebase (Bollacker et al., 2008) and Wikidata. KGs can also be constructed in a semi-supervised way, in which a semi-automatic extractor is used to obtain the facts from web corpora (e.g., NELL (Carlson et al., 2010) and Knowledge Vault (Dong et al., 2014)). Humans however still need to interact with the extractor to improve the quality of the discovered facts. Therefore, human supervision, which is often expensive, is required in constructing KGs.

Recent progress in language models (LMs), such as BERT (Devlin et al., 2018) and GPT-2/3 (Radford et al., 2019; Brown et al., 2020), has led to superior results even outperforming humans in a wide range of tasks, e.g., sentence classification (Wang et al., 2018), question answering (Brown et al., 2020). Pre-trained LMs are also capable to write poetry, music, and code, while such tasks often require we human to spend a significant amount of time in learning the relevant knowledge to work well. In fact, these pre-trained LMs automatically acquire factual knowledge from large-scale corpora (e.g., BookCorpus (Zhu et al., 2015), Common Crawl (Brown et al., 2020)) via pre-training. The learned knowledge in pre-trained LMs is the key to the current success. We therefore consider the following question: instead of using the manually created knowledge, *can we use the knowledge stored in pre-trained LMs to construct KGs?*

In this paper, we design an unsupervised approach called MAMA that successfully recovers the factual knowledge stored in LMs to build KGs from scratch. MAMA constructs a KG with a single forward pass of a pre-trained LM (without fine-tuning) over a textual corpus. As illustrated in Figure 1, MAMA has two stages: Match and Map. **Match** stage generates a set of candidate facts by matching the facts in the textual corpus with the knowledge in the pre-trained LM. General or world knowledge from large-scale corpora is embedded in the LM, thus candidate facts in the target corpus are often covered by the knowledge in the LM. The candidate facts are matched through an efficient beam search in the attention weight matrices of the pre-trained LM without fine-tuning. **Map** stage produces an open KG via mapping the matched candidate facts from Match stage to both fixed KG schema and open schema. If the schema of candidate facts exists in the KG schema, we map the candidate facts directly to the fixed KG schema. Otherwise, we reserve the unmapped candidate

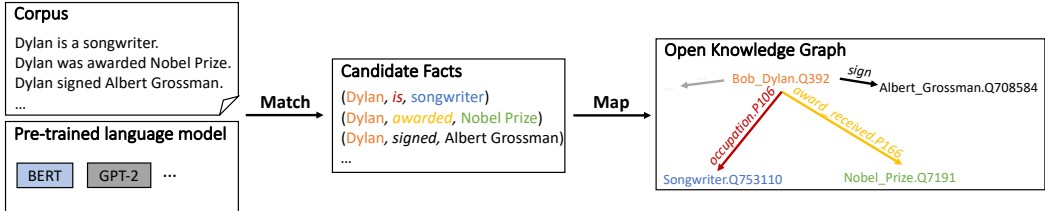

Figure 1: Overview of the proposed approach MAMA. MAMA constructs an open knowledge graph (KG) with a single forward pass of the pre-trained language model (LM) (without fine-tuning) over the corpus. Given the input: a textual corpus containing passages and sentences, e.g., English Wikipedia, and a pre-trained LM, e.g., BERT, GPT-2/3, MAMA (1) generates a set of candidate facts via *matching* the knowledge in the pre-trained LM with facts in the textual corpus, e.g., a candidate fact *(Dylan, is, songwriter)* from the sentence "Dylan is a songwriter.", and (2) produces an open KG by *mapping* the matched candidate facts to both an existing KG schema, e.g., *(Bob_Dylan.Q392, occupation.P106, Songwriter.Q753110)* in Wikidata schema, and an open schema, e.g., *(Bob_Dylan.Q392, sign, Albert_Grossman.Q708584)*.

facts in the open schema. This results in a new type of KG, *open KG*, with a mixture of mapped facts in fixed KG schema and unmapped facts in the open schema.

Our contributions are as follows:

1. We show *how to construct KGs from pre-trained LMs*. The KGs are constructed with a single forward pass of the pre-trained LMs without fine-tuning over the textual corpora. This helps researchers explicitly understand what the language models learn, bridging the deep LM and KG communities through enhanced model transparency.
2. We propose an unsupervised two-stage approach, MAMA, to first *match* the candidate facts in the corpora with the knowledge stored in LMs, then *map* the matched candidate facts to both fixed and open schema to produce a KG.
3. We generate a new type of KG, namely open KG, consists of *mapped facts* in the fixed KG schema of existing KGs (Wikidata and TAC KBP) annotated by humans; and *unmapped facts* in the open schema that are new in the reference KG schema. The reach of this result is broad and has downstream utility for knowledge graph construction, deep neural network interpretation, and information extraction.

## 2 MAMA

We introduce an unsupervised end-to-end approach Match and Map (MAMA) as illustrated in Figure 1 to construct open knowledge graphs (KGs) from language models (LMs). MAMA constructs the KGs with a single forward pass of the pre-trained LMs (without fine-tuning) over the corpora. The two stages of MAMA are:

**Match** generates a set of candidate facts from a textual corpus. LMs contain global or world knowledge learned from large-scale corpora, which often does not perfectly match the knowledge in the target corpus. The goal of this stage is to match the knowledge stored in pre-trained LMs with facts in the corpus. Each fact is represented as a triplet *(head, relation, tail)*[1], in short, $(h, r, t)$, and passed to Map stage. Match procedure is detailed in Sec. 2.1.

**Map** produces an open KG using the matched candidate facts from Match stage. The constructed open KG has two portions: (a) mapped candidate facts that are in a fixed KG schema, e.g., *(Dylan, is, songwriter)* is mapped to *(Bob_Dylan.Q392, occupation.P106, Songwriter.Q753110)* according to Wikidata schema; and (b) unmapped candidate facts that are in an open schema, e.g., a candidate fact *(Dylan, signed, Albert Grossman)* is partially mapped to *(Bob_Dylan.Q392, sign, Albert_Grossman.Q708584)* in the open schema. This stage is described in Sec. 2.2.

(a) Matching example.

(b) Attention matrix for matching degree calculation.

Figure 2: Illustration of Match stage. The upper part of (a) represents the general matching steps of generating the best matched candidate fact *(Dylan, is, songwriter)* from the sentence "Dylan is a songwriter." The lower portion shows the corresponding step-by-step process. Given a head-tail pair *(Dylan, songwriter)*, at each step, the search chooses one of the actions, i.e., START, YIELD, STOP to produce an intermediate candidate fact. The search *starts* by adding the head "Dylan" as an initial candidate (step 0). The matching degree of the candidate is initialized as 0. Next, a new candidate is *yielded* if the candidate has not reached the tail "songwriter" (step 1 and step 2), by appending the next largest attended token (with the largest score from the attention matrix (b) of the sentence) to the end of the current candidate, and the corresponding matching degrees are increased by the associated attention scores (0.3 and 0.4) to 0.3 (0+0.3) and 0.7 (0.3+0.4) respectively. Otherwise, the search *stops*, and the candidate fact with the best matching degree is returned for the head-tail pair (step 3). The attention matrix (b) is from the forward pass of the LM without fine-tuning over the sentence. "x" marks the tokens to prevent searching backward.

## 2.1 MATCH

We frame the matching procedure as a search problem. To obtain the best matched candidate facts of an input sentence, the candidates with the top matching degrees are returned from a search process. The matching degree is derived from the search in the attention weight matrices of the pre-trained LM, since the attention weight matrices are one of the main containers of the knowledge in the pre-trained LM. The attention weight matrices are simply from the forward pass of the LM without fine-tuning over the sentence.

### 2.1.1 BEAM SEARCH

We design a simple-yet-effective beam search to find the best matched candidate facts. For every head-tail pair $(h, t)$ in a sentence, the search maintains the $k$-best matched candidate facts of the pair. Let's first consider the search from left to right with beam size equals to 1. An example search process is shown in Figure 2. Given a head-tail pair *(Dylan, songwriter)*, at each step, the search performs one of the following actions:

**START** the search from the head. The head $h$ is added as an initial candidate into the beam. For simplicity, we use START($h$) to denote the action, which returns a candidate $(h,$. In Figure 2(a), at step 0, the head "Dylan" is added as *(Dylan,* into the beam. The matching degree is initialized to 0.

**YIELD** a new intermediate candidate in the beam if the current candidate has not reached the tail. The next largest attended token (with the largest score from the attention matrix) is appended to the end of the current candidate to yield the new candidate. The corresponding matching degrees are increased by the associated attention scores. At step 1 (orange arrow in Figure 2(a)), "is" is appended to the current candidate to yield *(Dylan, is,* , since "is" has the largest attention score with "Dylan" in the attention matrix. The attention score is 0.3 as highlighted in orange in Figure 2(b). The matching degree becomes 0.3 (i.e. 0+0.3). The multi-head attention is reduced to a single head so that every two tokens of the sentence are associated with one attention weight. We experiment with different reduction

---

[1] We use the term "head" and "tail" to denote head and tail's "entities" or "entity mentions" for simplicity.

---

**Algorithm 1** Beam search for matching candidate facts.

---

**Input:** Head-tail pair $(h, t)$, sentence $s$, attention matrix $\mathbf{A_s}$, action manager $\mathcal{O} = \{\text{START}, \text{YIELD}, \text{STOP}\}$, beam size $k$

**Output:** Candidate facts $\mathbb{T}_{(h,t)}$

1: $\mathbb{T}_{(h,t)} \leftarrow \{\text{START}(h)\}$                ▷ *Start* by adding the head as a candidate in the beam
2: **while** $\exists c \in \mathbb{T}_{(h,t)}[\mathcal{O}(c) = \text{YIELD}]$ **do**
3:      $\widetilde{\mathbb{T}}_{(h,t)} \leftarrow \emptyset$                  ▷ Initialize a new beam
4:      **for each** $c \in \mathbb{T}_{(h,t)}$ **do**
5:          **if** $\mathcal{O}(c) = \text{YIELD}$ **then**
6:             $\widetilde{\mathbb{T}}_{(h,t)} \leftarrow \widetilde{\mathbb{T}}_{(h,t)} \cup \{\text{YIELD}(c, s, \mathbf{A}_s)\}$      ▷ *Yield* a new candidate if not reached the tail
7:          **else**
8:             $\widetilde{\mathbb{T}}_{(h,t)} \leftarrow \widetilde{\mathbb{T}}_{(h,t)} \cup \{\text{STOP}(c, t)\}$      ▷ *Stop* then produce a valid fact if reached the tail
9:          **end if**
10:     **end for**
11:     $\mathbb{T}_{(h,t)} \leftarrow \text{TOP}(k, \widetilde{\mathbb{T}}_{(h,t)})$           ▷ Maintain $k$-best candidates in the beam
12: **end while**
13: **return** $\mathbb{T}_{(h,t)}$

---

setups in Sec. A.3. "x" marks the tokens (prior to the current token) that are not considered in the search to prevent searching backward. Step 2 similarly takes YIELD action to produce *(Dylan, is songwriter, .* The matching degree is now 0.7 (i.e. 0.3+0.4). We use YIELD$(c, s, \mathbf{A}_s)$ to denote the action, where $c$ is a current candidate, $s$ represents the sentence, and $\mathbf{A}_s$ is the attention matrix from the forward pass of the pre-trained LM over $s$, which yields a new candidate.

**STOP** the search step if the candidate has reached the tail, then add the candidate as a valid candidate fact into the beam. As beam size equals to 1, *(Dylan, is, songwriter)* is the only returned candidate fact for the given pair. The final matching degree of the candidate is 0.7. We denote this step using STOP$(c, t)$, which returns a valid fact.

The details of the proposed beam search are in Algorithm 1. The inputs of the search algorithm are a head-tail pair $(h, t)$, a sentence $s$, an attention matrix $\mathbf{A}_s$ of $s$. Both $h$ and $t$ are identified by the noun chunk in $s$. $\mathbf{A}_s$ is the attention matrix associated with $s$ from the forward pass of LM without fine-tuning. The search gets started by adding the head $h$ as the initial candidate in the beam (line 1). While there are still new candidates waiting to be yielded (line 2), the search continues, and the top $k$ candidates sorted by the matching degrees are maintained (line 3-11) in the beam. In practice, we implement an action manager $\mathcal{O}$ to decide which action to take at each step. Given a candidate $c$ in the beam, $\mathcal{O}(c) = \text{START}$ always happens at the beginning of the search. If $c$ has not reached the tail $t$ yet, $\mathcal{O}(c) = \text{YIELD}$. Otherwise, $\mathcal{O}(c) = \text{STOP}$. We convert the subwords to the corresponding full words. We also notice some facts are in reverse order in the sentence, e.g., "$\cdots$ said Jason Forcier , a vice president at battery maker A123 Systems Inc." for facts of relation "org:top_members_employees", thus enable bidirectionality by running the algorithm in both directions (left to right and right to left). The beam search is implemented by the breadth-first search, which is efficient as the time complexity is $O(k \cdot d)$, where $d$ is the maximum depth of the search tree.

### 2.1.2 FILTER

Although the basic functionality provided by beam search is sufficient for finding useful candidate facts, we have found a few constraints useful. Given a candidate fact $(h, r, t)$ from beam search result $\mathbb{T}_{(h,t)}$, it remains as a fact if satisfying all the following constraints.

**Constraint #1** The matching degree of $(h, r, t)$ is above a threshold. We compare the matching degrees corpus-wide to only reserve the facts that are matched better with the knowledge in LMs. For example, MAMA extracts a fact *(Rolling Stone, wrote, pop song)* from "Rolling Stone wrote: "No other pop song has so thoroughly challenged artistic conventions"", which is not an accurate fact based on the sentence. We observe the associated matching degree is below a proper threshold, while the matching degrees of high-quality facts from the same documents, e.g., *(Dylan, is, songwriter)*, or confident facts from the other documents are beyond the threshold.

**Constraint #2** The distinct frequency of $r$ is above a threshold. To avoid facts to be over-specified, e.g., *(Dylan, signed to Sam Peckinpah's film, Pat Garrett and Billy the Kid)*, we require $r$ should take many distinct head-tail pairs in the corpus.

**Constraint #3** Relation $r$ is a contiguous sequence in the sentence. We can avoid $r$ that has no meaningful interpretation (Fader et al., 2011), e.g., *(Rolling Stone, wrote challenged, conventions)* from the above sentence.

## 2.2 MAP

The objective of Map stage is to generate an open KG. The open KG contains (a) *mapped facts* in a KG schema (Sec. 2.2.1), e.g., Wikidata schema, if the schema of the candidate facts is within the existing KG schema; and (b) *unmapped facts* from (a) in an open schema (Sec. 2.2.2).

### 2.2.1 MAPPED FACTS IN KG SCHEMA

The goal is to map a candidate fact $(h, r, t)$ to a fact $(h_k, r_k, t_k)$ in the KG schema. The reason for mapping to an existing KG schema is to make use of the high-quality schema designed by experts (to avoid duplicated efforts of building from scratch) and enable evaluating the candidate facts with oracle KG facts contributed by human volunteers. We first map both entities $h, t$ to $h_k, t_k$, then map the relation $r$ to $r_k$ in the reference KG schema. Additional details are presented in Sec. A.1.

**Entity linking to KG schema** We adapt an unsupervised entity linker based on a mention-to-entity dictionary (Spitkovsky & Chang, 2012) to link the entities for scalability consideration. Besides, contextual information is crucial to link the entities correctly, we use the word embedding of the context to disambiguate the entities, which means we only link the entities with high contextual similarities based on the word embedding. We adopt the entity linker to map $h, t$ to $h_k, t_k$.

**Relation mapping with KG schema** We largely follow the relation mapping method proposed by Angeli et al. (2015) to construct an offline relation map between KG relation and relation phrases of the candidate facts. The basic idea is that the more often linked head-tail pairs (i.e., entities are with type information from the entity linking step) co-occur between the candidate facts and KG facts, the more likely the corresponding relations are mapped to each other. In addition, we normalize each relation phrase of the candidate facts by lemmatization, and removing inflection, auxiliary verbs, adjectives, adverbs. After the relation mapping is constructed, one author manually checks whether the top 15 relation phrases are true mappings for each KG relation. We only reserve the true ones in the final relation mapping. This process takes approximately one day. Later, $r$ is mapped to $r_k$ with an efficient look-up operation in the relation mapping.

### 2.2.2 UNMAPPED FACTS IN OPEN SCHEMA

An unmapped candidate fact $(h, r, t)$ means at least one of $h$, $r$, and $t$ is not mapped to the KG schema based on the method in Sec. 2.2.1. There are two types of unmapped candidate facts:

**Partially unmapped facts** represent at least one of $h$, $r$, and $t$ are mapped to the KG schema. It can be $h$ or $t$ mapped to $h_k$ or $t_k$ based on the entity linker in Sec. 2.2.1. It can also be $r$ that mapped to $r_k$ using the relation mapping in Sec. 2.2.1. This actually results in unmapped facts that are in a mixture of the KG schema and the open schema. As the overall schema of the unmapped facts is not in the KG schema, we use open schema to denote such unmapped facts in the rest of the paper for simplicity. An example is *(Dylan, signed, Albert Grossman)* in Figure 1, where both head and tail are linked to Wikidata schema based on the entity linker in Sec. 2.2.1, but the relation cannot be mapped since there is no relation mapping from "signed" to a KG relation in Wikidata schema.

**Completely unmapped facts** indicate all $h$, $r$, and $t$ are not mapped to the KG schema. This means neither the entity linker nor the relation mapping is able to map $h$, $r$, and $t$ to $h_k$, $r_k$, $t_k$ respectively. The resulting unmapped candidate facts stay in the open schema, e.g., a candidate fact *(Jacob, was, A Registered Mennonite)* stays the same in the open schema from a sentence "Jacob was a registered Mennonite in Amsterdam.".

The resulting open KG is a new type of KG that mixes the fixed KG schema with the flexible open schema, suggesting new directions for the next generation of KGs. The open KG not only contains existing knowledge in the reference KGs, but also extends the fixed KG schema with an additional open schema to improve the coverage of the KGs, that benefits the downstream KG based applications, e.g., QA and commonsense reasoning (Wang et al., 2019; Brown et al., 2020).

| KG | # of oracle facts | # of documents |
|---|---|---|
| TAC KBP | 27,655[3] | 3,877,207 |
| Wikidata | 27,368,562 | 6,047,494 |

Table 1: Dataset statistics of two knowledge graphs: TAC KBP and Wikidata. TAC KBP refers to TAC KBP Slot Filling 2013 challenge. # of oracle facts for TAC KBP is the number of oracle facts in the 2013 task. # of documents for TAC KBP is the number of the documents in the 2013 task. # of oracle facts for Wikidata is the total number of oracle facts in Wikidata. # of documents for Wikidata is the size of English Wikipedia.

## 3 EXPERIMENTS

*How well can language models generate knowledge graphs?* We experimentally explore how well can MAMA answer the question in the section. To measure the ability of LMs in generating KGs, we directly measure the quality of resulting open KGs. The open KG contains two types of facts: mapped facts in the fixed KG schema; and unmapped facts in the open schema. We first quantitatively evaluate MAMA by comparing the mapped facts to oracle KGs annotated by humans in Sec. 3.1, then conduct an in-depth analysis of the unmapped facts in Sec. 3.2.

### 3.1 RESULTS ON MAPPED FACTS

We first study the quality of the mapped facts. As the candidate facts have been mapped to the schema of oracle KGs, we are able to quantitively compare the candidate facts with the oracle facts in the reference KGs.

#### 3.1.1 DATASETS

We compare the mapped facts from MAMA with the facts in two KGs:

**TAC KBP** TAC Knowledge Base Population (KBP) Slot Filling is a task to search a document collection to fill in the tail/object entity for predefined relations (slots) for a given head/subject entity in a reference KG. We experiment with the reference KG in the 2013 challenge. We use the document collection and oracle facts of the 2013 task. The statistic of the dataset is shown in Table 1.

**Wikidata** We use popular Wikidata as another KG. We use all the oracle facts in Wikidata. We use the English Wikipedia as the text corpus, since a large amount of facts in Wikidata is from English Wikipedia. The statistic is in Table 1.

To evaluate the mapped facts, we first use Match stage of MAMA to run over the corresponding documents to generate the candidate facts. Then Map stage is leveraged to map the candidate facts to the schema of TAC KBP and Wikidata respectively. The parameter settings, such as beam size in Algorithm 1, are shared across TAC KBP and Wikidata based on the parameter study in Sec. A.3.

#### 3.1.2 TAC KBP

To verify the ability to produce correct facts, we compare candidate facts from MAMA to the outputs of two open information systems, which also produce triplets in the form of $(h, r, t)$. After collecting the triplets from the corresponding system, we use the same Map procedure with MAMA (Sec. 2.2.1) to map the triplets to the corresponding KG schema.

**Stanford OpenIE** leverages POS tag and dependency parser, and generates self-contained clauses from long sentences to extract the triplets, which is the best open information extraction system (Angeli et al., 2015) on TAC KBP (Surdeanu, 2013).

**OpenIE 5.1** [2] is one of the state-of-the-art open information extraction systems, which is the successor to Ollie (Schmitz et al., 2012), and it improves extractions from noun relations, numerical sentences, and conjunctive sentences depending on the linguistic patterns.

We use two families of pre-trained LMs with MAMA. We use BERT$_{\text{BASE}}$ and BERT$_{\text{LARGE}}$ from Devlin et al. (2018) with MAMA, namely **MAMA-BERT$_{\text{BASE}}$** and **MAMA-BERT$_{\text{LARGE}}$**. Besides, GPT-2s from Radford et al. (2019) are used, i.e., **MAMA-GPT-2**, **MAMA-GPT-2$_{\text{MEDIUM}}$**, **MAMA-GPT-2$_{\text{LARGE}}$**, and **MAMA-GPT-2$_{\text{XL}}$**.

---

[2] https://github.com/dair-iitd/OpenIE-standalone

| Method | #Params of LM | Precision% | Recall% | F1% |
|---|---|---|---|---|
| OpenIE 5.1 [2] | - | 56.98 | 14.54 | 23.16 |
| Stanford OpenIE (Angeli et al., 2015) | - | 61.55 | 17.35 | 27.07 |
| MAMA-BERT$_{\text{BASE}}$ (ours) | 109M | 61.57 | 18.79 | 28.79 |
| MAMA-BERT$_{\text{LARGE}}$ (ours) | 335M | 61.69 | 18.99 | 29.05 |
| MAMA-GPT-2 (ours) | 117M | 61.62 | 18.17 | 28.07 |
| MAMA-GPT-2$_{\text{MEDIUM}}$ (ours) | 345M | 62.10 | 18.65 | 28.69 |
| MAMA-GPT-2$_{\text{LARGE}}$ (ours) | 774M | 62.38 | 19.00 | 29.12 |
| MAMA-GPT-2$_{\text{XL}}$ (ours) | 1558M | 62.69 | 19.47 | 29.72 (+2.65) |

Table 2: Compare the quality of mapped facts on TAC KBP. #Params of LM refers to the number of parameters of the pre-trained LM.

Table 2 shows the results on TAC KBP. We use the official scorer of TAC KBP Slot Filling 2013 to evaluate precision, recall, and F1 on TAC KBP [3].

**MAMA constructs improved KGs compared to open IE**. From the results, we find that all our methods achieve competitive precision, which is greater than 60%, given the unsupervised nature of MAMA. All the proposed methods outperform the two open IE systems. This shows that MAMA is able to produce high-quality knowledge directly from pre-trained LMs by a single forward pass without human supervision. The results show the effectiveness of MAMA in generating candidate facts from Match stage, and producing high-quality KGs through Map stage. We also find that MAMA-GPT-2$_{\text{XL}}$ performs the best. MAMA-GPT-2$_{\text{XL}}$ outperforms the previous state-of-the-art Stanford OpenIE by over 2.6% in F1. This shows the proposed end-to-end MAMA is able to recover the knowledge stored in pre-trained LMs without relying on any extra linguistic features, such as POS tag and dependency parser used in open IE systems. The main reason leading to the moderate results of OpenIE 5.1 is that the system generates objects of the triplets with extraneous words, which hurt the performance in slot filling tasks. Even though the proposed methods all outperform the two open IE systems in the recall, however improving recall is clearly the future direction to further improve the performance of MAMA. We find that the main cause of the moderate recall is the incorrect entities caused by spaCy noun chunk as summarized in Sec. A.2.

**Larger/deeper LMs produce KGs of higher quality**. BERT$_{\text{LARGE}}$ outperforms BERT$_{\text{BASE}}$ since the doubling parameter size. GPT-2s share similar trends, where we observe performance increases when the model size increases. This complies with our intuition on more knowledge is stored in deeper and larger models. Such increases in performance seem subtle on TAC KBP, we find this might due to the relatively small number of oracle facts by noticing a more significant improvement on Wikidata in Sec. 3.1.3. We plan to further improve the results with larger pre-trained LMs, e.g., GPT-3 (Brown et al., 2020), Megatron-LM (Shoeybi et al., 2019).

**BERT LMs outperform GPT-2 LMs under similar model sizes**. More specifically, BERT$_{\text{BASE}}$ performs better than MAMA-GPT-2 in F1, and MAMA-BERT$_{\text{LARGE}}$ outperforms MAMA-GPT-2$_{\text{MEDIUM}}$ in F1. BERT$_{\text{BASE}}$ and MAMA-GPT-2 are similar in size, while MAMA-BERT$_{\text{LARGE}}$ and MAMA-GPT-2$_{\text{MEDIUM}}$ are similar in model size as well. This is mainly because that the recall of BERT LMs is higher than that of corresponding GPT-2 LMs. The result indicates that the Cloze-style loss function (i.e., masked language model) of BERT is more effective and flexible in recovering more knowledge than the autoregressive LM objective. We also notice that the precision of GPT-2 LMs is higher than that of according BERT LMs. The reason is that the autoregressive LM objective captures more accurate knowledge than Cloze-style loss does by not introducing extra noise (e.g., masked tokens) during pre-training.

### 3.1.3 WIKIDATA

We select our best BERT based method MAMA-BERT$_{\text{LARGE}}$, and GPT-2 based method MAMA-GPT-2$_{\text{XL}}$ on TAC KBP to compare with Stanford OpenIE (the best open IE system on TAC KBP) for scalability experiments on Wikidata. We follow the same definition as the slot filling task to calculate precision, recall, and F1 on Wikidata. Table 3 summarizes the results.

**MAMA is scalable to larger corpora**. Similar to the trends on TAC KBP, MAMA-GPT-2$_{\text{XL}}$ performs the best in precision, recall, and F1. The results show the effectiveness of MAMA in generating candidate facts and high-quality KGs. We also find that MAMA-GPT-2$_{\text{XL}}$ outperforms MAMA-BERT$_{\text{LARGE}}$ by over 1% in F1. This shows that the larger model (GPT-2$_{\text{XL}}$ has 5x more

---

[3]There are 2,383 correct oracle facts based on the "manual runs" assessment in TAC KBP.

| Method | #Params of LM | Precision% | Recall% | F1% |
|---|---|---|---|---|
| Stanford OpenIE (Angeli et al., 2015) | - | 23.32 | 13.09 | 16.77 |
| MAMA-BERT$_{\text{LARGE}}$ (ours) | 335M | 29.52 | 16.56 | 21.22 |
| MAMA-GPT-2$_{\text{XL}}$ (ours) | 1558M | 31.32 | 17.42 | **22.39 (+5.62)** |

Table 3: Compare the quality of mapped facts on Wikidata. #Params of LM refers to the number of parameters of the pre-trained LM.

parameters compared to BERT$_{\text{LARGE}}$) contains more knowledge, and MAMA is able to restore the knowledge. When larger or deeper models (e.g., GPT-3) are used with MAMA, we can expect more gains of the KG quality. Thanks to the efficient nature of MAMA, which relies only on the forward pass of the LMs without fine-tuning, the results suggest that MAMA is scalable to large KGs.

**Larger corpora embed more complete KGs**. In particular, MAMA-GPT-2$_{\text{XL}}$ outperforms Stanford OpenIE by 5.6% in F1. MAMA-BERT$_{\text{LARGE}}$ outperforms Stanford OpenIE by approximately 4.4% in F1. Both F1 gains are larger compared to that on TAC KBP. This is because that the LMs contain world knowledge from pre-training corpora, e.g. Wikipedia and Common Crawl. The larger the textual corpora are, the more knowledge our method is able to recover and match to the knowledge stored in LMs. The finding is particularly important, since we are now able to construct larger KGs of high quality from scratch when larger datasets are used, such as WebText2 and Common Crawl (Raffel et al., 2019; Brown et al., 2020). Similar to the observations on TAC KBP, the precision is higher compared to recall. Wikidata is not fully built from Wikipedia, MAMA could improve the recall by running on those larger corpora to collect more facts.

### 3.2 ANALYSIS OF UNMAPPED FACTS

The open KG constructed by MAMA is a new type of KG combining the fixed KG schema with the flexible open schema. We turn to study the quality of the candidate facts that are not mapped to the above reference KG schema, but are in the open schema generated by MAMA. We manually judge such unmapped facts generated by our best method MAMA-GPT-2$_{\text{XL}}$ from 100 sampled documents in Wikidata and TAC KBP respectively.

**The quality of unmapped facts is verified by human annotators**. We find 35.3% of the unmapped facts are true on Wikidata. We find 83.2% of those true facts are partially unmapped facts as defined in Sec. 2.2.2, e.g., *(Bob_Dylan.Q392, tour_with, the_Grateful_Dead.Q212533)*, whose relation is not within the schema of Wikidata, while both head and tail are in the schema. The remaining true facts are completely unmapped facts (Sec. 2.2.2), e.g., a candidate fact *(Jacob, was, A Registered Mennonite)* stays the same in the open schema.

**Accurate entity detection is desired**. We also notice 45.5% of the untrue unmapped facts on Wikidata are due to the *incorrect entities* detected by the spaCy. *Incorrect or missing entity linking* (to either head or tail) in Sec. 2.2.1 causes additional 9.1% errors in the unmapped facts. 4.5% of the untrue unmapped facts are caused by the *missing relation mapping* in Sec. 2.2.1. The rest errors made by MAMA-GPT-2$_{\text{XL}}$ are *incorrect relation phrases*, such as uninformative relation phrases, e.g., *(Dylan, made, his breakthrough)*, which is similar to the errors made by open IE systems (Fader et al., 2011). Both entity linking and relation mapping of Map stage rely heavily on the accuracy of entity detection from the spaCy noun chunk. We conclude that the main root cause of the untrue unmapped facts is due to the errors made by the spaCy noun chunk.

We observe similar trends on TAC KBP. We plan to leverage crowdsourcing platforms, e.g., Mechanical Turk, to conduct quantitative evaluations over the unmapped facts to better understand the strengths and shortage of MAMA. We plan to identify more accurate entities by relying on attention weights in LMs (Clark et al., 2019; Hewitt & Manning, 2019) instead of using extra resources. We will also investigate stronger entity linkers (Kolitsas et al., 2018) and learn a more robust relation mapping through weak or distant supervision (Mintz et al., 2009; Ratner et al., 2017). We will investigate more sophisticated approaches, such as graph neural networks (Kipf & Welling, 2016), to generate more accurate relation phrases from the attention weight matrices by considering structural information.

## 4 RELATED WORK

**Knowledge graph construction** can be generally categorized into two groups, 1) supervised approaches. Wikidata, Freebase (Bollacker et al., 2008), YAGO (Suchanek et al., 2007), YAGO2 (Hof-

fart et al., 2013), DBpedia (Auer et al., 2007) are built based on human supervision from Wikipedia infoboxes and other structured data sources; 2) semi-supervised approaches. Open information extraction systems, e.g., OLLIE (Schmitz et al., 2012), Reverb (Fader et al., 2011), Stanford OpenIE (Angeli et al., 2015), and OpenIE 5.1 [2] aim to leverage carefully-designed patterns based on linguistic features (e.g., dependencies and POS tags), to extract triplets from web corpora for open schema KG. Besides, NELL (Carlson et al., 2010), DeepDive (Niu et al., 2012), Knowledge Vault (Dong et al., 2014) extract information based on a fixed schema or ontology, where humans help improve the accuracy of the extractions. Probase (Wu et al., 2012) produces taxonomies instead of rich typed relations in general KGs. MAMA instead uses learned knowledge stored in pre-trained LMs without human supervision to construct an open KG, which is a mixture of fixed schema and open schema. Different from commonsense knowledge construction using Transformers (Davison et al., 2019; Bosselut et al., 2019), the proposed method is unsupervised and end-to-end, and constructs general-purpose KGs instead of commonsense knowledge.

**Language models**, e.g., BERT (Devlin et al., 2018), GPT (Radford et al., 2018), GPT-2/3 (Radford et al., 2019; Brown et al., 2020), ELMo (Peters et al., 2018), Transformer-XL (Dai et al., 2019), ALBERT (Lan et al., 2019), RoBERTa (Liu et al., 2019), XLNet (Yang et al., 2019) and Megatron-LM (Shoeybi et al., 2019) contain factual knowledge obtained via pre-training on large-scale corpora such as Wikipedia and BookCorpus (Zhu et al., 2015). Studies have leveraged the pre-trained LMs as virtual KGs, and show reasonable performance in QA tasks (Dhingra et al., 2020; Guu et al., 2020), and language modeling (Khandelwal et al., 2019). LMs are further enhanced by KGs (Peters et al., 2019) to improve knowledge-driven tasks. While the existing work utilizes knowledge in an implicit way, the main difference is that our approach explicitly extracts knowledgeable facts from the LMs. Compare to the joint training with knowledge base to improve shallow word embedding (Wang et al., 2014), we show that the knowledge is already stored in the deep LMs. We plan to incorporate domain knowledge into language models to construct domain-specific KGs. The main difference between LAMA (Petroni et al., 2019; 2020) and MAMA is mainly two-fold: (1) LAMA aims to complete Cloze-style statement, e.g., given "Dylan is a _", LAMA predicts which words/phrases should fill the blank "_", which has no direct connection to KGs. There are several fundamental limitations when adapting LAMA to construct KGs, e.g., additional queries must be constructed first, and the answers for the queries must be linked to KGs. MAMA aims to solve a reasoning problem, e.g., given a passage, MAMA directly matches the fact in the form of a triplet *(Dylan, is, songwriter)* at the first step, then maps the fact to produce a KG. (2) The benchmark datasets used with MAMA are larger compared to the LAMA benchmark, e.g., Wikidata is 3 orders of magnitude larger compared to the largest dataset in the LAMA benchmark.

**Neural network interpretation** here specifically refers to pre-trained deep language model analysis. There has been a lot of work to understand what the neural networks learn (Linzen et al., 2016; Adi et al., 2016; Tenney et al., 2019). With regards to analyzing Transformer (Vaswani et al., 2017) based language models (e.g., BERT and GPT-3), substantial recent work focuses on both visualizing and analyzing the attention (Vig, 2019; Jain & Wallace, 2019; Clark et al., 2019; Michel et al., 2019; Vig et al., 2020; Ramsauer et al., 2020; Hendrycks et al., 2020). Instead of analyzing or visualizing, we use LMs to generate structured KGs to directly recover what LMs learn from the corpora.

## 5 CONCLUSION

We show that the knowledge graphs can be constructed by a single forward pass of the language models over textual corpora. We propose a two-stage unsupervised approach MAMA to first match the facts in the corpus with the internal knowledge of the language model, and then map the matched facts to produce a knowledge graph. We demonstrate the quality of the resultant open knowledge graphs by comparing to two knowledge graphs (Wikidata and TAC KBP). The open knowledge graph also features new facts in the open schema, which could have broad implications for knowledge graphs and their downstream applications. The results also suggest that larger language models store richer knowledge than existing knowledge graphs, and generating on even larger high-quality text corpora could continue improving knowledge graphs. Additionally, the knowledge graphs generated by our approach can help researchers to look into what the language models learn, so our interpretable knowledge graphs establish a bridge between the deep learning and knowledge graph communities.

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

# A  ADDITIONAL DETAILS AND ANALYSIS OF MAMA

## A.1  METHOD DETAILS

**Map stage details**  To evaluate the mapped facts, we first use Match stage of MAMA to run over the corresponding documents to generate the candidate facts. For Map stage on TAC KBP, we link to the oracle annotation of the entities or spans in the TAC KBP corpus. On Wikidata, the entity linking method described in Sec. 2.2.1 is first leveraged to link entities in the candidate facts to Wikipedia anchors. We build an enhanced mention-to-entity dictionary based on Spitkovsky & Chang (2012). In particular, we add new Wikipedia anchors to the dictionary which results in 26 million entries comparing to 21 million entries in Spitkovsky & Chang (2012). Then a Wikipedia anchor to the Wikidata item dictionary is constructed and used to further link the entities to Wikidata. If the head or tail is a pronoun, we further use neuralcoref [4] for coreference resolution. We use GloVe (Pennington et al. 2014) embedding for disambiguation. The relation mapping is constructed offline for TAC KBP and Wikidata respectively using the method in Sec. 2.2.1. Besides the automatic relation mapping method proposed in Angeli et al. (2015), we manually check whether the top relation phrases are true as described in Sec. 2.2.1. For relation mapping, we randomly sampled a hold-out dataset including 2,000 documents from the TAC KBP corpus and English Wikipedia for the relation mapping construction on TAC KBP and Wikidata respectively. For oracle facts in Wikidata, we only preserve those facts describing relations between entities that could be linked to corresponding Wikipedia anchors. We rule out facts of attributes about entities and facts of auxiliary relations (such as *topic's main category.P901*) and finally results in 27,368,562 oracle facts.

**Implementation details**  For Wikidata, at Match stage, we randomly split the English Wikipedia data into 20 partitions, and map the data partitions to 20 distributed servers to run. Each server is configured with four Tesla K80 12Gs. We set the max sequence length to 256, and batch size as 32 for MAMA-BERT$_{\text{LARGE}}$ and 4 for MAMA-GPT-2$_{\text{XL}}$. We use implementations of pre-trained LMs in Transformers package [5]. We use spaCy sentencizer [6] to segment the documents into sentences. MAMA-BERT$_{\text{LARGE}}$ takes approximately 48 hours, and MAMA-GPT-2$_{\text{XL}}$ costs around 96 hours. The resulting candidate facts of Match stage from the 20 servers are then reduced a data server, where a MongoDB database is maintained to store the oracle Wikidata and entity linking results to enable the efficient Map stage. To produce the open KGs, Map stage takes around 18 hours. The setup is similar to TAC KBP. Match stage is done within 48 hours for all the settings. The batch sizes of MAMA-BERT$_{\text{BASE}}$, MAMA-GPT-2, MAMA-GPT-2$_{\text{MEDIUM}}$, MAMA-GPT-2$_{\text{LARGE}}$ are 64, 32, 16, 8 respectively.

**Parameter settings**  The parameter settings are shared across TAC KBP and Wikidata. All the choices are based on the parameter study in Sec. A.3. The beam size of Algorithm 1 is set to 6. The matching degree threshold of Constraint #1 (Sec. 2.1.2) is set to 0.005, and the number of distinct head-tail pairs of Constraint #2 (Sec. 2.1.2) is set to 10. To generate the attention weight matrix $\mathbf{A}_s$ of a sentence, we reduce the weights of every attention head in the last layer of pre-trained LMs using the mean operator.

## A.2  ERROR ANALYSIS

There is still significant room to improve MAMA. To further understand the shortage of MAMA, we conduct an error analysis of the errors in precision (i.e., incorrect facts returned by MAMA) of Table 2 and Table 3. We choose our best method MAMA-GPT-2$_{\text{XL}}$ for the study. We sample 100 documents from the Wikidata dataset, and manually check the reasons for the errors. We find 33.1% of the errors are caused by *incorrect entities*, while the relation phrases are correct. The errors are due to the incorrect noun chunk detected by the spaCy [7]. 18.3% of the errors are due to the *missing relation mapping* created in Sec. 2.2.1. Note that we find approximately 23.8% of the errors are actually *correct facts that are new in the reference KGs*. e.g., *(Bob_Dylan.Q392, residence.P551, Nashville.Q23197)* (in Figure 16) is not an existing fact in Wikidata, but it is a correct mapped fact based on our annotation. The rest errors made by MAMA-GPT-2$_{\text{XL}}$ are *incorrect relation phrases*,

---

[4] https://github.com/huggingface/neuralcoref
[5] https://github.com/huggingface/transformers
[6] https://spacy.io/api/sentencizer
[7] https://spacy.io/usage/linguistic-features/#noun-chunks

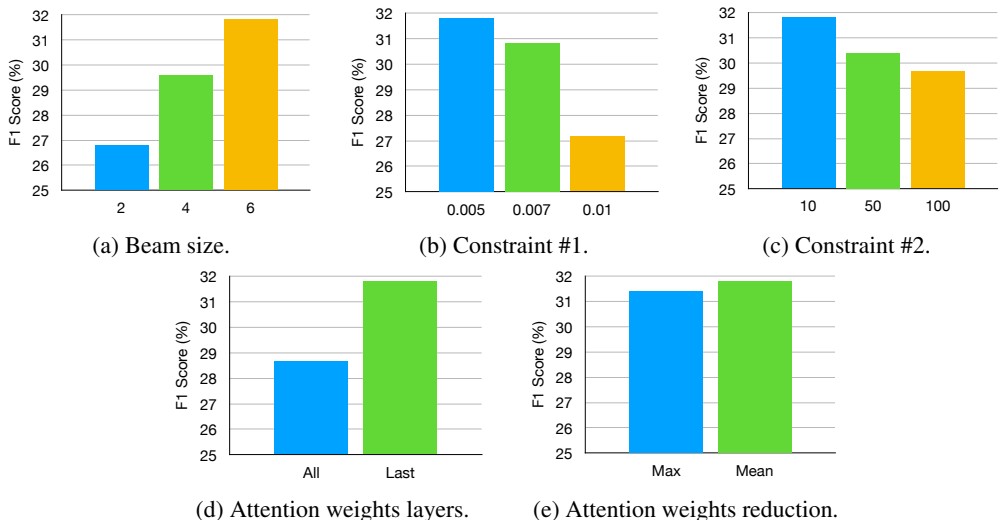

Figure 3: Parameter study with MAMA-BERT$_{\text{BASE}}$ on TAC KBP hold-out subset.

such as uninformative relation phrases. We find similar errors are made by MAMA-GPT-2$_{\text{XL}}$ on TAC KBP. Similar to Sec. 3.2, enhancing the entity detection, entity linker, relation mapping, and relation generation are helpful. We also plan to leverage lifelong learning (Carlson et al., 2010) to add true facts to the reference KGs to improve the evaluation.

## A.3 PARAMETER STUDY

We study the effects of the parameters using MAMA-BERT$_{\text{BASE}}$ on TAC KBP. We randomly sample 20% of the oracle query entities as a hold-out dataset to tune the parameters, and use the best parameter setting achieved for both TAC KBP and Wikidata experiments. When studying the effect of a certain parameter, we keep the remaining parameters as default described in Sec. A.1. We use F1 to measure the effects.

**Effects of beam size**  Figure 3(a) illustrates the effects of various beam sizes in Algorithm 1. We find that in general, the larger the beam size is, the better F1 the setting achieves. This is because that MAMA is able to reserve more potentially correct facts when more candidates are allowed in the Match stage. However, F1 improvement gradually becomes subtle, while the computation costs increase more significantly. For sake of the efficiency, we do not explore larger beam sizes. We set the beam size as 6.

**Effects of search constraints**  Figure 3(b) compares the effect of different thresholds of the matching degree of Constraint #1 in Sec. 2.1.2. We set the threshold as 0.005 since it achieves the best result. Note that the summed attention score is normalized by the length of the fact to penalize the cumbersome facts. The matching degree threshold is effective, which is mainly because of the knowledge contained in the self-attention matrix. The score in the attention matrix is representing the chance of the facts to be the true facts based on the stored knowledge. Figure 3(c) shows the impact of the number of distinct head-tail pairs in identifying common relations of Constraint #2 in Sec. 2.1.2. The best result is achieved when it equals 10. This shows that while MAMA mostly identifies frequent relations, it is also able to capture some rare relations for the open schema.

**Effects of attention weights**  Figure 3(d) shows the comparison between attention weights of the last layer and the mean of all layers. The attention weights of the last layer perform better. This is due to the attention weights in lower layers are low-level linguistic knowledge according to (Clark et al., 2019; Ramsauer et al., 2020), which are less relevant to the factual knowledge for the KG construction. Figure 3(e) compares the impact of different attention reduction, i.e., mean, max, over the attention heads of the last layer. We find the "mean" perform better. The reason is that the token often intensively attends to several specific tokens in the sequence (Michel et al., 2019), and the "mean" operator is more sensitive to such information.

## B    SAMPLES FROM MAMA ON TAC KBP

### B.1    MAPPED FACTS

We randomly sample 100 documents from TAC KBP corpus, then randomly sample sentences from those documents. The uncurated candidate facts and the corresponding mapped facts of the sampled sentences based on our best methods MAMA-BERT$_{\text{LARGE}}$ and MAMA-GPT-2$_{\text{XL}}$ are shown in Figure 4 and Figure 5 respectively. We also randomly sample several sentences in which MAMA-BERT$_{\text{LARGE}}$ differs from MAMA-GPT-2$_{\text{XL}}$ in the resulting facts for comparison, which is illustrated in Figure 6. In each table, "**ID**" represents the document ID of a sampled sentence in TAC KBP corpus. "**Sentence**" indicates the sampled sentence. "**Candidate facts to mapped facts**" column contains the candidate facts (on the left side of "→") and their corresponding mapped facts (on the right side of "→").

### B.2    UNMAPPED FACTS

We randomly sample 100 documents from TAC KBP corpus. From those documents, we show unmapped facts from the sampled sentences from those documents. We manually check the correctness of the unmapped facts according to Sec. 3.2, and show the correct ones. The original candidate facts with the corresponding unmapped facts of the sampled sentences generated by MAMA-BERT$_{\text{LARGE}}$ and MAMA-GPT-2$_{\text{XL}}$ are shown in Figure 7 and Figure 8. A further comparison of the unmapped candidate facts is illustrated in Figure 9. In each table, "**ID**" represents the document ID of a sampled sentence in TAC KBP corpus. "**Sentence**" indicates the sampled sentence. "**Candidate facts to unmapped facts**" column contains the candidate facts (on the left side of "→") and their corresponding unmapped facts (on the right side of "→").

## C    SAMPLES FROM MAMA ON WIKIDATA

### C.1    MAPPED FACTS

Similar to TAC KBP, we randomly sample 100 documents from the Wikidata corpus (i.e., English Wikipedia), then randomly sample sentences from those documents. The uncurated candidate facts and the corresponding mapped facts of the sampled sentences based on our best methods MAMA-BERT$_{\text{LARGE}}$ and MAMA-GPT-2$_{\text{XL}}$ are shown in Figure 10 and Figure 11 respectively. We also randomly sample several sentences in which MAMA-BERT$_{\text{LARGE}}$ differs from MAMA-GPT-2$_{\text{XL}}$ in the resulting facts for comparison, which is illustrated in Figure 12. In each table, "**ID**" represents the Wikipedia page's title of a sampled sentence. "**Sentence**" indicates the sampled sentence. "**Candidate facts to mapped facts**" column contains the candidate facts (on the left side of "→") and their corresponding mapped facts (on the right side of "→").

### C.2    UNMAPPED FACTS

Similar to TAC KBP, we randomly sample 100 documents from the Wikidata corpus. From those documents, we show unmapped facts from several sampled sentences from those documents. We manually check the correctness of the unmapped facts according to Sec. 3.2, and show the correct ones. The original candidate facts with the corresponding unmapped facts of the sampled sentences generated by MAMA-BERT$_{\text{LARGE}}$ and MAMA-GPT-2$_{\text{XL}}$ are shown in Figure 13 and Figure 14. A further comparison of the unmapped candidate facts is illustrated in Figure 15. In each table, "**ID**" represents the Wikipedia page's title of a sampled sentence. "**Sentence**" indicates the sampled sentence. "**Candidate facts to unmapped facts**" column contains the candidate facts (on the left side of "→") and their corresponding unmapped facts (on the right side of "→").

## D    ADDITIONAL OPEN KG SUBGRAPHS FROM MAMA ON WIKIDATA

We sample several documents from the Wikidata corpus. We visualize the mapped facts and unmapped facts from those documents as examples of subgraphs in the resulting open KGs. We show the snapshots of the subgraphs generated by MAMA-BERT$_{\text{LARGE}}$ from Figure 16 to Figure 24. We

| ID | Sentence | Candidate facts to mapped facts |
|---|---|---|
| SF13_ENG_001 | Bashardost left Pakistan for France in 1981 . | (Bashardost, left, Pakistan) → (Ramazan Bashardost, per:countries_of_residence, Pakistan) |
| SF13_ENG_004 | " Douglas Flint will succeed Stephen Green as Group Chairman and Stuart Gulliver will be appointed Group Chief Executive , following Michael Geoghegan ' s decision to retire early next year , " HSBC said in a statement . | (Douglas Flint, as, Group Chairman) → (Douglas Flint, per:title, Chairman) |
| SF13_ENG_007 | Mohammed Tantawi , head of Al - Azhar University , told a schoolgirl to remove her niqab when he spotted her during a tour of an Al - Azhar affiliated school , the independent Al - Masry al - Youm newspaper reported this week . | (Mohammed Tantawi, head of, Al-Azhar University) → (Mohammed Sayed Tantawi, per:employee_or_member_of, Al-Azhar University) |
| SF13_ENG_009 | He took office in 2006 by defeating longtime incumbent Anwar Chowdhry from Pakistan , who was later barred for alleged financial corruption . | (Longtime Incumbent Anwar Chowdhry, from, Pakistan) → (Anwar Chowdhry, per:origin, Pakistan) |
| SF13_ENG_010 | In addition to his wife , Wendy , Dio is survived by son Daniel , grandchildren Julie and Joey , and father Pat . | (Dio, is survived by, son Daniel) → (Ronnie James Dio, per:children, Daniel) |
| SF13_ENG_011 | Marshall is charged with grand larceny and fraud and faces up to 25 years in prison if convicted . | (Marshall, is charged, Fraud) → (Anthony Marshall, per:charges, fraud) |
| SF13_ENG_011 | Marshall , a Tony Award - winning Broadway producer and former U . S . diplomat , sat stonelike as the jury forewoman read each verdict aloud , the word " guilty " clearly resonating in the otherwise silent courtroom . | (Marshall, , A Tony Award-Winning Broadway Producer) → (Anthony Marshall, per:title, producer) |
| SF13_ENG_011 | The blueblood scion of one of America ' s most illustrious families appeared to listen impassively as verdicts finding him guilty on 14 counts of grand larceny , conspiracy and fraud were read to a packed courtroom . | (Him, guilty, Grand Larceny) → (Anthony Marshall, per:charges, grand larceny) |
| SF13_ENG_011 | But Marshall ' s son , Philip , told a different story . | (Marshall'S Son, ,, Philip) → (Anthony Marshall, per:children, Philip) |
| SF13_ENG_012 | Al - Qaeda ' s American spokesman Adam Gadahn , also known as Azzam the American , called on Muslims in the West on Sunday to carry out more attacks like the deadly shooting at the US base in Fort Hood , Texas . | (Adam Gadahn, also known, Azzam) → (Adam Gadahn, per:alternate_names, Azzam) |
| SF13_ENG_012 | Gadahn , also known as Azzam the American , was born in 1978 . | (Gadahn, , also known as, Azzam) → (Adam Gadahn, per:alternate_names, Azzam) |
| SF13_ENG_012 | Gadahn grew up in California and converted to Islam before he moved to Pakistan in 1998 and attended an al - Qaida training camp six years later , according to media reports . | (Gadahn, in, California) → (Adam Gadahn, per:statesorprovinces_of_residence, California) |
| SF13_ENG_012 | Gadahn moved to Pakistan in 1998 , according to the FBI , and is said to have attended an al - Qaida training camp six years later , serving as a translator and consultant for the group . | (Gadahn, moved to, Pakistan) → (Adam Gadahn, per:countries_of_residence, Pakistan) |
| SF13_ENG_014 | After the Munich attack , he lived in Lebanon , Jordan and several Eastern European countries , where he had close ties to Communist bloc intelligence agencies . | (He, lived in, Lebanon) → (Mohammed Oudeh, per:countries_of_residence, Lebanon) |
| SF13_ENG_015 | Clifton attended Howard University but left before graduating to pursue poetry . | (Clifton, attended, Howard University) → (Lucille Clifton, per:schools_attended, Howard University) |
| SF13_ENG_017 | " Alexander Haig devoted his career to serving our country , both as a soldier and as a diplomat , " Albright said . " | (Alexander Haig, devoted his, soldier) → (Alexander Haig, per:title, soldier) |
| SF13_ENG_017 | In 1979 , he resigned and retired from the Army . | (He, resigned and retired from, The Army) → (Alexander Haig, per:employee_or_member_of, Army) |
| SF13_ENG_019 | McGregor is survived by his wife , Lori , and four children , daughters Jordan , Taylor and Landri , and a son , Logan . | (Mcgregor, his wife, Lori) → (Keli McGregor, per:spouse, Lori) |
| SF13_ENG_020 | " Mike was a first - rate journalist , a valued member of our staff for 25 years and we will miss him , " Times Editor Russ Stanton said . " | (Mike, was, A First-Rate Journalist) → (Mike Penner, per:title, first-rate journalist) |
| SF13_ENG_020 | Penner is survived by his brother , John , a copy editor at the Times , and his former wife , Times sportswriter Lisa Dillman . | (Penner, his brother ,, John) → (Mike Penner, per:siblings, John) |
| SF13_ENG_024 | She was charged with theft in Beaumont , Texas , for allegedly failing to pay for $ 10 , 000 worth of dental work in 2006 . | (She, was charged, Theft) → (Crystal Taylor, per:charges, theft) |
| SF13_ENG_025 | Michigan native Nancy Kissel was convicted of murder and sentenced in Hong Kong ' s High Court in September 2005 . | (Michigan Native Nancy Kissel, was convicted, Murder) → (Nancy Kissel, per:charges, murder) |
| SF13_ENG_026 | Neal returned to New York and concentrated on stage work . | (Neal, returned to, New York) → (Patricia Neal, per:statesorprovinces_of_residence, New York) |
| SF13_ENG_026 | In 1953 , she married Roald Dahl , the British writer famed for " Charlie and the Chocolate Factory , " " James and the Giant Peach " and other tales for children . | (She, married, Roald Dahl) → (Patricia Neal, per:spouse, Roald Dahl) |
| SF13_ENG_026 | Oscar - winning actress Patricia Neal has died of lung cancer at her home on Martha ' s Vineyard , Massachusetts , on Sunday . | (Patricia, died of, Lung Cancer) → (Patricia Neal, per:cause_of_death, lung cancer) |
| SF13_ENG_027 | Al - Hakim ' s son , Ammar al - Hakim , has been groomed for months to take his father ' s place . | (Al-Hakim'S Son, ,, Ammar Al-Hakim) → (Abdul Aziz Al-Hakim, per:children, Ammar al-Hakim) |
| SF13_ENG_027 | Al - Hakim is the head of Supreme Iraqi Islamic Council ( SIIC ) , the largest Shiite party in Iraq . | (Al-Hakim, is, Supreme Iraqi Islamic Council) → (Abdul Aziz Al-Hakim, per:employee_or_member_of, Supreme Iraqi Islamic Council) |
| SF13_ENG_027 | His former Shiite partners have gathered again to form their own group , the Iraqi National Alliance ( INA ) , which includes the influential Supreme Iraqi Islamic Council ( SIIC ) of Ammar al - Hakim , who succeeded his father Abdul Aziz al - Hakim , who died in a hospital in Iran last month after a long battle with cancer . | (Al-Hakim, , who died in, Iran) → (Abdul Aziz Al-Hakim, per:country_of_death, Iran) |
| SF13_ENG_028 | " I ' d rather have Sully doing this than some stranger , or some hotshot trying to be the next Billy Mays , " said the guy who actually is the next Billy Mays , his son Billy Mays III . | (The Next Billy Mays, his son, Billy Mays Iii) → (Billy Mays, per:children, Billy Mays III) |
| SF13_ENG_029 | Fignon continued cycling during and after a stint in the Army , and drew attention in the early 1980s when he managed to keep up with Hinault during a race in which amateurs and professionals rode together . | (Fignon, during and after a stint in, The Army) → (Laurent Fignon, per:employee_or_member_of, the Army) |
| SF13_ENG_029 | Laurent Patrick Fignon was born in Paris on Aug . 12 , 1960 . | (Laurent Patrick Fignon, was born in, Paris) → (Laurent Fignon, per:city_of_birth, Paris) |
| SF13_ENG_030 | Anderson became the Tigers ' manager in June 1979 and built on a foundation that included Alan Trammell at shortstop , Lou Whitaker at second base , Kirk Gibson in the outfield and Jack Morris on the pitching staff . | (Anderson, became, The Tigers' Manager) → (Sparky Anderson, per:title, manager) |
| SF13_ENG_030 | In addition to his wife , Carol , Anderson is survived by his sons , Lee and Albert ; his daughter , Shirlee Englebrecht ; and many grandchildren . | (Anderson, is survived by his sons, Lee) → (Sparky Anderson, per:children, Albert) |
| SF13_ENG_031 | Blake Edwards , a writer and director who was hailed as a Hollywood master of screwball farces and rude comedies like " Victor / Victoria " and the " Pink Panther " movies , died Wednesday night in Santa Monica , Calif . He was 88 . | (Blake Edwards, , a writer) → (Blake Edwards, per:title, writer) |
| SF13_ENG_032 | Hwang , who lives in Seoul under tight police security , has written books and delivered speeches condemning Kim ' s regime as authoritarian and dictatorial . | (Hwang, who lives, Seoul) → (Hwang Jang-Yop, per:cities_of_residence, Seoul) |
| SF13_ENG_035 | Kaczynska , who was 67 , married Kaczynski in 1978 after meeting him in the northern Polish city of Gdansk , where they were both academics . | (Kaczynska, married, Kaczynski) → (Maria Kaczynska, per:spouse, Kaczynski) |
| SF13_ENG_036 | Upon his release he went into exile in India , where he masterminded the 1973 hijacking of a Royal Nepal Airlines plane known to be carrying hundreds of thousands of dollars in cash to fund his banned Nepali Congress party . | (His Release, exile, India) → (Girija Prasad Koirala, per:countries_of_residence, India) |
| SF13_ENG_036 | Koirala began his political career as a union organiser and was imprisoned for seven years in 1960 after a failed uprising against the monarchy . | (Koirala, began his, A Union Organiser) → (Girija Prasad Koirala, per:title, union organiser) |
| SF13_ENG_036 | Koirala was born in 1925 in Bihar of India where his father Krishna Prasad Koirala and his family were living in exile . | (Koirala, was born in 1925 in, Bihar) → (Girija Prasad Koirala, per:city_of_birth, Bihar) |
| SF13_ENG_036 | Koirala was born in 1925 in Bihar of India at the time when his father Krishna Prasad Koirala along with his family was exiled by Rana rulers . | (Koirala, was born in 1925 in, Bihar) → (Girija Prasad Koirala, per:city_of_birth, Bihar) |
| SF13_ENG_037 | Chabrol ' s survivors also include his third wife , Aurore Pajot , who acted as his script supervisor on nearly all of his movies from 1968 on and whom he married in 1981 ; and Pajot ' s daughter , Cecile Maistre , who was an assistant director on his films and wrote the script with him for " The Girl Cut in Two " ( 2007 ) . | (Chabrol'S Survivors, third wife, Aurore Pajot) → (Claude Chabrol, per:spouse, Aurore Pajot) |
| SF13_ENG_038 | The joint statement said Cunningham was " an inspiring performer and dancer into his 80s , and a visionary choreographer and dedicated teacher throughout his life , he led quietly and by example , " the statement said . " | (Cunningham, was ", An Inspiring Performer) → (Merce Cunningham, per:title, performer) |
| SF13_ENG_038 | Merce Cunningham , the nonagenarian choreographer , is planning for a world without him . | (Merce Cunningham, ,, The Nonagenarian Choreographer) → (Merce Cunningham, per:title, choreographer) |
| SF13_ENG_039 | A court on Monday cleared the widower of British reality television star Jade Goody , who died of cancer last year , of rape . | (British Reality Television Star Jade, who died, Cancer) → (Jade Goody, per:cause_of_death, cancer) |
| SF13_ENG_041 | Don Hewitt the CBS newsman who invented the highly popular TV newsmagazine " 60 Minutes " and produced it for 36 years , died Wednesday . | (Don Hewitt, , The Cbs Newsman) → (Don Hewitt, per:title, newsman) |
| SF13_ENG_041 | " He was the consummate television newsman , " Don Hewitt , a longtime CBS News executive and creator of the long - running " 60 Minutes " news program , told Reuters . " | (Don Hewitt, , executive) → (Don Hewitt, per:title, executive) |
| SF13_ENG_041 | Hewitt was already a highly respected TV newsman . | (Hewitt, was, A Highly Respected Tv Newsman) → (Don Hewitt, per:title, TV newsman) |
| SF13_ENG_041 | Donald Shepard Hewitt was born in New York on Dec . 14 , 1922 , and grew up in the suburb of New Rochelle . | (Donald Shepard Hewitt, born, New York) → (Don Hewitt, per:stateorprovince_of_birth, New York) |
| SF13_ENG_043 | Eleanor Louise Greenwich was born in Brooklyn on Oct . 23 , 1940 . | (Eleanor Louise Greenwich, was born, Oct.) → (Ellie Greenwich, per:date_of_birth, 1940-10-23) |
| SF13_ENG_044 | A little more than a year after Dunne died from bladder cancer , the colorful remnants of his estate have been consigned by his family to Stair Galleries in Hudson , N . Y . , which will auction them Nov . 20 . | (Dunne, died, Bladder Cancer) → (Dominick Dunne, per:cause_of_death, bladder cancer) |
| SF13_ENG_045 | Charles Gwathmey , an architect known for his influential modernist home designs and famous clients like director Steven Spielberg , has died . | (Charles Gwathmey, ,, architect) → (Charles Gwathmey, per:title, architect) |
| SF13_ENG_049 | Besides his wife , Mandelbrot is survived by two sons , Laurent , of Paris , and Didier , of Newton , Mass . , and three grandchildren . | (Mandelbrot, is survived by two sons ,, Laurent) → (Benoit Mandelbrot, per:children, Laurent) |
| SF13_ENG_049 | For years , he worked for IBM in New York . | (He, for, Ibm) → (Benoit Mandelbrot, per:employee_or_member_of, IBM) |
| SF13_ENG_049 | After several years spent largely at the Centre National de la Recherche Scientifique in Paris , Mandelbrot was hired by IBM in 1958 to work at the Thomas J . Watson Research Center in Yorktown Heights , N . Y . Although he worked frequently with academic researchers and spent time as a visiting professor at Harvard and the Massachusetts Institute of Technology , it was not until 1987 that he began to teach at Yale , where he earned tenure in 1999 . | (Mandelbrot, was hired by, Ibm) → (Benoit Mandelbrot, per:employee_or_member_of, IBM) |
| SF13_ENG_056 | " It ' s an issue for everybody in the state because peanuts are a big part of our economy , " said Don Koehler , executive director of the Georgia Peanut Commission . " | (The Georgia Peanut Commission, director, Don Koehler) → (Georgia Peanut Commission, org:top_members_employees, Don Koehler) |
| SF13_ENG_060 | " We ' ll be meeting with scientists , university and science policy officials to explore practical opportunities for exchange and collaboration , " Agre , the AAAS president , was quoted as saying . | (Aaas President, ,, Agre) → (American Association for the Advancement of Science, org:top_members_employees, Peter C. Agre) |
| SF13_ENG_060 | However , Alan Leshner , chief executive officer of the American Association for the Advancement of Science , noted that Nobels are generally given for work that ' s a decade old or more , and that the U . S . mustn ' t become complacent . | (The American Association, chief executive, Alan Leshner) → (American Association for the Advancement of Science, org:top_members_employees, Alan Leshner) |
| SF13_ENG_062 | " First of all , they never have enough funding , " said Andy Kunz , president of the U . S . High Speed Rail Association , a nonprofit that advocates a national high - speed rail network . " | (The U.S. High Speed Rail Association, ,, Andy Kunz) → (U.S. High Speed Rail Association, org:top_members_employees, Andy Kunz) |
| SF13_ENG_064 | China ' s shock at NATO ' s military campaign in the former Yugoslavia helped prod Beijing into playing a bigger role in U . N . peacekeeping , said Bates Gill , director of the Stockholm International Peace Research Institute and co - author of a recent report on China ' s peacekeeping activities . | (The Stockholm International Peace Research Institute, director of, Bates Gill) → (Stockholm International Peace Research Institute, org:top_members_employees, Bates Gill) |
| SF13_ENG_064 | " Non - state actors , for example , a small group of pirates off the coast of Somalia , Al Qaida and Taliban who operate across borders and have more and more sophisticated means of violence , are becoming bigger and bigger challenges to the international system , " said Bates Gill , director of the Stockholm International Peace Research Institute . | (The Stockholm International Peace Research Institute, director of, Bates Gill) → (Stockholm International Peace Research Institute, org:top_members_employees, Bates Gill) |
| SF13_ENG_068 | " It is not surprising that one primarily cosmetic business is trying to throw another under the bus by transferring a tax from rich doctors and their wealthy customers to struggling small businesses , " John Overstreet , director of the Indoor Tanning Association , said in a statement Saturday . | (The Indoor Tanning Association, director of, John Overstreet) → (Indoor Tanning Association, org:top_members_employees, John Overstreet) |
| SF13_ENG_076 | The chairman of the Swiss Bankers Association , Patrick Odier , told weekly NZZ am Sonntag that Italy and France have shown interest in deals like ones Switzerland signed this week with Germany and Britain . | (The Swiss Bankers Association, ,, Patrick Odier) → (Swiss Bankers Association, org:top_members_employees, Patrick Odier) |
| SF13_ENG_076 | The majority of voters in Switzerland , which manages more than 25 percent of the world ' s foreign - held private wealth , support banking secrecy , according to a survey published last month by the Swiss Bankers Association in Basel . | (The Swiss Bankers Association, in, Basel) → (Swiss Bankers Association, org:city_of_headquarters, Basel) |
| SF13_ENG_078 | " Americans have a right to know the truth - - Islam is a religion of intolerance and violence , " said Richard Thompson , legal director of the Thomas More Law Center in Ann Arbor . " | (The Thomas More Law Center, in, Ann Arbor) → (Thomas More Law Center, org:city_of_headquarters, Ann Arbor) |
| SF13_ENG_082 | New solutions may be enacted for these orphans , though , said Mary Robinson , CEO of the National Council for Adoption . | (The National Council, ,, Mary Robinson) → (National Council for Adoption, org:top_members_employees, Mary Robinson) |
| SF13_ENG_082 | " When you close a country , you end up causing more problems than you prevented , " said Chuck Johnson , CEO of the National Council for Adoption . " | (Adoption, Council for, Chuck Johnson) → (National Council for Adoption, org:top_members_employees, Chuck Johnson) |
| SF13_ENG_084 | " This is definitely a Goldilocks problem , " said Jason Grumet , president of the Bipartisan Policy Center and an energy adviser to the Obama campaign last year . | (The Bipartisan Policy Center, ,, Jason Grumet) → (Bipartisan Policy Center, org:top_members_employees, Jason Grumet) |
| SF13_ENG_085 | " Banks are in strong need for the capital markets ( to raise funds ) , " Li Fuan , a director at the China Banking Regulatory Commission , was quoted as saying at a forum over the weekend . | (The China Banking Regulatory Commission, director at, Li Fuan) → (China Banking Regulatory Commission, org:top_members_employees, Li Fuan) |
| SF13_ENG_089 | RIA Novosti and Interfax cite Anatoly Isaikin , head of Rosoboronexport , as saying Thursday " nothing is blocking the continuation of military - technical cooperation " with Iran . | (Rosoboronexport, head of, Anatoly Isaikin) → (Rosoboronexport, org:top_members_employees, Anatoly Isaikin) |
| SF13_ENG_089 | Rosoboronexport is the only company in Russia that is allowed to export arms , dual - use products and military - related services . | (Rosoboronexport, is the only company in, Russia) → (Rosoboronexport, org:country_of_headquarters, Russia) |
| SF13_ENG_091 | With his wife , Cornelie , Middelhoff invested money in 2000 and 2001 with Esch in funds that were formed to buy five properties from KarstadtQuelle , as Arcandor was then known , and leased back to the department store chain before Middelhoff joined the company , according to Middelhoff ' s spokesman . | (Arcandor, ,, Karstadtquelle) → (Arcandor, org:alternate_names, KarstadtQuelle) |
| SF13_ENG_091 | Arcandor ' s mail - order subsidiary Quelle is in worse shape however , Klaus - Hubert Goerg told a press conference in the western city of Essen , where Arcandor is based . | (Arcandor, ,, Essen) → (Arcandor, org:city_of_headquarters, Essen) |

Figure 4: Mapped facts: MAMA-BERT$_{\text{LARGE}}$ on TAC KBP.

similarly illustrate the snapshots of the subgraphs constructed by MAMA-GPT-2$_{\text{XL}}$ from Figure 25 to Figure 32. In each figure, the blue node and arrow represent the mapped facts in the Wikidata schema, while the yellow node and arrow denote the unmapped facts in the open schema. We additionally visualize the correct facts that are new in Wikidata according to Sec. A.2 in yellow.

| ID | Sentence | Candidate facts to mapped facts |
|---|---|---|
| SF13_ENG_001 | But in a sign of just how disenchanted some Afghans have become with their government, Bashardost, a doctoral scholar who lived in France for two decades, is widely believed to be at least fourth in popularity among 42 candidates in the August elections. | (Bashardost, , A Doctoral Scholar) → (Ramazan Bashardost, per:title, scholar) |
| SF13_ENG_001 | Bashardost left Pakistan for France in 1981. | (Bashardost, left, Pakistan) → (Ramazan Bashardost, per:countries_of_residence, Pakistan) |
| SF13_ENG_007 | Mohammed Tantawi, head of Al-Azhar University, told a schoolgirl to remove her niqab when he spotted her during a tour of an Al-Azhar affiliated school, the independent Al-Masry al-Youm newspaper reported this week. | (Mohammed Tantawi, head, Al-Azhar University) → (Mohammed Sayed Tantawi, per:employee_or_member_of, Al-Azhar University) |
| SF13_ENG_009 | He took office in 2006 by defeating longtime incumbent Anwar Chowdhry from Pakistan, who was later barred for alleged financial corruption. | (Longtime Incumbent Anwar Chowdhry, from, Pakistan) → (Anwar Chowdhry, per:origin, Pakistan) |
| SF13_ENG_010 | In addition to his wife, Wendy, Dio is survived by son Daniel, grandchildren Julie and Joey, and father Pat. | (Dio, is survived by son, Daniel) → (Ronnie James Dio, per:children, Daniel) |
| SF13_ENG_011 | Marshall is charged with grand larceny and fraud and faces up to 25 years in prison if convicted. | (Marshall, with, Fraud) → (Anthony Marshall, per:charges, fraud) |
| SF13_ENG_011 | Marshall, a Tony Award-winning Broadway producer and former U.S. diplomat, sat stonelike as the jury forewoman read each verdict aloud, the word "guilty" clearly resonating in the otherwise silent courtroom. | (Marshall, , producer) → (Anthony Marshall, per:title, producer) |
| SF13_ENG_011 | The blueblood scion of one of America's most illustrious families appeared to listen impassively as verdicts finding him guilty on 14 counts of grand larceny, conspiracy and fraud were read to a packed courtroom. | (Him, guilty on, Grand Larceny) → (Anthony Marshall, per:charges, grand larceny) |
| SF13_ENG_011 | But Marshall's son, Philip, told a different story. | (Marshall'S Son, , Philip) → (Anthony Marshall, per:children, Philip) |
| SF13_ENG_012 | Al-Qaeda's American spokesman Adam Gadahn, also known as Azzam the American, called on Muslims in the West on Sunday to carry out more attacks like the deadly shooting at the US base in Fort Hood, Texas. | (Adam Gadahn, also known as, Azzam) → (Adam Gadahn, per:alternate_names, Azzam) |
| SF13_ENG_012 | Gadahn, also known as Azzam the American, was born in 1978. | (Gadahn, also known as, Azzam) → (Adam Gadahn, per:alternate_names, Azzam) |
| SF13_ENG_012 | Gadahn grew up in California and converted to Islam before he moved to Pakistan in 1998 and attended an al-Qaida training camp six years later, according to media reports. | (Gadahn, in, California) → (Adam Gadahn, per:statesorprovinces_of_residence, California) |
| SF13_ENG_012 | Gadahn moved to Pakistan in 1998, according to the FBI, and is said to have attended an al-Qaida training camp six years later, serving as a translator and consultant for the group. | (Gadahn, moved to, Pakistan) → (Adam Gadahn, per:countries_of_residence, Pakistan) |
| SF13_ENG_014 | After the Munich attack, he lived in Lebanon, Jordan and several Eastern European countries, where he had close ties to Communist bloc intelligence agencies. | (He, lived in, Lebanon) → (Mohammed Oudeh, per:countries_of_residence, Lebanon) |
| SF13_ENG_015 | Clifton attended Howard University but left before graduating to pursue poetry. | (Clifton, attended, Howard University) → (Lucille Clifton, per:schools_attended, Howard University) |
| SF13_ENG_017 | "Alexander Haig devoted his career to serving our country, both as a soldier and as a diplomat," Albright said. " | (Alexander Haig, as, soldier) → (Alexander Haig, per:title, soldier) |
| SF13_ENG_017 | In 1979, he resigned and retired from the Army. | (He, resigned and, The Army) → (Alexander Haig, per:employee_or_member_of, Army) |
| SF13_ENG_019 | McGregor is survived by his wife, Lori, and four children, daughters Jordan, Taylor and Landri, and a son, Logan. | (Mcgregor, by his wife, Lori) → (Keli McGregor, per:spouse, Lori) |
| SF13_ENG_020 | "Mike was a first-rate journalist, a valued member of our staff for 25 years and we will miss him," Times Editor Russ Stanton said. " | (Mike, was, A First-Rate Journalist) → (Mike Penner, per:title, first-rate journalist) |
| SF13_ENG_020 | Penner is survived by his brother, John, a copy editor at the Times, and his former wife, Times sportswriter Lisa Dillman. | (Penner, by his brother, John) → (Mike Penner, per:siblings, John) |
| SF13_ENG_024 | She was charged with theft in Beaumont, Texas, for allegedly failing to pay for $10,000 worth of dental work in 2006. | (She, was charged with, Theft) → (Crystal Taylor, per:charges, theft) |
| SF13_ENG_025 | Michigan native Nancy Kissel was convicted of murder and sentenced in Hong Kong's High Court in September 2005. | (Michigan Native Nancy Kissel, convicted of, Murder) → (Nancy Kissel, per:charges, murder) |
| SF13_ENG_026 | Neal returned to New York and concentrated on stage work. | (Neal, returned, New York) → (Patricia Neal, per:statesorprovinces_of_residence, New York) |
| SF13_ENG_026 | In 1953, she married Roald Dahl, the British writer famed for "Charlie and the Chocolate Factory," "James and the Giant Peach" and other tales for children. | (She, married, Roald Dahl) → (Patricia Neal, per:spouse, Roald Dahl) |
| SF13_ENG_026 | Oscar-winning actress Patricia Neal has died of lung cancer at her home on Martha's Vineyard, Massachusetts, on Sunday. | (Patricia Neal, died of, Lung Cancer) → (Patricia Neal, per:cause_of_death, lung cancer) |
| SF13_ENG_027 | Al-Hakim's son, Ammar al-Hakim, has been groomed for months to take his father's place. | (Al-Hakim'S Son, , Ammar al-Hakim) → (Abdul Aziz Al-Hakim, per:children, Ammar al-Hakim) |
| SF13_ENG_027 | Al-Hakim is the head of Supreme Iraqi Islamic Council (SIIC), the largest Shiite party in Iraq. | (Al-Hakim, of, Supreme Iraqi Islamic Council) → (Abdul Aziz Al-Hakim, per:employee_or_member_of, Supreme Iraqi Islamic Council) |
| SF13_ENG_027 | His former Shiite partners have gathered again to form their own group, the Iraqi National Alliance (INA), which includes the influential Supreme Iraqi Islamic Council (SIIC) of Ammar al- Hakim, who succeeded his father Abdul Aziz al-Hakim, who died in a hospital in Iran last month after a long battle with cancer. | (Abdul Aziz Al-Hakim, died in a, Iran) → (Abdul Aziz Al-Hakim, per:country_of_death, Iran) |
| SF13_ENG_028 | "I'd rather have Sally doing this than some stranger, or some hotshot trying to be the next Billy Mays," said the guy who actually is the next Billy Mays, his son Billy Mays III. | (The Next Billy Mays, , his son, Billy Mays Iii) → (Billy Mays, per:children, Billy Mays III) |
| SF13_ENG_029 | Fignon continued cycling during and after a stint in the Army, and drew attention in the early 1980s when he managed to keep up with Hinault during a race in which amateurs and professionals rode together. | (Fignon, during and after a stint in, The Army) → (Laurent Fignon, per:employee_or_member_of, the Army) |
| SF13_ENG_030 | Laurent Patrick Fignon was born in Paris on Aug. 12, 1960. | (Laurent Patrick Fignon, was born in, Paris) → (Laurent Fignon, per:city_of_birth, Paris) |
| SF13_ENG_030 | Anderson became the Tigers' manager in June 1979 and built on a foundation that included Alan Trammell at shortstop, Lou Whitaker at second base, Kirk Gibson in the outfield and Jack Morris on the pitching staff. | (Anderson, became, The Tigers' Manager) → (Sparky Anderson, per:title, manager) |
| SF13_ENG_030 | In addition to his wife, Carol, Anderson is survived by his sons, Lee and Albert; his daughter, Shirlee Englebrecht; and many grandchildren. | (Anderson, is survived by his sons, Lee) → (Sparky Anderson, per:children, Lee) |
| SF13_ENG_031 | Blake Edwards, a writer and director who was hailed as a Hollywood master of screwball farces and rude comedies like "Victor/Victoria" and the "Pink Panther" movies, died Wednesday night in Santa Monica, Calif. He was 88. | (Blake Edwards, , writer) → (Blake Edwards, per:title, writer) |
| SF13_ENG_032 | Hwang, who lives in Seoul under tight police security, has written books and delivered speeches condemning Kim's regime as authoritarian and dictatorial. | (Hwang, , lives in, Seoul) → (Hwang Jang-Yop, per:cities_of_residence, Seoul) |
| SF13_ENG_035 | Kaczynska, who was 67, married Kaczynski in 1978 after meeting him in the northern Polish city of Gdansk, where they were both academics. | (Kaczynska, married, Kaczynski) → (Maria Kaczynska, per:spouse, Kaczynski) |
| SF13_ENG_036 | Upon his release he went into exile in India, where he masterminded the 1973 hijacking of a Royal Nepal Airlines plane known to be carrying hundreds of thousands of dollars in cash to fund his banned Nepali Congress party. | (His Release, he went into exile, India) → (Girija Prasad Koirala, per:countries_of_residence, India) |
| SF13_ENG_036 | Koirala began his political career as a union organiser and was imprisoned for seven years in 1960 after a failed uprising against the monarchy. | (Koirala, career as, A Union Organiser) → (Girija Prasad Koirala, per:title, union organiser) |
| SF13_ENG_036 | Koiarala was born in 1925 in Bihar of India where his father Krishna Prasad Koirala and his family were living in exile. | (Koirala, was born in 1925 in, Bihar) → (Girija Prasad Koirala, per:city_of_birth, Bihar) |
| SF13_ENG_036 | Koirala was born in 1925 in Bihar of India at the time when his father Krishna Prasad Koirala along with his family was exiled by Rana rulers. | (Koirala, born in 1925, Bihar) → (Girija Prasad Koirala, per:city_of_birth, Bihar) |
| SF13_ENG_037 | Chabrol's survivors also include his third wife, Aurore Pajot, who acted as his script supervisor on nearly all of his movies from 1968 on and whom he married in 1981; Pajot's daughter, Cecile Maistre, who was an assistant director on his films and wrote the script with him for "The Girl Cut in Two" (2007). | (Chabrol'S Survivors, wife, Aurore Pajot) → (Claude Chabrol, per:spouse, Aurore Pajot) |
| SF13_ENG_038 | The joint statement said Cunningham was "an inspiring performer and dancer into his 80s, and a visionary choreographer and dedicated teacher throughout his life, he led quietly and by example," the statement said. " | (Cunningham, was, An Inspiring Performer) → (Merce Cunningham, per:title, performer) |
| SF13_ENG_038 | Merce Cunningham, the nonagenarian choreographer, is planning for a world without him. | (Merce Cunningham, , The Nonagenarian Choreographer) → (Merce Cunningham, per:title, choreographer) |
| SF13_ENG_039 | A court on Monday cleared the widower of British reality television star Jade Goody, who died of cancer last year, of rape. | (Goody, , Cancer) → (Jade Goody, per:cause_of_death, cancer) |
| SF13_ENG_040 | Don Hewitt, the CBS newsman who invented the highly popular TV newsmagazine "60 Minutes" and produced it for 36 years, died Wednesday. | (Don Hewitt, , The Cbs Newsman) → (Don Hewitt, per:title, newsman) |
| SF13_ENG_041 | "He was the consummate television newsman," Don Hewitt, a longtime CBS News executive and creator of the long-running "60 Minutes" news program, told Reuters. " | (Don Hewitt, , executive) → (Don Hewitt, per:title, executive) |
| SF13_ENG_041 | Hewitt was already a highly respected TV newsman. | (Hewitt, was, A Highly Respected Tv Newsman) → (Don Hewitt, per:title, TV newsman) |
| SF13_ENG_043 | Donald Shepard Hewitt was born in New York on Dec. 14, 1922, and grew up in the suburb of New Rochelle. | (Donald Shepard Hewitt, born in, New York) → (Don Hewitt, per:stateorprovince_of_birth, New York) |
| SF13_ENG_043 | Eleanor Louise Greenwich was born in Brooklyn on Oct. 23, 1940. | (Eleanor Louise Greenwich, born in, Oct.) → (Ellie Greenwich, per:date_of_birth, 1940-10-23) |
| SF13_ENG_044 | A little more than a year after Dunne died from bladder cancer, the colorful remnants of his estate have been consigned by his family to Stair Galleries in Hudson, N.Y., which will auction them Nov. 20. | (Dunne, died, Bladder Cancer) → (Dominick Dunne, per:cause_of_death, bladder cancer) |
| SF13_ENG_048 | Charles Gwathmey, an architect known for his influential modernist home designs and famous clients like director Steven Spielberg, has died. | (Charles Gwathmey, , architect) → (Charles Gwathmey, per:title, architect) |
| SF13_ENG_049 | Besides his wife, Mandelbrot is survived by two sons, Laurent, of Paris, and Didier, of Newton, Mass., and three grandchildren. | (Besides his wife, Laurent) → (Benoit Mandelbrot, per:children, Laurent) |
| SF13_ENG_049 | For years, he worked for IBM in New York. | (He, worked, Ibm) → (Benoit Mandelbrot, per:employee_or_member_of, IBM) |
| SF13_ENG_049 | After several years spent largely at the Centre National de la Recherche Scientifique in Paris, Mandelbrot was hired by IBM in 1958 to work at the Thomas J. Watson Research Center in Yorktown Heights, N.Y. Although he worked frequently with academic researchers and served as a visiting professor at Harvard and the Massachusetts Institute of Technology, it was not until 1987 that he began to teach at Yale, where he earned tenure in 1999. | (Mandelbrot, hired by, Ibm) → (Benoit Mandelbrot, per:employee_or_member_of, IBM) |
| SF13_ENG_056 | "It's an issue for everybody in the state because peanuts are a big part of our economy," said Don Koehler, executive director of the Georgia Peanut Commission. " | (The Georgia Peanut Commission, director of, Don Koehler) → (Georgia Peanut Commission, org:top_members_employees, Don Koehler) |
| SF13_ENG_060 | "We'll be meeting with scientists, university and science policy officials to explore practical opportunities for exchange and collaboration," Agre, the AAAS president, was quoted as saying. | (Aaas President, , Agre) → (American Association for the Advancement of Science, org:top_members_employees, Peter C. Agre) |
| SF13_ENG_060 | However, Alan Leshner, chief executive officer of the American Association for the Advancement of Science, noted that Nobels are generally given for work that's a decade old or more, and that the U.S. mustn't become complacent. | (The American Association, , Alan Leshner) → (American Association for the Advancement of Science, org:top_members_employees, Alan Leshner) |
| SF13_ENG_060 | Norman Augustine, the former chairman and chief executive of the Lockheed Martin Corporation; Patricia Goldman, former vice chairman of the National Transportation Safety Board; Mary Good, a former president of the American Association for the Advancement of Science; Roger Martin, dean of the Rotman School of Management; Brian O'Neill, a former president of the Insurance Institute for Highway Safety; and Sheila Widnall, a professor at the Massachusetts Institute of Technology. | (The American Association, of, Mary Good) → (American Association for the Advancement of Science, org:top_members_employees, Mary Good) |
| SF13_ENG_062 | "First of all, they never have enough funding," said Andy Kunz, president of the U.S. High Speed Rail Association, a nonprofit that advocates a national high- speed rail network. " | (The U.S. High Speed Rail Association, of, Andy Kunz) → (U.S. High Speed Rail Association, org:top_members_employees, Andy Kunz) |
| SF13_ENG_064 | China's shock at NATO's military campaign in the former Yugoslavia helped prod Beijing into playing a bigger role in U.N. peacekeeping, said Bates Gill, director of the Stockholm International Peace Research Institute and co-author of a recent report on China's peacekeeping activities. | (The Stockholm International Peace Research Institute, , Bates Gill) → (Stockholm International Peace Research Institute, org:top_members_employees, Bates Gill) |
| SF13_ENG_064 | "Non-state actors, for example, a small group of pirates off the coast of Somalia, Al Qaida and Taliban who operate across borders and have more and more sophisticated means of violence, are becoming bigger and bigger challenges to the international system," said Bates Gill, director of the Stockholm International Peace Research Institute. | (The Stockholm International Peace Research Institute, , director of, Bates Gill) → (Stockholm International Peace Research Institute, org:top_members_employees, Bates Gill) |
| SF13_ENG_068 | "It is not surprising that one primarily cosmetic business is trying to throw another under the bus by transferring a tax from rich doctors and their wealthy customers to struggling small businesses," John Overstreet, director of the Indoor Tanning Association, said in a statement Saturday. | (The Indoor Tanning Association, director of, John Overstreet) → (Indoor Tanning Association, org:top_members_employees, John Overstreet) |
| SF13_ENG_076 | The chairman of the Swiss Bankers Association, Patrick Odier, told weekly NZZ am Sonntag that Italy and France have shown interest in deals like ones Switzerland signed this week with Germany and Britain. | (The Swiss Bankers Association, , Patrick Odier) → (Swiss Bankers Association, org:top_members_employees, Patrick Odier) |
| SF13_ENG_076 | The majority of voters in Switzerland, which manages more than 25 percent of the world's foreign-held private wealth, support banking secrecy, according to a survey published last month by the Swiss Bankers Association in Basel. | (The Swiss Bankers Association, in, Basel) → (Swiss Bankers Association, org:city_of_headquarters, Basel) |
| SF13_ENG_078 | "Americans have a right to know the truth – Islam is a religion of intolerance and violence," said Richard Thompson, legal director of the Thomas More Law Center in Ann Arbor. " | (The Thomas More Law Center, in, Ann Arbor) → (Thomas More Law Center, org:city_of_headquarters, Ann Arbor) |
| SF13_ENG_082 | New solutions may be enacted for these orphans, though, said Mary Robinson, CEO of the National Council for Adoption. | (The National Council, of, Mary Robinson) → (National Council for Adoption, org:top_members_employees, Mary Robinson) |
| SF13_ENG_082 | "When you close a country, you end up causing more problems than you prevented," said Chuck Johnson, CEO of the National Council for Adoption. " | (The National Council, CEO of, Chuck Johnson) → (National Council for Adoption, org:top_members_employees, Chuck Johnson) |
| SF13_ENG_083 | "This is definitely a Goldilocks problem," said Jason Grumet, president of the Bipartisan Policy Center and an energy adviser to the Obama campaign last year. | (The Bipartisan Policy Center, , Jason Grumet) → (Bipartisan Policy Center, org:top_members_employees, Jason Grumet) |
| SF13_ENG_085 | "Banks are in strong need for the capital markets (to raise funds)," Li Fuan, a director at the China Banking Regulatory Commission, was quoted as saying at a forum over the weekend. | (The China Banking Regulatory Commission, a director, Li Fuan) → (China Banking Regulatory Commission, org:top_members_employees, Li Fuan) |
| SF13_ENG_089 | RIA Novosti and Interfax cite Anatoly Isaikin, head of Rosoboronexport, as saying Thursday "nothing is blocking the continuation of military-technical cooperation" with Iran. | (Rosoboronexport, , head, of, Anatoly Isaikin) → (Rosoboronexport, org:top_members_employees, Anatoly Isaikin) |
| SF13_ENG_089 | Rosoboronexport is the only company in Russia that is allowed to export arms, dual-use products and military-related services. | (Rosoboronexport, is the only company in, Russia) → (Rosoboronexport, org:country_of_headquarters, Russia) |
| SF13_ENG_091 | With his wife, Cornelie, Middelhoff invested money in 2000 and 2001 with Esch in funds that were formed to buy five properties from KarstadtQuelle, as Arcandor was then known, and leased back to the department store chain before Middelhoff joined the company, according to Middelhoff's spokesman. | (Arcandor, , Karstadtquelle) → (Arcandor, org:alternate_names, KarstadtQuelle) |
| SF13_ENG_091 | Arcandor's mail-order subsidiary Quelle is in worse shape however, Klaus-Hubert Goerg told a press conference in the western city of Essen, where Arcandor is based. | (Arcandor, , Essen) → (Arcandor, org:city_of_headquarters, Essen) |

Figure 5: Mapped facts: MAMA-GPT-2$_{XL}$ on TAC KBP.

| ID | Sentence | BERT$_{\text{LARGE}}$ | GPT-2$_{\text{XL}}$ |
|---|---|---|---|
| | | | **Candidate facts to mapped facts** |
| SF13_ENG_005 | The men , who live in the Chicago area , were identified as David Coleman Headley , 49 , a US citizen , and Tahawwur Hussain Rana , 48 , Rana , a native of Pakistan and citizen of Canada . | (Rana, , a native of, Canada) → (Tahawwur Hussain Rana, per:origin, Canada) | - |
| SF13_ENG_006 | Khan died at his music centre near San Francisco in the United States after a prolonged kidney ailment , the news agency said . | (Khan, died at his, The United States) → (Ali Akbar Khan, per:country_of_death, United States) | - |
| SF13_ENG_006 | By his early 20s he was music director of All-India Radio in Lucknow, broadcasting as a solo artist and composing for the radio's orchestra. | - | (He, of, All-India Radio) → (Ali Akbar Khan, per:employee_or_member_of, All-India Radio) |
| SF13_ENG_006 | Defying his father , Khan moved to Bombay and began writing scores for films , including Chetan Anand ' s " Aandhiyan " ( 1952 ) , Satyajit Ray ' s " Devi " ( 1960 ) and Tapan Sinha ' s " Hungry Stones " ( 1960 ) . | (Khan, moved, Bombay) → (Ali Akbar Khan, per:cities_of_residence, Bombay) | - |
| SF13_ENG_007 | In 1966 he graduated from the faculty of theology at Al-Azhar, the 10th century university that has trained the majority of Sunni Muslim clerics from Africa and Asia. | - | (He, graduated from, Al-Azhar) → (Mohammed Sayed Tantawi, per:schools_attended, Al-Azhar) |
| SF13_ENG_009 | He became AIBA president in 2006 by defeating longtime incumbent Anwar Chowdhry of Pakistan, who was later barred for alleged financial corruption. | - | (Longtime Incumbent Anwar Chowdhry, by, Aiba President) → (Anwar Chowdhry, per:employee_or_member_of, AIBA) |
| SF13_ENG_011 | Marshall could get a lengthy prison term if convicted of grand larceny. | - | (Marshall, of, Grand Larceny) → (Anthony Marshall, per:charges, larceny) |
| SF13_ENG_012 | Gadahn , also known as Azzam al - Amriki , was dressed in white robes and wearing a white turban as he called for attacks on what he described as " high - value targets ." | (Gadahn, known as, Azzam Al-Amriki) → (Adam Gadahn, per:alternate_names, Azzam al-Amriki) | - |
| SF13_ENG_012 | Gadahn grew up on a farm in California and converted to Islam before moving to Pakistan in 1998 and reportedly attending an al - Qaida training camp . | (Gadahn, to, Pakistan) → (Adam Gadahn, per:countries_of_residence, Pakistan) | - |
| SF13_ENG_012 | He was born in Oregon on Sept. 1, 1978 to a Jewish family. | - | (He, born in, Sept.) → (Adam Gadahn, per:date_of_birth, 1978-09-01) |
| SF13_ENG_015 | Besides her sister , Philip , survivors include a son , Graham ; three daughters , Sidney , Gillian and Alexia ; and three grandchildren . | (Her Sister, son ,, Graham) → (Lucille Clifton, per:children, Graham) | - |
| SF13_ENG_015 | In addition to her daughter , of Columbia , other survivors include two daughters , Sidney Clifton of Los Angeles and Gillian Clifton - Monell of Fort Washington , Md . , ; a son , Graham Clifton of Wichita ; a sister ; and three grandchildren . | (Her Daughter, ,, Sidney Clifton) → (Lucille Clifton, per:children, Sidney Clifton) | - |
| SF13_ENG_017 | Haig , 85 , died early Saturday in a Baltimore , Maryland , hospital . | (Haig, died early, A Baltimore, Maryland, Hospital) → (Alexander Haig, per:stateorprovince_of_death, Maryland) | - |
| SF13_ENG_019 | McGregor was born in Iowa , but grew up in Colorado where his father , Brian , was a coaching legend at Arvada West . | (Mcgregor, but grew up, Brian) → (Keli McGregor, per:parents, Brian) | - |
| SF13_ENG_025 | Kissel was born in Adrian, Michigan, but her family had also lived in Minneapolis. | - | (Kissel, in, Minneapolis) → (Nancy Kissel, per:cities_of_residence, Minneapolis) |
| SF13_ENG_025 | Kissel, 46, of Adrian, Michigan, was convicted of drugging then bashing her husband Robert to death in a luxury Hong Kong apartment and sentenced to life in prison in September 2005. | - | (Kissel, of, Michigan) → (Nancy Kissel, per:stateorprovinces_of_residence, Michigan) |
| SF13_ENG_026 | Neal had lung cancer and died at her home in Edgartown , Massachusetts on Martha ' s Vineyard , said longtime friend Bud Albers of Knoxville . | (Neal, died at her home in, Edgartown) → (Patricia Neal, per:city_of_death, Edgartown) | - |
| SF13_ENG_026 | She grew up in Knoxville, Tenn., where Neal was turned onto acting at 10 after seeing an impassioned speech about "demon rum" at the local temperance union speaking contest. | - | (She, in, Tenn.) → (Patricia Neal, per:stateorprovinces_of_residence, Tennessee) |
| SF13_ENG_027 | Al-Hakim, who died Wednesday of lung cancer in Tehran, was a symbol for many of the re-emergence of Iraq's Shiite political majority after decades of oppression under Saddam Hussein's Sunni-led regime. | - | (Al-Hakim, who died, Tehran) → (Abdul Aziz Al-Hakim, per:city_of_death, Tehran) |
| SF13_ENG_028 | TAMPA Heart disease , exacerbated by cocaine use , killed celebrated TV pitchman Billy Mays , according to the Hillsborough County medical examiner ' s final autopsy report released Friday . | (Celebrated Tv Pitchman Billy Mays, killed, Tampa) → (Billy Mays, per:cause_of_death, TAMPA Heart disease) | - |
| SF13_ENG_030 | Family spokesman Dan Ewald said Anderson died from complications from dementia . | (Anderson, died from, Dementia) → (Sparky Anderson, per:cause_of_death, dementia) | - |
| SF13_ENG_031 | He served in the Coast Guard during World War II and had a rare leading role in the low - budget thriller " Strangler of the Swamp " ( 1946 ) . | (He, in, The Coast Guard) → (Blake Edwards, per:employee_or_member_of, Coast Guard) | - |
| SF13_ENG_031 | His agent Lou Pitt confirmed he had died , while the " Entertainment Tonight " news show and website reported that he died in Santa Monica on Wednesday night from complications of pneumonia , with Andrews at his side . | (He, died, Complications) → (Blake Edwards, per:cause_of_death, complications) | - |
| SF13_ENG_033 | In a career that spanned seven decades, Ginzburg authored several groundbreaking studies in various fields – such as quantum theory, astrophysics, radio-astronomy and diffusion of cosmic radiation in the Earth's atmosphere – that were of "Nobel Prize caliber," said Gennady Mesyats, the director of the Lebedev Physics Institute in Moscow, where Ginzburg worked. | - | (Ginzburg, in, The Lebedev Physics Institute) → (Vitaly Ginzburg, per:employee_or_member_of, Lebedev Physics Institute) |
| SF13_ENG_035 | Kaczynska , who died aged 59 , married Kaczynski in 1978 . | (Kaczynska, married, Kaczynski) → (Maria Kaczynska, per:spouse, Kaczynski) | - |
| SF13_ENG_037 | Prime Minister Francois Fillon called Chabrol a " great director , producer and screenwriter ( who ) was one of the grand figures of the ' Nouvelle vague , ' which revolutionized the style and techniques of cinema by looking at real experience , true life , that which is indiscreet and subtle . " | (Afp Chabrol, was, director) → (Claude Chabrol, per:title, film director); (Chabrol, director, Producer) → (Claude Chabrol, per:title, producer) | - |
| SF13_ENG_038 | Cunningham's survivors include a brother, Jack Cunningham of Centralia. | - | (Cunningham'S Survivors, brother, Jack Cunningham) → (Merce Cunningham, per:siblings, Jack Cunningham) |
| SF13_ENG_038 | Mercier Philip Cunningham was born in Centralia, Wash., on April 16, 1919. | - | (Mercier Philip Cunningham, was born in, April) → (Merce Cunningham, per:date_of_birth, 1919-04-16) |
| SF13_ENG_041 | A year after enrolling in New York University on a track scholarship , Hewitt dropped out . | (Hewitt, ,, New York University) → (Don Hewitt, per:schools_attended, New York University) | - |
| SF13_ENG_041 | By the time Hewitt was 6, his family was living in New Rochelle, N.Y. | - | (Hewitt, in New Rochelle) → (Don Hewitt, per:cities_of_residence, New Rochelle) |
| SF13_ENG_043 | Greenwich also worked as an arranger and singer , a role that saw her working with artists including Frank Sinatra and Ella Fitzgerald . | (Greenwich, worked as an, Singer) → (Ellie Greenwich, per:title, singer) | - |
| SF13_ENG_044 | Dunne was born in 1925 in Hartford, Connecticut, to a wealthy Roman Catholic family and grew up in some of the same social circles as the Kennedys. | - | (Dunne, was born in 1925 in, Connecticut) → (Dominick Dunne, per:stateorprovince_of_birth, Connecticut) |
| SF13_ENG_044 | In 1957 , Dunne moved to Los Angeles to work on the CBS showcase " Playhouse 90 ." | (Dunne, moved to, Los Angeles) → (Dominick Dunne, per:cities_of_residence, Los Angeles) | - |
| SF13_ENG_044 | Dunne found his greatest prominence as a celebrity journalist while covering the 1995 murder trial of football star and actor O.J. Simpson, who had been accused of killing his ex-wife Nicole Brown Simpson and her friend Ronald Goldman. | - | (Dunne, as, A Celebrity Journalist) → (Dominick Dunne, per:title, celebrity journalist) |
| SF13_ENG_045 | Goldstein died in New York on Aug . 28 . | (Goldstein, died, Aug.) → (Adam Goldstein, per:date_of_death, XXXX-08-28) | - |
| SF13_ENG_046 | After acting in movies and television , Meredith receded into a quiet life in Santa Fe , writing , painting , golfing and acting in a stage production of " The Odd Couple . " | (Meredith, in, Santa Fe) → (Don Meredith, per:cities_of_residence, Santa Fe) | - |
| SF13_ENG_046 | In high school and at Southern Methodist University , where , already known as Dandy Don ( a nickname bestowed on him by his brother ) , Meredith became an all - American . | (Dandy Don, where ,, Southern Methodist University) → (Don Meredith, per:schools_attended, Southern Methodist University) | - |
| SF13_ENG_046 | He spent much of his life backing away from the nickname Dandy Don , particularly during his secluded later decades in New Mexico . | (He, the nickname, Dandy Don) → (Don Meredith, per:alternate_names, Dandy Don) | - |
| SF13_ENG_048 | The group, known variously as the Five or the Whites (for the color of most of their buildings) or the New York School, consisted of Gwathmey, Michael Graves, Eisenman, John Hejduk and Richard Meier. | - | (Gwathmey, of, The New York School) → (Charles Gwathmey, per:employee_or_member_of, New York School) |
| SF13_ENG_048 | Meier , who said he had known Gwathmey for 50 years , has particularly fond memories from the time when Gwathmey was first courting his second wife , Bette - Ann Damson , and they all picked corn for dinner in a field adjacent to a barn Meier was renting on the East End of Long Island . | (Gwathmey, his second wife ,, Bette-Ann Damson) → (Charles Gwathmey, per:spouse, Bette-Ann Damson) | - |
| SF13_ENG_057 | By 2013, automakers will have dozens of plug-in electric hybrid vehicles and fully electric vehicles, said Jason Forcier, a vice president at battery maker A123 Systems Inc. | - | (A123 Systems Inc., a vice president, Jason Forcier) → (A123 Systems Inc., org:top_members_employees, Jason Forcier) |
| SF13_ENG_058 | Access Industries , a privately held company founded in 1986 by Len Blavatnik , has a diverse portfolio of investments in industry , real estate , media and telecommunications . | (Access Industries, founded in 1986 by, Len Blavatnik) → (Access Industries, org:founded_by, Len Blavatnik) | - |
| SF13_ENG_069 | "The indications are positive," said Vincent Cogliano, director of the Monographs program at IARC, which decides on carcinogen classifications. | - | (Iarc, , director of the, Vincent Cogliano) → (International Agency for Research on Cancer, org:top_members_employees, Vincent Cogliano) |
| SF13_ENG_082 | "There hasn't been a concerted push to open doors for Muslim orphans because the expectation would be that those efforts would fall flat," said Chuck Johnson, chief executive of the National Council for Adoption, a policy group in Alexandria, Va. | - | (Adoption, in, Alexandria) → (National Council for Adoption, org:city_of_headquarters, Alexandria) |
| SF13_ENG_084 | " The scenario itself is secret , " said Eileen McMenamin , vice president of communications for the Bipartisan Policy Center ( BPC ) , which is hosting the event dubbed " Cyber ShockWave . " | (The Bipartisan Policy Center, vice president of, Eileen Mcmenamin) → (Bipartisan Policy Center, org:top_members_employees, Eileen McMenamin) | - |
| SF13_ENG_088 | Bank Julius Baer Co., based in Basel, Switzerland, sued because WikiLeaks posted accountholder information from its Cayman outpost amid allegations of money laundering and tax evasion. | - | (Bank Julius Baer Co., based, Basel) → (Bank Julius Baer, org:city_of_headquarters, Basel) |
| SF13_ENG_094 | The other manufacturers are two US companies, L-3 Communications and Rapiscan Systems, a unit of OSI Systems, and British rival Smiths Detection. | - | (Rapiscan, ,, Two Us Companies) → (Rapiscan Systems, org:country_of_headquarters, US) |
| SF13_ENG_096 | Freedom was founded in the 1930s by R.C. Hoiles and is still majority owned by the Hoiles family. | - | (Freedom, was founded in the, R.C. Hoiles) → (Freedom Communications, org:founded_by, R.C. Hoiles) |
| SF13_ENG_098 | Shares of Millipore have more than doubled since Martin D . Madaus took over as chief executive in January 2005 . | (Millipore, have, Martin D.) → (Millipore Corp., org:top_members_employees, Martin D. Madaus) | - |

Figure 6: Mapped facts: MAMA-BERT$_{\text{LARGE}}$ vs. MAMA-GPT-2$_{\text{XL}}$ on TAC KBP.

| ID | Sentence | Candidate facts to unmapped facts |
|---|---|---|
| SF13_ENG_001 | Bashardost was born in 1965 in the southern Ghanzi province and his family migrated to Iran and then to Pakistan after successive coup and factional fighting in Afghanistan . | (Bashardost, in, The Southern Ghanzi Province) → (Ramazan Bashardost, per:stateorprovince_of_birth, The Southern Ghanzi Province) |
| SF13_ENG_012 | Gadahn , also known as Azzam the American , was born in 1978 . | (Gadahn, ,, Azzam) → (Adam Gadahn, per:alternate_names, Azzam) |
| SF13_ENG_015 | Besides her sister , Philip , survivors include a son , Graham ; three daughters , Sidney , Gillian and Alexia ; and three grandchildren . | (Her Sister, ,, Philip) → (Lucille Clifton, per:siblings, Philip) |
| SF13_ENG_015 | The poet laureate of Maryland from 1979 to 1985 , Clifton was a writer in residence at Coppin State College , now Coppin State University , a historically black college in Baltimore . | (Clifton, in, Coppin State College) → (Lucille Clifton, per:school_attended, Coppin State College) |
| SF13_ENG_015 | Her father , Samuel , was a steelworker ; her mother , Thelma , worked in a laundry . | (Her Father, ,, Samuel) → (Lucille Clifton, per:parents, Samuel) |
| SF13_ENG_015 | Lucille Clifton , a distinguished American poet whose work trained lenses wide and narrow on the experience of being black and female in the 20th century , exploring vast subjects like the indignities of history and intimate ones like the indignities of the body , died on Saturday in Baltimore . | (Lucille Clifton, ,, poet) → (Lucille Clifton, per:title, poet) |
| SF13_ENG_015 | The precise cause of death had not been determined , her sister , Elaine Philip , told The Associated Press on Sunday . | (Her Sister, ,, Elaine Philip) → (Lucille Clifton, per:siblings, Elaine Philip) |
| SF13_ENG_037 | CLAUDE CHABROL , PIONEER FRENCH FILMMAKER , DIES AT 80 Joseph Berger contributed reporting from New York , and Maia de la Baume contributed from Paris . | (Claude Chabrol, ,, Pioneer French Filmmaker) → (Claude Chabrol, per:title, Pioneer French Filmmaker) |
| SF13_ENG_037 | Chabrol ' s survivors also include his third wife , Aurore Pajot , who acted as his script supervisor on nearly all of his movies from 1968 on and whom he married in 1981 ; and Pajot ' s daughter , Cecile Maistre , who was an assistant director on his films and wrote the script with him for " The Girl Cut in Two " ( 2007 ) . | (His Third Wife, ,, Aurore Pajot) → (Claude Chabrol, per:spouse, Aurore Pajot) |
| SF13_ENG_044 | Dominick Dunne , a novelist and journalist who chronicled true - crime tales of the rich and infamous , including O . J . Simpson and Claus von Bulow , and in turn became a celebrity in his own right , died of bladder cancer Aug . 26 at his home in New York City . | (Dominick Dunne, ,, A Novelist) → (Dominick Dunne, per:title, A Novelist) |
| SF13_ENG_044 | Dunne and his wife , Ellen Griffin Dunne , known as Lenny , were married in 1954 . | (His Wife, ,, Ellen Griffin Dunne) → (Dominick Dunne, per:spouse, Ellen Griffin Dunne) |
| SF13_ENG_044 | Dunne was born in 1925 in Hartford , Connecticut , to a wealthy Roman Catholic family and grew up in some of the same social circles as the Kennedys . | (Dunne, ,, Connecticut) → (Dominick Dunne, per:stateorprovince_of_birth, Connecticut) |
| SF13_ENG_049 | A professor emeritus at Yale University , Mandelbrot was born in Poland but as a child moved with his family to France where he was educated . | (Mandelbrot, ,, Yale University) → (Benoit Mandelbrot, per:employee_or_member_of, Yale University) |
| SF13_ENG_058 | Access Industries , a privately held company founded in 1986 by Len Blavatnik , has a diverse portfolio of investments in industry , real estate , media and telecommunications . | (Access Industries, ,, A Privately Held Company) → (Access Industries, properties, A Privately Held Company) |
| SF13_ENG_079 | Water and its links to development , peace and conflict were key words in the annual sessions , Anders Berntell , executive director of Stockholm International Water Institute ( SIWI ) , said in his opening address . | (Stockholm International, ,, Anders Berntell) → (Stockholm International Water Institute, per:top_members_or_employees, Anders Berntell) |

Figure 7: Unmapped facts: MAMA-BERT$_{\text{LARGE}}$ on TAC KBP.

| ID | Sentence | Candidate facts to unmapped facts |
|---|---|---|
| SF13_ENG_009 | He took office in 2006 by defeating longtime incumbent Anwar Chowdhry from Pakistan, who was later barred for alleged financial corruption. | (Longtime Incumbent Anwar Chowdhry, from, Pakistan) → (Anwar Chowdhry, per:countries_of_residence, Pakistan) |
| SF13_ENG_012 | The message also featured several appearances of Adam Gadahn, also known as Azzam al-Amriki, an American who grew up in Riverside County and converted to Islam and joined al-Qaida. | (Adam Gadahn, also, Azzam Al-Amriki) → (Adam Gadahn, per:alternate_names, Azzam Al-Amriki); (Adam Gadahn, ,, Azzam Al-Amriki) → (Adam Gadahn, ,, Azzam Al-Amriki) |
| SF13_ENG_014 | Mohammed Oudeh, a former math teacher who became the mastermind of the deadly attack on Israeli athletes at the 1972 Munich Olympics, died Friday in Damascus. | (Mohammed Oudeh, ,, teacher) → (Mohammed Oudeh, per:title, teacher) |
| SF13_ENG_014 | He settled in the West Bank town of Ramallah, but in 1999, after a trip to Jordan, he was barred by Israel from returning. | (He, in, The West Bank Town) → (Mohammed Oudeh, per:cities_of_residence, The West Bank Town) |
| SF13_ENG_014 | Mohammed Oudeh, a former math teacher who became the mastermind of the deadly attack on Israeli athletes at the 1972 Munich Olympics, died Friday in Damascus. | (Mohammed Oudeh, ,, A Former Math Teacher) → (Mohammed Oudeh, per:title, A Former Math Teacher) |
| SF13_ENG_014 | In later years, as a graying member of the Palestinian old guard, Oudeh, most commonly known by his guerrilla name, Abu Daoud, showed no remorse for the botched hostage taking and killings of 11 members of the Israeli Olympic team that shook the world. | (Oudeh, ,, Abu Daoud) → (Mohammed Oudeh, per:alternate_names, Abu Daoud) |
| SF13_ENG_015 | After her husband died in 1984, Clifton taught at the University of California at Santa Cruz before returning to Maryland in 1989. | (Clifton, of, Santa Cruz) → (Lucille Clifton, per:cities_of_residence, Santa Cruz) |
| SF13_ENG_015 | Lucille Clifton, a National Book Award-winning poet and Pulitzer finalist, has died. | (Lucille Clifton, ,, poet) → (Lucille Clifton, per:title, poet) |
| SF13_ENG_015 | When she was a girl, Lucille Clifton sat on her mother's lap and listened to her recite poetry. | (Lucille Clifton, ,, A Girl) → (Lucille Clifton, gender, A Girl) |
| SF13_ENG_015 | In 1967, they moved to Baltimore, and Clifton worked in Washington for the old U.S. Office of Education. | (They, to, Baltimore) → (Lucille Clifton, per:cities_of_residence, Baltimore) |
| SF13_ENG_015 | Clifton's sister, Elaine Philip, said the former poet laureate of Maryland passed away Saturday morning at Johns Hopkins University Hospital in Baltimore. | (Clifton'S Sister, ,, Elaine Philip) → (Lucille Clifton, per:siblings, Elaine Philip) |
| SF13_ENG_018 | Harvey Pekar, the writer of comics whose autobiographical comic book series "American Splendor" chronicled his life as a filing clerk, record collector, freelance jazz critic and one of life's all-around misfits, was found dead July 12 at his home near Cleveland. | (Harvey Pekar, ,, writer) → (Harvey Pekar, per:title, writer) |
| SF13_ENG_018 | Pekar attended what became Case Western Reserve University, served in the Navy in the late 1950s and worked a series of menial jobs before taking a 30-year job as a filing clerk at a VA hospital in Cleveland. | (Pekar, what became, Case Western Reserve University) → (Harvey Pekar, per:schools_attended, Case Western Reserve University) |
| SF13_ENG_025 | Kissel, 46, of Adrian, Michigan, was convicted of drugging then bashing her husband Robert to death in a luxury Hong Kong apartment and sentenced to life in prison in September 2005. | (Kissel, ,, Adrian) → (Nancy Kissel, per:cities_of_residence, Adrian); (Kissel, ,, Michigan) → (Nancy Kissel, per:stateorprovinces_of_residence, Michigan) |
| SF13_ENG_038 | Mercier Philip Cunningham was born in Centralia, Wash., on April 16, 1919. | (Mercier Philip Cunningham, was born in, Centralia) → (Merce Cunningham, was born in, Centralia); (Mercier Philip Cunningham, in, Wash.) → (Merce Cunningham, per:stateorprovince_of_birth, Wash.); (Mercier Philip Cunningham, in, April) → (Merce Cunningham, per:date_of_birth, 1919-04-16) |
| SF13_ENG_038 | Cunningham's survivors include a brother, Jack Cunningham of Centralia. | (Cunningham'S Survivors, ,, Jack Cunningham) → (Merce Cunningham, per:siblings, Jack Cunningham) |
| SF13_ENG_041 | By the time Hewitt was 6, his family was living in New Rochelle, N.Y. | (Hewitt, in, New Rochelle) → (Don Hewitt, per:cities_of_residence, New Rochelle) |
| SF13_ENG_041 | He is survived by his third wife, former television news correspondent Marilyn Berger; his sons, Steven and Jeffrey; his daughter, Lisa Cassara; his stepdaughter, Jilian Childers Hewitt, whom Hewitt adopted; and three grandchildren. | (Hewitt, ,, Jilian Childers Hewitt) → (Don Hewitt, per:children, Jilian Childers Hewitt) |
| SF13_ENG_044 | Dunne was born in 1925 in Hartford, Connecticut, to a wealthy Roman Catholic family and grew up in some of the same social circles as the Kennedys. | (Dunne, was born in 1925 in, Connecticut) → (Dominick Dunne, per:stateorprovince_of_birth, Connecticut) |
| SF13_ENG_044 | Dunne was part of a famous family that also included his brother, novelist and screenwriter John Gregory Dunne; his brother's wife, author Joan Didion; and his son, Griffin. | (His Brother, ,, Novelist) → (His brother, per:title, Novelist) |
| SF13_ENG_044 | Dunne and his wife, Ellen Griffin Dunne, known as Lenny, were married in 1954. | (Dunne, and, Ellen Griffin Dunne) → (Dominick Dunne, per:spouse, Ellen Griffin Dunne) |
| SF13_ENG_047 | The women were accompanied by Reyna Luisa Tamayo, the mother of political prisoner Orlando Zapata, who died at age 42 in a hunger strike February 23 to protest prison conditions. | (Political Prisoner Orlando Zapata, ,, Reyna Luisa Tamayo) → (Orlando Zapata, per:children, Reyna Luisa Tamayo) |
| SF13_ENG_061 | InterContinental Hotels Group, owners of Holiday Inn Worldwide, has been ordered by a court to pay $25 million in damages to a franchisee for fraud. | (Intercontinental Hotels Group, ,, Holiday Inn Worldwide) → (InterContinental Hotels Group, org:parents, Holiday Inn Worldwide) |
| SF13_ENG_079 | Water and its links to development, peace and conflict were key words in the annual sessions, Anders Berntell, executive director of Stockholm International Water Institute (SIWI), said in his opening address. | (Stockholm International, director, Anders Berntell) → (Stockholm International Water Institute, per:top_members_or_employees, Anders Berntell); (Stockholm International, , executive director, Anders Berntell) → (Stockholm International Water Institute, per:top_members_or_employees, Anders Berntell) |
| SF13_ENG_090 | ECO member countries include Afghanistan, Azerbaijan, Iran, Kazakhstan, Kyrgyzstan, Pakistan, Tajikistan, Turkey, Turkmenistan and Uzbekistan. | (Eco Member Countries, ,, Azerbaijan) → (Economic Cooperation Organization, org:members, Azerbaijan) |
| SF13_ENG_096 | A phone message left Sunday at Freedom's Irvine, California, headquarters was not immediately returned. | (Freedom'S Irvine, ,, California) → (Freedom Communications, org:stateorprovince_of_headquarters, California) |
| SF13_ENG_097 | STX Finland is part of the international STX Europe Group, with shipyards in Brazil, Norway, France, Romania and Vietnam. | (Stx Finland, is part of, The International Stx Europe Group) → (STX Finland, org:subsidaries, The International Stx Europe Group) |

Figure 8: Unmapped facts: MAMA-GPT-2$_{\text{XL}}$ on TAC KBP.

| ID | Sentence | BERT$_{LARGE}$ | GPT-2$_{XL}$ |
|---|---|---|---|
| | | **Candidate facts to mapped facts** | |
| SF13_ENG_001 | Bashardost was born in 1965 in the southern Ghanzi province and his family migrated to Iran and then to Pakistan after successive coup and factional fighting in Afghanistan . | (Bashardost, in, The Southern Ghanzi Province) → (Ramazan Bashardost, per:stateorprovince_of_birth, The Southern Ghanzi Province) | - |
| SF13_ENG_009 | He took office in 2006 by defeating longtime incumbent Anwar Chowdhry from Pakistan, who was later barred for alleged financial corruption. | (Longtime Incumbent Anwar Chowdhry, from, Pakistan) → (Anwar Chowdhry, per:origin, Pakistan) | (Longtime Incumbent Anwar Chowdhry, from, Pakistan) → (Anwar Chowdhry, per:countries_of_residence, Pakistan) |
| SF13_ENG_012 | Gadahn, also known as Azzam the American , was born in 1978 . | (Gadahn, ,, Azzam) → (Adam Gadahn, per:alternate_names, Azzam) | (Gadahn, also known as, Azzam) → (Adam Gadahn, per:alternate_names, Azzam) |
| SF13_ENG_012 | The message also featured several appearances of Adam Gadahn, also known as Azzam al-Amriki, an American who grew up in Riverside County and converted to Islam and joined al-Qaida. | (Adam Gadahn, also known as, Azzam Al-Amriki) → (Adam Gadahn, per:alternate_names, Azzam Al-Amriki) | (Adam Gadahn, also, Azzam Al-Amriki) → (Adam Gadahn, per:alternate_names, Azzam Al-Amriki); (Adam Gadahn, ,, Azzam Al-Amriki) → (Adam Gadahn, per:alternate_names, Azzam Al-Amriki) |
| SF13_ENG_014 | Mohammed Oudeh, a former math teacher who became the mastermind of the deadly attack on Israeli athletes at the 1972 Munich Olympics, died Friday in Damascus. | (Mohammed Oudeh, ,, teacher) → (Mohammed Oudeh, per:title, teacher) | - |
| SF13_ENG_014 | He settled in the West Bank town of Ramallah, but in 1999, after a trip to Jordan, he was barred by Israel from returning. | (He, in, The West Bank Town) → (Mohammed Oudeh, per:cities_of_residence, The West Bank Town) | (He, settled, in, The West Bank Town) → (Mohammed Oudeh, per:cities_of_residence, The West Bank Town) |
| SF13_ENG_014 | Mohammed Oudeh, a former math teacher who became the mastermind of the deadly attack on Israeli athletes at the 1972 Munich Olympics, died Friday in Damascus. | (Mohammed Oudeh, ,, A Former Math Teacher) → (Mohammed Oudeh, per:title, A Former Math Teacher) | (Mohammed Oudeh, ,, teacher) → (Mohammed Oudeh, per:title, teacher) |
| SF13_ENG_014 | In later years, as a graying member of the Palestinian old guard, Oudeh, most commonly known by his guerrilla name, Abu Daoud, showed no remorse for the botched hostage taking and killings of 11 members of the Israeli Olympic team that shook the world. | (Oudeh, ,, Abu Daoud) → (Mohammed Oudeh, per:alternate_names, Abu Daoud) | - |
| SF13_ENG_015 | The precise cause of death had not been determined , her sister , Elaine Philip , told The Associated Press on Sunday . | - | (Her Sister, ,, Elaine Philip) → (Lucille Clifton, per:siblings, Elaine Philip) |
| SF13_ENG_015 | After her husband died in 1984, Clifton taught at the University of California at Santa Cruz before returning to Maryland in 1989. | (Clifton, of, Santa Cruz) → (Lucille Clifton, per:cities_of_residence, Santa Cruz) | (Clifton, taught at, the University) → (Lucille Clifton, per:employee_or_member_of, the University of California) |
| SF13_ENG_015 | Lucille Clifton, a National Book Award-winning poet and Pulitzer finalist, has died. | (Lucille Clifton, ,, poet) → (Lucille Clifton, per:title, poet) | (Lucille Clifton, ,, poet) → (Lucille Clifton, per:title, poet) |
| SF13_ENG_015 | When she was a girl, Lucille Clifton sat on her mother's lap and listened to her recite poetry. | - | (Lucille Clifton, ,, A Girl) → (Lucille Clifton, gender, A Girl) |
| SF13_ENG_015 | In 1967, they moved to Baltimore, and Clifton worked in Washington for the old U.S. Office of Education. | (Clifton, worked in, old U.S. Office) → (Lucille Clifton, per:employee_or_member_of, U.S. Office of Education) | (They, to, Baltimore) → (Lucille Clifton, per:cities_of_residence, Baltimore) |
| SF13_ENG_015 | Clifton's sister, Elaine Philip, said the former poet laureate of Maryland passed away Saturday morning at Johns Hopkins University Hospital in Baltimore. | - | (Clifton'S Sister, ,, Elaine Philip) → (Lucille Clifton, per:siblings, Elaine Philip) |
| SF13_ENG_018 | Harvey Pekar, the writer of comics whose autobiographical comic book series "American Splendor" chronicled his life as a filing clerk, record collector, freelance jazz critic and one of life's all-around misfits, was found dead July 12 at his home near Cleveland. | - | (Harvey Pekar, ,, writer) → (Harvey Pekar, per:title, writer) |
| SF13_ENG_018 | Pekar attended what became Case Western Reserve University, served in the Navy in the late 1950s and worked a series of menial jobs before taking a 30-year jobas a filing clerk at a VA hospital in Cleveland. | - | (Pekar, what became, Case Western Reserve University) → (Harvey Pekar, per:schools_attended, Case Western Reserve University) |
| SF13_ENG_025 | Kissel, 46, of Adrian, Michigan, was convicted of drugging then bashing her husband Robert to death in a luxury Hong Kong apartment and sentenced to life in prison in September 2005. | - | (Kissel, ,, Adrian) → (Nancy Kissel, per:cities_of_residence, Adrian); (Kissel, ,, Michigan) → (Nancy Kissel, per:stateprovinces_of_residence, Michigan) |
| SF13_ENG_037 | CLAUDE CHABROL , PIONEER FRENCH FILMMAKER , DIES AT 80 Joseph Berger contributed reporting from New York , and Maia de la Baume contributed from Paris . | (Claude Chabrol, ,, Pioneer French Filmmaker) → (Claude Chabrol, per:title, Pioneer French Filmmaker) | - |
| SF13_ENG_037 | Chabrol ' s survivors also include his third wife , Aurore Pajot , who acted as his script supervisor on nearly all of his movies from 1968 on and whom he married in 1981 ; and Pajot ' s daughter , Cecile Maistre , who was an assistant director on his films and wrote the script with him for " The Girl Cut in Two " ( 2007 ) . | (His Third Wife, ,, Aurore Pajot) → (Claude Chabrol, per:spouse, Aurore Pajot) | (Chabrol's survivors, third wife, Aurore Pajot) → (Claude Chabrol, per:spouse, Aurore Pajot) |
| SF13_ENG_038 | Mercier Philip Cunningham was born in Centralia, Wash., on April 16, 1919. | - | (Mercier Philip Cunningham, was born in, Centralia) → (Merce Cunningham, was born in, Centralia); (Mercier Philip Cunningham, in, Wash.) → (Merce Cunningham, per:stateorprovince_of_birth, Wash.); (Mercier Philip Cunningham, in, April) → (Merce Cunningham, per:date_of_birth, 1919-04-16) |
| SF13_ENG_038 | Cunningham's survivors include a brother, Jack Cunningham of Centralia. | - | (Cunningham'S Survivors, ,, Jack Cunningham) → (Merce Cunningham, per:siblings, Jack Cunningham) |
| SF13_ENG_041 | By the time Hewitt was 6, his family was living in New Rochelle, N.Y. | - | (Hewitt, in, New Rochelle) → (Don Hewitt, per:cities_of_residence, New Rochelle) |
| SF13_ENG_041 | He is survived by his third wife, former television news correspondent Marilyn Berger; his sons, Steven and Jeffrey; his daughter, Lisa Cassara; his stepdaughter, Jilian Childers Hewitt, whom Hewitt adopted; and three grandchildren. | - | (Hewitt, ,, Jilian Childers Hewitt) → (Don Hewitt, per:children, Jilian Childers Hewitt) |
| SF13_ENG_044 | Dominick Dunne , a novelist and journalist who chronicled true - crime tales of the rich and infamous , including O . J . Simpson and Claus von Bulow , and in turn became a celebrity in his own right , died of bladder cancer Aug . 26 at his home in New York City . | (Dominick Dunne, ,, A Novelist) → (Dominick Dunne, per:title, A Novelist) | (Dunne, ,, novelist) → (Dominick Dunne, per:title, novelist) |
| SF13_ENG_044 | Dunne and his wife , Ellen Griffin Dunne , known as Lenny , were married in 1954 | (His Wife, ,, Ellen Griffin Dunne) → (Dominick Dunne, per:spouse, Ellen Griffin Dunne) | (Dunne, and, Ellen Griffin Dunne) → (Dominick Dunne, per:spouse, Ellen Griffin Dunne) |
| SF13_ENG_044 | Dunne was born in 1925 in Hartford, Connecticut, to a wealthy Roman Catholic family and grew up in some of the same social circles as the Kennedys. | (Dunne, was born in 1925 in, Connecticut) → (Dominick Dunne, per:stateorprovince_of_birth, Connecticut) | (Dunne, was born in 1925 in, Connecticut) → (Dominick Dunne, per:stateorprovince_of_birth, Connecticut) |
| SF13_ENG_044 | Dunne was part of a famous family that also included his brother, novelist and screenwriter John Gregory Dunne; his brother's wife, author Joan Didion; and his son, Griffin. | (His Brother, ,, Novelist) → (Dominick Dunne, per:title, Novelist) | (His Brother, ,, Novelist) → (His brother, per:title, Novelist) |
| SF13_ENG_047 | The women were accompanied by Reyna Luisa Tamayo, the mother of political prisoner Orlando Zapata, who died at age 42 in a hunger strike February 23 to protest prison conditions. | (Political Prisoner Orlando Zapata, ,, Reyna Luisa Tamayo) → (Orlando Zapata, per:children, Reyna Luisa Tamayo) | (Political Prisoner Orlando Zapata, ,, Reyna Luisa Tamayo) → (Orlando Zapata, per:children, Reyna Luisa Tamayo) |
| SF13_ENG_049 | A professor emeritus at Yale University , Mandelbrot was born in Poland but as a child moved with his family to France where he was educated . | (Mandelbrot, ,, Yale University) → (Benoit Mandelbrot, per:employee_or_member_of, Yale University) | (Mandelbrot, was born in, Poland) → (Benoit Mandelbrot, per:country_of_birth, Poland) |
| SF13_ENG_058 | Access Industries , a privately held company founded in 1986 by Len Blavatnik , has a diverse portfolio of investments in industry , real estate , media and telecommunications . | (Access Industries, ,, A Privately Held Company) → (Access Industries, properties, A Privately Held Company) | (Access Industries, founded in 1986 by, Len Blavatnik) → (Access Industries, org:founded_by, Len Blavatnik) |
| SF13_ENG_061 | InterContinental Hotels Group, owners of Holiday Inn Worldwide, has been ordered by a court to pay $25 million in damages to a franchisee for fraud. | (Intercontinental Hotels Group, ,, Holiday Inn Worldwide) → (InterContinental Hotels Group, org:parents, Holiday Inn Worldwide) | (Intercontinental Hotels Group, ,, Holiday Inn Worldwide) → (InterContinental Hotels Group, org:parents, Holiday Inn Worldwide) |
| SF13_ENG_079 | Water and its links to development, peace and conflict were key words in the annual sessions, Anders Berntell, executive director of Stockholm International Water Institute (SIWI), said in his opening address. | (Stockholm International, ,, Anders Berntell) → (Stockholm International Water Institute, per:top_members_or_employees, Anders Berntell) | (Stockholm International, , executive director, Anders Berntell) → (Stockholm International Water Institute, per:top_members_or_employees, Anders Berntell) |
| SF13_ENG_079 | Water and its links to development, peace and conflict were key words in the annual sessions, Anders Berntell, executive director of Stockholm International Water Institute (SIWI), said in his opening address. | (Stockholm International, executive director, Anders Berntell) → (Stockholm International Water Institute, per:top_members_or_employees, Anders Berntell) | (Stockholm International, , executive director, Anders Berntell) → (Stockholm International Water Institute, per:top_members_or_employees, Anders Berntell) |
| SF13_ENG_090 | ECO member countries include Afghanistan, Azerbaijan, Iran, Kazakhstan, Kyrgyzstan, Pakistan, Tajikistan, Turkey, Turkmenistan and Uzbekistan. | (Eco Member Countries, include, Azerbaijan) → (Economic Cooperation Organization, org:members, Azerbaijan) | (Eco Member Countries, ,, Azerbaijan) → (Economic Cooperation Organization, org:members, Azerbaijan) |
| SF13_ENG_096 | Freedom was founded in the 1930s by R.C. Hoiles and is still majority owned by the Hoiles family. | (Freedom, was founded in, The 1930S) → (Freedom Communications, org:date_founded, 1930-XX-XX) | (Freedom, was founded in, The 1930S) → (Freedom Communications, org:date_founded, 1930-XX-XX) |
| SF13_ENG_096 | A phone message left Sunday at Freedom's Irvine, California, headquarters was not immediately returned. | - | (Freedom'S Irvine, ,, California) → (Freedom Communications, org:stateorprovince_of_headquarters, California) |
| SF13_ENG_097 | STX Finland is part of the international STX Europe Group, with shipyards in Brazil, Norway, France, Romania and Vietnam. | (Stx Finland, of, The International Stx Europe Group) → (STX Finland, org:subsidiaries, The International Stx Europe Group) | (Stx Finland, of, The International Stx Europe Group) → (STX Finland, org:subsidiaries, The International Stx Europe Group) |

Figure 9: Unmapped facts: MaMa-BERT$_{LARGE}$ vs. MaMa-GPT-2$_{XL}$ on TAC KBP.

| ID | Sentence | Candidate facts to mapped facts |
|---|---|---|
| Appropriate_Rural_Technology_Institute | Appropriate Rural Technology Institute Appropriate Rural Technology Institute is an Indian non - governmental organization founded by a group of about 20 scientists and technologists to develop innovative and environmentally friendly rural technologies based on modern scientific knowledge in 1996 . | (Appropriate Rural Technology Institute, is, An Indian Non-Governmental Organization) → (appropriate_rural_technology_institute.Q16245764, country.P17, india.Q668) |
| Ayatollah_(Record_Producer) | Ayatollah Lamont Dorrell , known as Ayatollah , is a hip - hop record producer from Queens , New York who has produced for predominantly New York - based rappers including Mos Def , Talib Kweli , R . A . The Rugged Man , Tragedy Khadafi , Wordsworth , Vast Aire , Afu - Ra , Guru , M . O . P . , Inspectah Deck , Cormega , Ghostface Killah as well as many others . | (Ayatollah, is, A Hip-Hop Record Producer) → (ayatollah_(record_producer).Q4830984, occupation.P106, record_producer.Q183945) |
| Benjamin_Darbelet | Benjamin Darbelet Benjamin Darbelet is a French judoka . | (Benjamin Darbelet, is, A French Judoka) → (benjamin_darbelet.Q2456207, country_of_citizenship.P27, france.Q142); (Benjamin Darbelet, is, A French Judoka) → (benjamin_darbelet.Q2456207, sport.P641, judo.Q11420) |
| Bharat_Khawas | He plays for club Tribhuvan Army in Martyr ' s Memorial A - Division League as a striker . | (He, plays, Club Tribhuvan Army) → (bharat_khawas.Q4901128, member_of_sports_team.P54, nepal_army_club.Q6994327); (He, plays, Club Tribhuvan Army) → (bharat_khawas.Q4901128, sport.P641, association_football.Q2736) |
| Bill_White_(Shortstop) | White was born on May 1 , 1860 in Bridgeport , Ohio , and he began his professional baseball career in 1883 with the Pottsville Anthracites of the International Association . | (White, in, Bridgeport) → (bill_white_(shortstop).Q4911357, place_of_birth.P19, bridgeport_ohio.Q2062947) |
| Bob_Dylan.Q392 | Dylan married Sara Lownds , who had worked as a model and a secretary to Drew Associates , on November 22 , 1965 | (Dylan, married, Sara Lownds) → (bob_dylan.Q392, spouse.P26, Sara_Lownds.Q457433) |
| Bob_Dylan.Q392 | In 1991 , Dylan received a Grammy Lifetime Achievement Award from American actor Jack Nicholson . | (Dylan, received, Grammy Lifetime Achievement Award) → (bob_dylan.Q392, award_received.P166, Grammy_Lifetime_Achievement_Award.Q935843) |
| Boog_Powell | He also played for the Cleveland Indians and the Los Angeles Dodgers . | (He, also played for the, Cleveland Indians) → (boog_powell.Q4942810, member_of_sports_team.P54, cleveland_indians.Q642553) |
| Boog_Powell | He played in Major League Baseball as a first baseman and left fielder from through , most notably as a member of the Baltimore Orioles dynasty that won four American League pennants and two World Series championships from 1966 and 1971 . | (He, played, A First Baseman) → (boog_powell.Q4942810, position_played_on_team_/_speciality.P413, first_baseman.Q1326154) |
| Bouchraya_Hammoudi_Bayoun | He studied Economics at the University of Havana , Cuba . | (He, studied, Economics, Havana) → (bouchraya_hammoudi_bayoun.Q4949324, educated_at.P69, university_of_havana.Q837320) |
| Cezaro_Rossetti | Of Italian - Swiss derivation , he was born in Glasgow and lived in Britain . | (He, was, Glasgow) → (cezaro_rossetti.Q2947206, place_of_birth.P19, glasgow.Q4093) |
| Chan_Sau_Ying | Chan Sau Ying Chan Sau Ying is a retired athlete from Hong Kong who specialised in the 100 metres hurdles . | (Chan Sau Ying, is, A Retired Athlete) → (chan_sau_ying.Q21858345, sport.P641, sport_of_athletics.Q542) |
| Christian_Mcgrath | He was born in the Bronx in 1972 and became interested in art and illustration as a small child . | (He, was born in, The Bronx) → (christian_mcgrath.Q5109897, place_of_birth.P19, the_bronx.Q18426) |
| Cliff_Michelmore | He was a regular presenter on BBC1 ' s " Holiday " programme from 1969 to 1986 , and presented other shows for BBC TV , ITV and BBC Radio . | (He, was, A Regular Presenter) → (cliff_michelmore.Q5132704, occupation.P106, television_presenter.Q947873) |
| Crooked_I | Crooked I Dominick Antron Wickliffe , better known by his stage name Crooked I or Kxng Crooked , is an American rapper from Long Beach , California . | (Kxng Crooked, ,, An American Rapper) → (crooked_i.Q253695, country_of_citizenship.P27, united_states.Q30) |
| Denise_Johns | She graduated in 2002 , and moved to Atlanta , playing amateur beach volleyball on the east coast of America . | (She, , playing, Amateur Beach Volleyball) → (denise_johns.Q5257690, sport.P641, beach_volleyball.Q4543) |
| Diana_Rast | She also competed in the 1996 Summer Olympics . | (She, competed at, The 1996 Summer Olympics) → (diana_rast.Q2491787, participant_of.P1344, 1996_summer_olympics.Q8531) |
| Eduard_Friedrich_Poeppig | Eduard Friedrich Poeppig Eduard Friedrich Poeppig was a German botanist , zoologist and explorer . | (Eduard Friedrich, was, A German Botanist) → (eduard_friedrich_poeppig.Q61501, field_of_work.P101, botany.Q441) |
| Edward_L._Romero | After the Vietnam war , he became a county chairman of Democratic Party . | (A County Chairman, of, Democratic Party) → (edward_l_romero.Q5344025, member_of_political_party.P102, democratic_party_(united_states).Q29552) |
| Endre_Tilli | Endre Tilli Endre Tilli was a Hungarian fencer . | (Endre Tilli, was, A Hungarian Fencer) → (endre_tilli.Q1642140, country_of_citizenship.P27, hungary.Q28); (Endre Tilli, was, A Hungarian Fencer) → (endre_tilli.Q1642140, sport.P641, fencing.Q12100) |
| Enrico_Porro | Porro competed at the 1908 Summer Olympics in London where he won the gold medal in Greco - Roman wrestling , the lightweight class . | (Porro, competed, The 1908 Summer Olympics) → (enrico_porro.Q1343905, participant_of.P1344, 1908_summer_olympics.Q8111) |
| Erfan_Zeneli | Erfan Zeneli Erfan Zeneli is a Finnish footballer of Kosovar Albanian descent who currently plays for HJK Helsinki as a striker or left winger . | (Erfan Zeneli, is, A Finnish Footballer) → (erfan_zeneli.Q2053367, country_of_citizenship.P27, finland.Q33); (Erfan Zeneli, is, A Finnish Footballer) → (erfan_zeneli.Q2053367, sport.P641, association_football.Q2736); (Erfan Zeneli, is, A Finnish Footballer) → (erfan_zeneli.Q2053367, country_for.P1532, finland.Q33) |
| Erminio_Favalli | He then moved to Mantova before to join Palermo , where he spent seven seasons as a " rosanero " mainstay . | (He, then moved to, Mantova) → (erminio_favalli.Q1405871, member_of_sports_team.P54, mantova_1911_s.s.d..Q430993) |
| George_Dowty | Dowty was born in Pershore , Worcestershire in 1901 . | (Dowty, was born in, Pershore) → (george_dowty.Q1507239, place_of_birth.P19, pershore.Q767858) |
| Harald_Fairhair | In the Saga of Harald Fairhair in " Heimskringla " , which is the most elaborate although not the oldest or most reliable source to the life of Harald , it is written that Harald succeeded , on the death of his father Halfdan the Black Gudrödarson , to the sovereignty of several small , and somewhat scattered kingdoms in Vestfold , which had come into his father ' s hands through conquest and inheritance | (Harald Fairhair, of his father, Halfdan the Black) → (harald_fairhair.Q206123, father.P22, halfdan_the_black.Q504932) |
| Hector_Dyer | Born in Los Angeles , California , Hector Dyer enrolled at the Stanford University and won the IC4A championships in in 1930 . | (Hector Dyer, enrolled at, The Stanford University) → (hector_dyer.Q1364518, educated_at.P69, stanford_university.Q41506) |
| Henny_Magnussen | Thanks to the support she received , she was able to practise in all the courts of Denmark from 1909 , becoming the first woman to do so . | (The First Woman, ,, Denmark) → (henny_magnussen.Q56577333, country_of_citizenship.P27, denmark.Q35) |
| Hina_Pervaiz_Butt | Hina Pervaiz Butt Hina Pervaiz Butt is a Pakistani politician who was a Member of the Provincial Assembly of the Punjab , since May 2013 . | (Hina Pervaiz Butt, is, A Pakistani Politician) → (hina_pervaiz_butt.Q47918210, occupation.P106, politician.Q82955) |
| Isabella_Of_Valois,_Duchess_Of_Bourbon | She was the wife of Peter I , Duke of Bourbon . | (She, was the wife of, Peter I) → (isabella_of_valois,_duchess_of_bourbon.Q2363495, spouse.P26, peter_i,_duke_of_bourbon.Q444668) |
| Jahir_Butrón | Then from 1995 , he played for Peruvian Second Division side Guardia Republicana . | (He, played for, Peruvian Second Division Side Guardia Republicana) → (jahir_butrón.Q6123259, member_of_sports_team.P54, guardia_republicana.Q5614012) |
| Jahir_Butrón | Jahir Butrón Jahir Butrón Gotuzzo is a Peruvian footballer who plays as a center back . | (Jahir Butrón Gotuzzo, is, A Peruvian Footballer) → (jahir_butrón.Q6123259, country_of_citizenship.P27, peru.Q419); (Jahir Butrón Gotuzzo, is, A Peruvian Footballer) → (jahir_butrón.Q6123259, sport.P641, association_football.Q2736) |
| Joan_Of_France,_Duchess_Of_Berry | She was the second daughter of King Louis XI of France and of his second wife Charlotte of Savoy ; her surviving siblings were King Charles VIII of France and Anne of France . | (She, was the second daughter, King Louis Xi) → (joan_of_france,_duchess_of_berry.Q236220, father.P22, louis_xi_of_france.Q6058) |
| John_Ogilvie_(Footballer) | He played for Hibs in their most successful era renowned for their team known as The Famous Five . | (He, played for, Hibs) → (john_ogilvie_(footballer).Q6251147, member_of_sports_team.P54, hibernian_f.c._Q192597) |
| Johnedel_Cardel | He moved to Sta . | (He, to, Sta) → (johnedel_cardel.Q56599981, member_of_sports_team.P54, sta._lucia_realtors.Q3547751) |
| Joseph_W._Nega | Joseph W . Nega Joseph W . Nega is a Judge of the United States Tax Court . | (Joseph W. Nega, is, A Judge) → (joseph_w._nega.Q16733242, occupation.P106, judge.Q16533) |
| José_Alejandro_Aguilar_López | José Alejandro Aguilar López José Alejandro Aguilar López is a Mexican politician from the National Action Party . | (José Alejandro Aguilar López, is, A Mexican Politician) → (josé_alejandro_aguilar_lópez.Q18385891, country_of_citizenship.P27, mexico.Q96); (José Alejandro Aguilar López, is, A Mexican Politician) → (josé_alejandro_aguilar_lópez.Q18385891, occupation.P106, politician.Q82955); (José Alejandro Aguilar López, Mexican politician, The National Action Party) → (josé_alejandro_aguilar_lópez.Q18385891, member_of_political_party.P102, national_action_party_(mexico).Q851087) |
| Juan_Bautista_Azopardo | Juan Bautista Azopardo was born in Senglea , Malta , the son of Rosina and Salvatore Azopardo . | (Juan Bautista Azopardo, ,, Malta) → (juan_bautista_azopardo.Q3805864, country_of_citizenship.P27, malta.Q233); (Juan Bautista Azopardo, was born, Senglea) → (juan_bautista_azopardo.Q3805864, place_of_birth.P19, senglea.Q846593) |
| Kevin_Kerr_(Scottish_Footballer) | He signed for AGOVV Apeldoorn of Netherlands ' Eerste Divisie in January 2012 after his contract in Bielefeld had expired the previous summer . | (He, signed for, Agovv Apeldoorn) → (kevin_kerr_(scottish_footballer).Q326761, member_of_sports_team.P54, agovv_apeldoorn.Q292618) |
| Lachezar_Kotev | Lachezar Kotev Lachezar Kotev is a Bulgarian footballer who plays as a midfielder for Vitosha Bistritsa . | (Lachezar Kotev, is, A Bulgarian Footballer) → (lachezar_kotev.Q49560773, sport.P641, association_football.Q2736); (Lachezar Kotev, is, A Midfielder) → (lachezar_kotev.Q49560773, position_played_on_team_/_speciality.P413, midfielder.Q193592) |
| Marc_Zeno | In 1988 , Zeno was drafted in the 7th round by the Pittsburgh Steelers . | (Zeno, was drafted in the, The Pittsburgh Steelers) → (marc_zeno.Q16196353, member_of_sports_team.P54, pittsburgh_steelers.Q191477) |
| Marcel_Dheere | Marcel Dheere Marcel Albert " Ching " Dheere was a Canadian professional ice hockey forward who played 11 games in the National Hockey League for the Montreal Canadiens . | (Marcel Albert "Ching, in, A Canadian Professional Ice Hockey) → (marcel_dheere.Q6756083, country_of_citizenship.P27, canada.Q16); (The National Hockey League, for, The Montreal Canadiens) → (marcel_dheere.Q6756083, member_of_sports_team.P54, montreal_canadiens.Q188143); (A Canadian Professional Ice Hockey, played, The National Hockey League) → (marcel_dheere.Q6756083, league.P118, national_hockey_league.Q1215892); (Marcel Albert "Ching, played, A Canadian Professional Ice Hockey) → (marcel_dheere.Q6756083, sport.P641, ice_hockey.Q41466) |
| Mark_Schultz_(Wrestler) | Mark Schultz was born in 1960 in Palo Alto , California to Dorothy Jean St . Germain and Philip Gary Schultz . | (Mark Schultz, was born in 1960 in, Dorothy Jean St. Germain) → (mark_schultz_(wrestler).Q3487332, place_of_birth.P19, palo_alto_california.Q47265) |
| Michael_Jones_(Footballer) | Michael Jones Michael Jones is an English footballer , who plays for Cefn Druids . | (Michael Jones, is, An English Footballer) → (michael_jones_(footballer).Q6831707, sport.P641, association_football.Q2736) |
| Mike_Bellotti | He is an alumnus of the Delta Sigma Phi fraternity . | (He, is an alumnus of, The Delta Sigma Phi Fraternity) → (mike_bellotti.Q14950800, educated_at.P69, california_state_university_east_bay.Q1026916) |
| Olga_Semenova_Tyan-Shanskaya | She learned to play chess at the age of 18 ; four years later she won the Leningrad city women ' s chess championship . | (She, to play, Chess) → (olga_semenova_tyan-shanskaya.Q2392240, sport.P641, chess.Q718) |
| Owen_Paterson | Owen Paterson Owen William Paterson is a British Conservative Party politician who was the Secretary of State for Environment , Food and Rural Affairs from 2012 to 2014 . | (Owen Paterson, is, A British Conservative Party Politician) → (owen_paterson.Q197894, occupation.P106, politician.Q82955) |
| Pan_Gongsheng | Pan Gongsheng Pan Gongsheng is a Chinese economist , banker , reformist and bureaucrat . | (Pan Gongsheng, is, A Chinese Economist) → (pan_gongsheng.Q9309080, country_of_citizenship.P27, china.Q148) |
| Paul_Carrington (American Football) | Paul Carrington Paul Carrington is a former American football defensive end . | (Paul Carrington, is, A Former American Football Defensive End) → (paul_carrington_(american_football).Q9057011, sport.P641, american_football.Q41323) |
| Peter_Jebsen | Peter Jebsen Peter Jebsen was a Norwegian businessperson and politician . | (Peter Jebsen, was, A Norwegian Businessperson) → (peter_jebsen.Q1774966, country_of_citizenship.P27, norway.Q20); (Peter Jebsen, was, A Norwegian Businessperson) → (peter_jebsen.Q1774966, occupation.P106, businessperson.Q43845); (Peter Jebsen, was, A Politician) → (peter_jebsen.Q1774966, occupation.P106, politician.Q82955) |
| Pina_Piovani | Pina Piovani Pina Piovani was an Italian stage and film actress . | (Pina Piovani, was, An Italian Stage And Film Actress) → (pina_piovani.Q3388583, country_of_citizenship.P27, italy.Q38) |
| Robert_G._Cole | Rapidly advancing through the ranks at Fort Benning as the parachute infantry battalions were expanded to regiments , he was a lieutenant colonel commanding the 3rd Battalion of the 502nd Parachute Infantry Regiment on June 6 , 1944 , the date of his unit ' s first combat jump . | (He, was, A Lieutenant Colonel) → (robert_g._cole.Q969944, military_rank.P410, lieutenant_colonel.Q493898) |
| Robert_Holt_Carpenter | Robert Holt Carpenter Robert Holt Carpenter was a New Zealand bookbinder , local politician , bookseller and character | (Robert Holt Carpenter, was, A New Zealand Bookbinder) → (robert_holt_carpenter.Q7345593, country_of_citizenship.P27, new_zealand.Q664) |
| Robert_Van_Boxel | He joined amateur side FC Lisse in summer 2015 from Sparta . | (He, joined, Amateur Side Fc Lisse) → (robert_van_boxel.Q2017870, member_of_sports_team.P54, fc_lisse.Q1978325) |
| Rolf_Maurer | Rolf Maurer Rolf Maurer was a Swiss road racing cyclist who competed professionally between 1960 and 1969 . | (Rolf Maurer, was, A Swiss Road Racing Cyclist) → (rolf_maurer.Q870132, country_of_citizenship.P27, switzerland.Q39); (Rolf Maurer, was, A Swiss Road Racing Cyclist) → (rolf_maurer.Q870132, sport.P641, road_bicycle_racing.Q3609) |
| Ron_Maudsley | Maudsley was born in Lostock Gralam , Cheshire , and educated at Malvern College and Birmingham University , then served in the Royal Army Service Corps during World War II . | (Maudsley, was born in, Lostock Gralam) → (ron_maudsley.Q16009653, place_of_birth.P19, lostock_gralam.Q4640202) |
| Safa_Al-Safi | Safa al - Safi Safa al - Din Mohammed al - Safi is an Iraqi politician and former Justice Minister who is currently Minister of State for the Council of Representatives . | (Mohammed Al-Safi, is, An Iraqi Politician) → (safa_al-safi.Q7398343, occupation.P106, politician.Q82955) |
| Salvador_Luria | In 1959 , he became chair of Microbiology at the Massachusetts Institute of Technology . | (He, became chair of, Microbiology) → (salvador_luria.Q205667, field_of_work.P101, microbiology.Q7193) |
| Sanele_Vavae_Tuilagi | He now plays for US Carcassonne . | (He, now plays, Us Carcassonne) → (sanele_vavae_tuilagi.Q3472139, member_of_sports_team.P54, us_carcassonne.Q3550526) |
| Seth_Plum | Seth Plum Seth Plum was an English international footballer who played as a wing half . | (Seth Lewis Plum, was, An English International Footballer) → (seth_plum.Q7456581, sport.P641, association_football.Q2736); (Seth Lewis Plum, was, An English International Footballer) → (seth_plum.Q7456581, member_of_sports_team.P54, england_national_football_team.Q47762) |
| Sofija_Korkutytė | Sofija Korkutytė Sofija Korkutytė was a Lithuanian rower who won three European titles in the eights event in 1963 , 1965 and 1967 ; she finished second in 1964 and 1966 . | (Sofija Korkutytė, was, A Lithuanian Rower) → (sofija_korkutytė.Q16014909, sport.P641, rowing_(sport).Q159354) |
| Thomas_Rogne | He made his first start for Celtic at home against Dundee United in a 1 – 0 victory on 20 February 2010 , he played very well and impressed the coach Tony Mowbray . | (He, at, Celtic) → (thomas_rogne.Q609801, member_of_sports_team.P54, celtic_f.c._Q19593) |
| Uri_Avnery | Avnery was born in Beckum , near Münster in Westphalia , as Helmut Ostermann , the youngest of four children , to a well - established German Jewish family , his father being a private banker in the town . | (Westphalia, in, Beckum) → (uri_avnery.Q325679, place_of_birth.P19, beckum_germany.Q2707) |
| Ursula_Lehr | She was a member of the Christian Democratic Union . | (She, of, The Christian Democratic Union) → (ursula_lehr.Q290708, member_of_political_party.P102, christian_democratic_union_of_germany.Q49762) |
| Valdívia_(Brazilian_Footballer) | Valdívia Wanderson Ferreira de Oliveira , commonly known as Valdívia , is a Brazilian footballer who plays as an attacking midfielder for Avaí , on loan from Internacional . | (A Brazilian Footballer, who plays as, An Attacking Midfielder) → (valdívia_(brazilian_footballer).Q18176477, position_played_on_team_/_speciality.P413, midfielder.Q193592) |
| Victor-François,_2Nd_Duc_De_Broglie | His victory over Prince Ferdinand at Bergen won him the rank of marshal of France from the French King Louis XV and the title of Prince of the Holy Roman Empire " Reichsfürst " from Holy Roman Emperor Francis I . In 1759 , he won the Battle of Bergen and followed that with the capture the city of Minden , later fighting at the Battle of Minden under the command of Contades , whom he would succeed in command . | (His victory, won him the, Marshal) → (victor-françois,_2nd_duc_de_broglie.Q695929, award_received.P166, marshals_of_france.Q474485) |

Figure 10: Mapped facts: MaMa-BERT_LARGE on Wikidata.

| ID | Sentence | Candidate facts to mapped facts |
|---|---|---|
| Appropriate_Rural_Technology_Institute | Appropriate Rural Technology Institute Appropriate Rural Technology Institute is an Indian non-governmental organization founded by a group of about 20 scientists and technologists to develop innovative and environmentally friendly rural technologies based on modern scientific knowledge in 1996. | (Appropriate Rural Technology Institute, is, An Indian Non-Governmental Organization) → (appropriate_rural_technology_institute.Q16245764, country.P17, india.Q668) |
| Ayatollah_(Record_Producer) | Ayatollah Lamont Dorrell, known as Ayatollah, is a hip-hop record producer from Queens, New York who has produced for predominantly New York-based rappers including Mos Def, Talib Kweli, R.A. The Rugged Man, Tragedy Khadafi, Wordsworth, Vast Aire, Afu-Ra, Guru, M.O.P., Inspectah Deck, Cormega, Ghostface Killah as well as many others. | (Ayatollah, is, A Hip-Hop Record Producer) → (ayatollah_(record_producer).Q4830984, occupation.P106, record_producer.Q183945) |
| Benjamin_Darbelet | Benjamin Darbelet Benjamin Darbelet is a French judoka. | (Benjamin Darbelet, is, A French Judoka) → (benjamin_darbelet.Q2456207, country_of_citizenship.P27, france.Q142); (Benjamin Darbelet, is, A French Judoka) → (benjamin_darbelet.Q2456207, sport.P641, judo.Q11420) |
| Bharat_Khawas | He plays for club Tribhuvan Army in Martyr's Memorial A-Division League as a striker. | (He, plays for, Club Tribhuvan Army) → (bharat_khawas.Q4901128, member_of_sports_team.P54, nepal_army_club.Q6994327); (He, plays for, Club Tribhuvan Army) → (bharat_khawas.Q4901128, sport.P641, association_football.Q2736) |
| Bill_White_(Shortstop) | White was born on May 1, 1860 in Bridgeport, Ohio, and he began his professional baseball career in 1883 with the Pottsville Anthracites of the International Association. | (White, in, Bridgeport) → (bill_white_(shortstop).Q4911357, place_of_birth.P19, bridgeport_ohio.Q2062947) |
| Bob_Dylan.Q392 | Dylan married Sara Lownds , who had worked as a model and a secretary to Drew Associates , on November 22 , 1965 | (Dylan, married, Sara Lownds) → (Bob_Dylan.Q392, spouse.P26, Sara_Lownds.Q457433) |
| Bob_Dylan.Q392 | In 1991 , Dylan received a Grammy Lifetime Achievement Award from American actor Jack Nicholson . | (Dylan, received, A Grammy Lifetime Achievement Award) → (Bob_Dylan.Q392, award_received.P166, Grammy_Lifetime_Achievement_Award.Q935843) |
| Boog_Powell | He also played for the Cleveland Indians and the Los Angeles Dodgers. | (He, also played for, The Cleveland Indians) → (boog_powell.Q4942810, member_of_sports_team.P54, cleveland_indians.Q642553) |
| Boog_Powell | He played in Major League Baseball as a first baseman and left fielder from through , most notably as a member of the Baltimore Orioles dynasty that won four American League pennants and two World Series championships between 1966 and 1971. | (He, played in, A First Baseman) → (boog_powell.Q4942810, position_played_on_team_/_speciality.P413, first_baseman.Q1326154) |
| Bouchraya_Hammoudi_Bayoun | He studied Economics at the University of Havana, Cuba. | (He, studied, University of Havana) → (bouchraya_hammoudi_bayoun.Q4949324, educated_at.P69, university_of_havana.Q837320) |
| Cezaro_Rossetti | Of Italian-Swiss derivation, he was born in Glasgow and lived in Britain. | (He, was born in, Glasgow) → (cezaro_rossetti.Q2947206, place_of_birth.P19, glasgow.Q4093) |
| Chan_Sau_Ying | Chan Sau Ying Chan Sau Ying is a retired athlete from Hong Kong who specialised in the 100 metres hurdles. | (Sau Ying, is, A Retired Athlete) → (chan_sau_ying.Q21858345, sport.P641, sport_of_athletics.Q542) |
| Christian_Mcgrath | He was born in the Bronx in 1972 and became interested in art and illustration as a small child. | (He, was born in, The Bronx) → (christian_mcgrath.Q5109897, place_of_birth.P19, the_bronx.Q18426) |
| Cliff_Michelmore | He was a regular presenter on BBC1's "Holiday" programme from 1969 to 1986, and presented other shows for BBC TV, ITV and BBC Radio. | (He, was, A Regular Presenter) → (cliff_michelmore.Q5132704, occupation.P106, television_presenter.Q947873) |
| Crooked_I | Crooked I Dominick Antron Wickliffe , better known by his stage name Crooked I or Kxng Crooked, is an American rapper from Long Beach, California. | (Crooked I, is, An American Rapper) → (crooked_i.Q253695, country_of_citizenship.P27, united_states.Q30) |
| Denise_Johns | She graduated in 2002, and moved to Atlanta, playing amateur beach volleyball on the east coast of America. | (She, playing, Amateur Beach Volleyball) → (denise_johns.Q5257690, sport.P641, beach_volleyball.Q4543) |
| Diana_M_Fennell | Fennell was born in Emporia, Virginia and attended the public schools there. | (Fennell, was born in, Emporia) → (diana_m_fennell.Q28037279, place_of_birth.P19, emporia_virginia.Q1777678) |
| Diana_Rast | She also competed at the 1996 Summer Olympics. | (She, also competed at, The 1996 Summer Olympics) → (diana_rast.Q2491787, participant_of.P1344, 1996_summer_olympics.Q8531) |
| Eduard_Friedrich_Poeppig | Eduard Friedrich Poeppig Eduard Friedrich Poeppig was a German botanist, zoologist and explorer. | (Eduard Friedrich, was a, German Botanist) → (eduard_friedrich_poeppig.Q61501, field_of_work.P101, botany.Q441) |
| Edward_L._Romero | After the Vietnam war, he became a county chairman of Democratic Party. | (He, of, Democratic Party) → (edward_l._romero.Q5344025, member_of_political_party.P102, democratic_party_(united_states).Q29552) |
| Endre_Tilli | Endre Tilli Endre Tilli was a Hungarian fencer. | (Endre Tilli, was, A Hungarian Fencer) → (endre_tilli.Q1642140, country_of_citizenship.P27, hungary.Q28); (Endre Tilli, was, A Hungarian Fencer) → (endre_tilli.Q1642140, sport.P641, fencing.Q12100) |
| Enrico_Porro | Porro competed at the 1908 Summer Olympics in London where he won the gold medal in Greco-Roman wrestling, in the lightweight class. | (Porro, competed at, The 1908 Summer Olympics) → (enrico_porro.Q1343905, participant_of.P1344, 1908_summer_olympics.Q8111) |
| Erfan_Zeneli | Erfan Zeneli Erfan Zeneli is a Finnish footballer of Kosovar Albanian descent who currently plays for HJK Helsinki as a striker or left winger. | (Erfan Zeneli, is, A Finnish Footballer) → (erfan_zeneli.Q2053367, country_of_citizenship.P27, finland.Q33); (Erfan Zeneli, is, A Finnish Footballer) → (erfan_zeneli.Q2053367, sport.P641, association_football.Q2736); (Erfan Zeneli, is, A Finnish Footballer) → (erfan_zeneli.Q2053367, sport_for.P1532, finland.Q33) |
| Erminio_Favalli | He then moved to Mantova before to join Palermo, where he spent seven seasons as a "rosanero" mainstay. | (He, then moved to, Mantova) → (erminio_favalli.Q1405871, member_of_sports_team.P54, mantova_1911_s.s.d..Q430993) |
| George_Dowty | Dowty was born in Pershore, Worcestershire in 1901. | (Dowty, was born in, Pershore) → (george_dowty.Q1507239, place_of_birth.P19, pershore.Q767858) |
| Harald_Fairhair | In the Saga of Harald Fairhair in "Heimskringla", which is the most elaborate although not the oldest or most reliable source to the life of Harald, it is written that Harald succeeded, on the death of his father Halfdan the Black Guðrøðarson, to the sovereignty of several small, and somewhat scattered kingdoms in Vestfold, which had come into his father's hands through conquest and inheritance. | (Harald, of his father, Halfdan the Black) → (harald_fairhair.Q206123, father.P22, halfdan_the_black.Q504932) |
| Hector_Dyer | Born in Los Angeles, California, Hector Dyer enrolled at the Stanford University and won the IC4A championships in 1930. | (Hector Dyer, at, The Stanford University) → (hector_dyer.Q1364518, educated_at.P69, stanford_university.Q41506) |
| Henny_Magnussen | Thanks to the support she received, she was able to practise in all the courts of Denmark from 1909, becoming the first woman to do so. | (She, , she, Denmark) → (henny_magnussen.Q56577333, country_of_citizenship.P27, denmark.Q35) |
| Hina_Pervaiz_Butt | Hina Pervaiz Butt Hina Pervaiz Butt is a Pakistani politician who was a Member of the Provincial Assembly of the Punjab, since May 2013. | (Hina Pervaiz Butt, is, A Pakistani Politician) → (hina_pervaiz_butt.Q47918210, occupation.P106, politician.Q82955) |
| Isabella_Of_Valois,_Duchess_Of_Bourbon | She was the wife of Peter I, Duke of Bourbon. | (She, was the wife of, Peter I) → (isabella_of_valois,_duchess_of_bourbon.Q2363495, spouse.P26, peter_i,_duke_of_bourbon.Q444668) |
| Jahir_Butrón | Then from 1995, he played for Peruvian Second Division side Guardia Republicana. | (He, played for, Peruvian Second Division Side Guardia Republicana) → (jahir_butrón.Q6123259, member_of_sports_team.P54, guardia_republicana.Q5614012) |
| Jahir_Butrón | Jahir Butrón Jahir Butrón Gotuzzo is a Peruvian footballer who plays as a center back. | (Jahir Butrón, is, A Peruvian Footballer) → (jahir_butrón.Q6123259, country_of_citizenship.P27, peru.Q419); (Jahir Butrón Gotuzzo, is, A Peruvian Footballer) → (jahir_butrón.Q6123259, sport.P641, association_football.Q2736) |
| Janina_Altman | In 1950, Hescheles emigrated to Israel, where she eventually earned a doctorate in chemistry at Technion . | (She, earned a doctorate in, Technion) → (janina_altman.Q22574794, educated_at.P69, technion_–_israel_institute_of_technology.Q333705) |
| Joan_Of_France,_Duchess_Of_Berry | She was the second daughter of King Louis XI of France and of his second wife Charlotte of Savoy; her surviving siblings were King Charles VIII of France and Anne of France. | (She, was the second daughter of, King Louis Xi) → (joan_of_france,_duchess_of_berry.Q236220, father.P22, louis_xi_of_france.Q8018) |
| John_Ogilvie_(Footballer) | He played for Hibs in their most successful era renowned for their front five known as The Famous Five. | (He, played for, Hibs) → (john_ogilvie_(footballer).Q6251147, member_of_sports_team.P54, hibernian_f.c..Q192597) |
| Johnedel_Cardel | He moved to Sta. | (He, moved to, Sta) → (johnedel_cardel.Q56599981, member_of_sports_team.P54, sta._lucia_realtors.Q3547751) |
| Joseph_W._Nega | Joseph W. Nega Joseph W. Nega is a Judge of the United States Tax Court. | (Joseph W. Nega, is, A Judge) → (joseph_w._nega.Q16733242, occupation.P106, judge.Q16533) |
| José_Alejandro_Aguilar_López | José Alejandro Aguilar López José Alejandro Aguilar López is a Mexican politician from the National Action Party. | (José Alejandro Aguilar López, is a, The National Action Party) → (josé_alejandro_aguilar_lópez.Q18385891, member_of_political_party.P102, national_action_party_(mexico).Q851087); (José Alejandro Aguilar López, is, A Mexican Politician) → (josé_alejandro_aguilar_lópez.Q18385891, country_of_citizenship.P27, mexico.Q96); (José Alejandro Aguilar López, is, A Mexican Politician) → (josé_alejandro_aguilar_lópez.Q18385891, occupation.P106, politician.Q82955) |
| Juan_Bautista_Azopardo | Juan Bautista Azopardo was born in Senglea, Malta, the son of Rosina and Salvatore Azopardo. | (Juan Bautista Azopardo, was born in, Senglea) → (juan_bautista_azopardo.Q3805864, place_of_birth.P19, senglea.Q846593); (Juan Bautista Azopardo, was born in, Malta) → (juan_bautista_azopardo.Q3805864, country_of_citizenship.P27, malta.Q233) |
| Kevin_Kerr_(Scottish_Footballer) | He signed for AGOVV Apeldoorn of Netherlands' Eerste Divisie in January 2012 after his contract in Bielefeld had expired the previous summer. | (He, signed for, Agovv Apeldoorn) → (kevin_kerr_(scottish_footballer).Q326761, member_of_sports_team.P54, agovv_apeldoorn.Q292618) |
| Lachezar_Kotev | Lachezar Kotev Lachezar Kotev is a Bulgarian footballer who plays as a midfielder for Vitosha Bistritsa. | (Lachezar Kotev, is, A Bulgarian Footballer) → (lachezar_kotev.Q49560773, sport.P641, association_football.Q2736) |
| Marc_Zeno | In 1988, Zeno was drafted in the 7th round by the Pittsburgh Steelers. | (Zeno, by, The Pittsburgh Steelers) → (marc_zeno.Q16196353, member_of_sports_team.P54, pittsburgh_steelers.Q191477) |
| Marcel_Dheere | Marcel Dheere Marcel Albert "Ching" Dheere was a Canadian professional ice hockey forward who played 11 games in the National Hockey League for the Montreal Canadiens. | (Marcel Dheere, was, A Canadian Professional Ice Hockey) → (marcel_dheere.Q6756083, country_of_citizenship.P27, canada.Q16); (Marcel Albert "Ching, was, A Canadian Professional Ice Hockey) → (marcel_dheere.Q6756083, sport.P641, ice_hockey.Q41466); (Marcel Albert "Ching, for, The Montreal Canadiens) → (marcel_dheere.Q6756083, member_of_sports_team.P54, montreal_canadiens.Q188143) |
| Mark_Schultz_(Wrestler) | Mark Schultz was born in 1960 in Palo Alto, California to Dorothy Jean St. Germain and Philip Gary Schultz. | (Mark Schultz, was born in 1960 in, Palo Alto) → (mark_schultz_(wrestler).Q3487332, place_of_birth.P19, palo_alto_california.Q47265) |
| Michael_Jones_(Footballer) | Michael Jones Michael Jones is an English footballer, who plays for Cefn Druids. | (Michael Jones, is, An English Footballer) → (michael_jones_(footballer).Q6831427, sport.P641, association_football.Q2736) |
| Olga_Semenova_Tyan-Shanskaya | She learned to play chess at the age of 18; four years later she won the Leningrad city women's chess championship. | (She, learned, Chess) → (olga_semenova_tyan-shanskaya.Q3922240, sport.P641, chess.Q718) |
| Owen_Paterson | Owen Paterson Owen William Paterson is a British Conservative Party politician who was the Secretary of State for Environment, Food and Rural Affairs from 2012 to 2014. | (Owen William Paterson, is, A British Conservative Party Politician) → (owen_paterson.Q197894, occupation.P106, politician.Q82955) |
| Pan_Gongsheng | Pan Gongsheng Pan Gongsheng is a Chinese economist, banker, reformist and bureaucrat. | (Pan Gongsheng, is, A Chinese Economist) → (pan_gongsheng.Q9309080, country_of_citizenship.P27, china.Q148) |
| Paul_Carrington_(American_Football) | Paul Carrington Paul Carrington is a former American football defensive end. | (Paul Carrington, is, A Former American Football Defensive End) → (paul_carrington_(american_football).Q9057011, sport.P641, american_football.Q41323) |
| Peter_Jebsen | Peter Jebsen Peter Jebsen was a Norwegian businessperson and politician. | (Peter Jebsen, was, A Norwegian Businessperson) → (peter_jebsen.Q1774966, country_of_citizenship.P27, norway.Q20); (Peter Jebsen, was, A Norwegian Businessperson) → (peter_jebsen.Q1774966, occupation.P106, businessperson.Q43845) |
| Pina_Piovani | Pina Piovani Pina Piovani was an Italian stage and film actress. | (Pina Piovani, was, An Italian Stage And Film Actress) → (pina_piovani.Q3388583, country_of_citizenship.P27, italy.Q38) |
| Robert_G._Cole | Rapidly advancing through the ranks at Fort Benning as the parachute infantry battalions were expanded to regiments, he was a lieutenant colonel commanding the 3rd Battalion of the 502nd Parachute Infantry Regiment on June 6, 1944, the date of his unit's first combat jump. | (He, was, A Lieutenant Colonel) → (robert_g._cole.Q969944, military_rank.P410, lieutenant_colonel.Q493898) |
| Robert_Holt_Carpenter | Robert Holt Carpenter Robert Holt Carpenter was a New Zealand bookbinder, local politician, bookseller and character. | (Robert Holt Carpenter, was, A New Zealand Bookbinder) → (robert_holt_carpenter.Q7345593, country_of_citizenship.P27, new_zealand.Q664) |
| Robert_Van_Boxel | He joined amateur side FC Lisse in summer 2015 from Sparta. | (He, joined, Amateur Side Fc Lisse) → (robert_van_boxel.Q2017870, member_of_sports_team.P54, fc_lisse.Q978325) |
| Rolf_Maurer | Rolf Maurer Rolf Maurer was a Swiss road racing cyclist who competed professionally between 1960 and 1969. | (Rolf Maurer, was, A Swiss Road Racing Cyclist) → (rolf_maurer.Q870132, country_of_citizenship.P27, switzerland.Q39); (Rolf Maurer, was, A Swiss Road Racing Cyclist) → (rolf_maurer.Q870132, sport.P641, road_bicycle_racing.Q3609) |
| Ron_Maudsley | Maudsley was born in Lostock Gralam, Cheshire, and educated at Malvern College and Birmingham University, then served in the Royal Army Service Corps during World War II. | (Maudsley, was born in, Lostock Gralam) → (ron_maudsley.Q16009653, place_of_birth.P19, lostock_gralam.Q4640202) |
| Safa_Al-Safi | Safa al-Safi Safa al-Din Mohammed al-Safi is an Iraqi politician and former Justice Minister who is currently Minister of State for the Council of Representatives. | (Mohammed Al-Safi, is, An Iraqi Politician) → (safa-al-safi.Q398343, occupation.P106, politician.Q82955) |
| Salvador_Luria | In 1959, he became chair of Microbiology at the Massachusetts Institute of Technology . | (He, became chair of, Microbiology) → (salvador_luria.Q205667, field_of_work.P101, microbiology.Q7193) |
| Sanele_Vavae_Tuilagi | He now plays for US Carcassonne. | (He, now plays for, Us Carcassonne) → (sanele_vavae_tuilagi.Q3472139, member_of_sports_team.P54, us_carcassonne.Q3550526) |
| Seth_Plum | Seth Plum Seth Lewis Plum was an English international footballer who played as a wing half. | (Seth Lewis Plum, was, An English International Footballer) → (seth_plum.Q7456581, sport.P641, association_football.Q2736); (Seth Lewis Plum, was, An English International Footballer) → (seth_plum.Q7456581, member_of_sports_team.P54, england_national_football_team.Q47762) |
| Sofija_Korkutytė | Sofija Korkutytė Sofija Korkutytė was a Lithuanian rower who won three European titles in the eights event in 1963, 1965 and 1967; she finished second in 1964 and 1966. | (Sofija Korkutytė, was, A Lithuanian Rower) → (sofija_korkutytė.Q16014909, sport.P641, rowing_(sport).Q159354) |
| Thomas_Rogne | He made his first start for Celtic at home against Dundee United in a 1–0 victory on 20 February 2010, he played very well and impressed the coach Tony Mowbray. | (He, first start for, Celtic) → (thomas_rogne.Q609801, member_of_sports_team.P54, celtic_f.c..Q19593) |
| Uri_Avnery | Avnery was born in Beckum, near Münster in Westphalia, as Helmut Ostermann, the youngest of four children, to a well-established German Jewish family, his father being a private banker in the town. | (Avnery, born in, Beckum) → (uri_avnery.Q325679, place_of_birth.P19, beckum_germany.Q2707) |
| Ursula_Lehr | She was a member of the Christian Democratic Union . | (She, was a member of, The Christian Democratic Union) → (ursula_lehr.Q290708, member_of_political_party.P102, christian_democratic_union_of_germany.Q49762) |
| Valdívia_(Brazilian_Footballer) | Valdívia Wanderson Ferreira de Oliveira , commonly known as Valdívia, is a Brazilian footballer who plays as an attacking midfielder for Avaí, on loan from Internacional. | (A Brazilian Footballer, plays as, An Attacking Midfielder) → (valdívia_(brazilian_footballer).Q18176477, position_played_on_team_/_speciality.P413, midfielder.Q193592) |
| Victor-François,_2Nd_Duc_De_Broglie | His victory over Prince Ferdinand at Bergen won him the rank of marshal of France from the French King Louis XV and the title of Prince of the Holy Roman Empire "Reichsfürst" from Holy Roman Emperor Francis I. In 1759, he won the Battle of Bergen and followed that with the capture the city of Minden, later fighting at the Battle of Minden under the command of Contades, whom he would succeed in command. | (His victory, won him the, Marshal) → (victor-françois,_2nd_duc_de_broglie.Q695929, award_received.P166, marshals_of_france.Q474485) |

Figure 11: Mapped facts: MAMA-GPT-2$_{\text{XL}}$ on Wikidata.

| ID | Sentence | BERT$_{\text{LARGE}}$ | Candidate facts to mapped facts GPT-2$_{\text{XL}}$ |
|---|---|---|---|
| Andy_Hilbert | In the same season he was claimed off waivers by the Pittsburgh Penguins on March 9, 2006. | - | (He, by, The Pittsburgh Penguins) → (andy_hilbert.Q526158, member_of_sports_team.P54, pittsburgh_penguins.Q193643) |
| Bob_Dylan.Q392 | Dylan married his backup singer Carolyn Dennis on June 4, 1986. | - | (Dylan, married, Carolyn Dennis) → (Bob_Dylan.Q392, spouse.P26, Carolyn_Dennis.Q5045345) |
| Bones_Hillman | During the hiatus of Midnight Oil , Hillman returned to New Zealand , working as a studio and live musician with Dave Dobbyn and recorded the album '' Available Light ''. | (Hillman, returned, New Zealand) → (bones_hillman.Q2910026, country_of_citizenship.P27, new_zealand.Q664) | - |
| Brent_Hinds | Hinds continues to concentrate on Mastodon, with the majority of his time spent touring or in the studio. | - | (His Time, with, Mastodon) → (brent_hinds.Q909746, member_of.P463, mastodon.(band).Q548844) |
| Butch_Vig | Their debut album, "Garbage", was an unexpected smash, selling over 4 million copies and certified double platinum in the UK, United States and Australia. | - | (Over 4 Million Copies, in, United States) → (butch_vig.Q451084, country_of_citizenship.P27, united_states.Q30) |
| C-Kan | In addition to being known as a rapper, C-Kan is also known for supporting the free use of marijuana in México. | - | (Marijuana, in, México) → (c-kan.Q27734073, country_of_citizenship.P27, mexico.Q96) |
| Charles_Gratiot_Sr. | Gratiot died of a stroke in St . Louis . | (Gratiot, died, St. Louis) → (charles_gratiot_sr.Q2959257, place_of_death.P20, st._louis.Q38022) | - |
| Cheryl_Chan | Cheryl Chan Cheryl Chan Wei Ling , PBM , is a Singaporean politician . | (Cheryl Chan Wei Ling, is, A Singaporean Politician) → (cheryl_chan.Q22003605, occupation.P106, politician.Q82955) | - |
| Cyba_Audi | Audi is an experienced moderator and presenter at local, regional, and international business events. | - | (Audi, is, Presenter) → (cyba_audi.Q18921358, occupation.P106, television.presenter.Q947873) |
| Douglas_Lenat | He is a Fellow of the AAAS , AAAI , and Cognitive Science Society , and an editor of the J . Automated Reasoning , J . Learning Sciences , and J . Applied Ontology . | (He, is a Fellow, Aaai) → (douglas_lenat.Q559334, member_of.P463, association_for_the_advancement_of_artificial_intelligence.Q2739680) | - |
| Douglas_Lenat | While attending the University of Pennsylvania, Lenat supported himself through programming, notably designing and developing a natural language interface to a U.S. Navy data base question-answering system serving as an early online shipboard operations manual used on US aircraft carriers. | - | (Lenat, ., The University of Pennsylvania) → (douglas_lenat.Q559334, educated_at.P69, university_of_pennsylvania.Q49117) |
| Fred_Stewart_(Football_Manager) | Despite managing for 39 years he only ever took charge of two clubs, Stockport County and Cardiff City, and he holds the record for longest serving manager in the history of both clubs. | - | (He, charge of, Stockport County) → (fred_stewart_(football_manager).Q5496345, member_of_sports_team.P54, stockport_county_f.c..Q18526) |
| Félix_Tanco | Born Félix Manuel de Jesús Tanco y Bosmeniel , in Bogotá , Colombia , he arrived in Cuba at a very young age . | (Bogotá, ., Cuba) → (félix_tanco.Q5511649, country_of_citizenship.P27, cuba.Q241) | - |
| Gary_Ablett_Jr. | He also won his second Brownlow Medal , becoming the first Gold Coast player to win the award and the 14th player in VFL / AFL history to win it twice . | (He, also won, His Second Brownlow Medal) → (gary_ablett_jr.Q3098509, award_received.P166, leigh_matthews_trophy.Q6519632); (He, also won, His Second Brownlow Medal) → (gary_ablett_jr..Q3098509, award_received.P166, brownlow_medal.Q3853498) | - |
| Geeto_Mongol | After meeting his would - be tag team partner , Bepo Mongol , Tattrie returned to the WWWF as Geto Mongol and The Mongols were brought to the United States in 1968 . | (The Mongols, were, The United States) → (geeto_mongol.Q11324907, country_of_citizenship.P27, united_states.Q30) | - |
| Georgi_Plekhanov | Georgi Plekhanov Georgi Valentinovich Plekhanov was a Russian revolutionary , philosopher and a Marxist theoretician . | (Georgi Valentinovich Plekhanov, was, A Marxist Theoretician) → (georgi_plekhanov.Q182905, occupation.P106, philosopher.Q4964182) | - |
| Győző_Drozdy | Győző Drozdy Győző Drozdy was a Hungarian teacher , journalist , and politician . | (Győző Drozdy, was, Politician) → (győző_drozdy.Q910001, occupation.P106, politician.Q82955) | - |
| Horace_Rawlins | Rawlins died on 22 January 1935 in a nursing home in Reading , Berkshire . | (Rawlins, died, Reading) → (horace_rawlins.Q4436711, place_of_death.P20, reading._berkshire.Q161491) | - |
| Horace_Rawlins | He played in a professional event at Stanmore Golf Club in June 1894 but finished well out of the prizes. | - | (He, played in, Stanmore Golf Club) → (horace_rawlins.Q4436711, sport.P641, golf.Q5377) |
| Hugo_Winterhalter | Winterhalter remained with RCA Victor until 1963, at which time he moved to Kapp; that same year, he also penned the main title theme for the film, "Diamond Head". | - | (Winterhalter, to, Kapp) → (hugo_winterhalter.Q1635872, record_label.P264, kapp_records.Q1728690) |
| Ioannis_Talianos | He competed in the men ´ s 4 × 100 metres relay at the 1924 Summer Olympics . | (He, competed in the, The 1924 Summer Olympics) → (ioannis_talianos.Q35692847, participant_of.P1344, 1924_summer_olympics.Q8132) | - |
| Joan_Of_France,_Duchess_Of_Berry | The nuns of the Order of the Annunciation of the Blessed Virgin Mary still maintain their way of life in monasteries in France, Belgium, Costa Rica and Poland. | - | (Their Way, in, France) → (joan_of_france,_duchess_of_berry.Q236220, country_of_citizenship.P27, france.Q142) |
| John_Maynard_Keynes | John Maynard Keynes John Maynard Keynes , 1st Baron Keynes , was a British economist , whose ideas fundamentally changed the theory and practice of macroeconomics and the economic policies of governments . | (John Maynard Keynes, was, A British Economist) → (john.maynard_keynes.Q9317, occupation.P106, economist.Q188094) | - |
| Jonathan_Culler | Currently , he is Class of 1916 Professor of English and Comparative Literature at Cornell University . | (He, is, English) → (jonathan_culler.Q933332, languages_spoken,_written_or_signed.P1412, english_language.Q1860) | - |
| Jordi_Cañas_Pérez | In the regional elections of Catalonia in November 2010 , he was chosen in the primary as number three for Ciutadans ´ candidacy in the province of Barcelona . | (The Province, of, Barcelona) → (jordi_cañas_pérez.Q557693, member_of_political_party.P102, citizens_(spanish_political_party).Q1393123) | - |
| Jules_Deligny | He competed in the freestyle lightweight event at the 1920 Summer Olympics . | (He, competed in the, The 1920 Summer Olympics) → (jules_deligny.Q16207961, participant_of.P1344, 1920_summer_olympics.Q8128) | - |
| Liz_Mcgregor | Liz McGregor Liz McGregor is a South African author and a journalist who worked for leading South African newspapers such as the " Sunday Times " and the " Rand Daily Mail ". | (Liz Mcgregor, is, A Journalist) → (liz_mcgregor.Q23541069, occupation.P106, journalist.Q1930187) | - |
| Louise_Shropshire | The granddaughter of slaves, Louise Shropshire was born Louise Jarrett on February 15, 1913 in Coffee County, Alabama. | - | (Louise Shropshire, was, Coffee County, Alabama) → (louise_shropshire.Q15430857, place_of_birth.P19, coffee_county,_alabama.Q485660) |
| Macit_Gürdal | He competed in the men ´ s tournament at the 1952 Summer Olympics . | (He, competed in the, The 1952 Summer Olympics) → (macit_gürdal.Q57313835, participant_of.P1344, 1952_summer_olympics.Q8407) | - |
| Markus_Pröll | At the beginning of his career , Pröll became a target of some ridicule because his name " Pröll " literally means something like " lout " in German . | ("Pröll, " in, German) → (markus_pröll.Q704630, country_of_citizenship.P27, germany.Q183) | - |
| Monde_Hadebe | Their Round Four match against the saw Hadebe score the first senior try of his career in a 27 – 10 victory , and the Sharks XV again finished top of the Southern Section log . | (His Career, ,, The Sharks Xv) → (monde_hadebe.Q6898677, member_of_sports_team.P54, sharks_(currie_cup).Q744636) | - |
| Neville_Southall | He moved on to Everton for £150,000 in 1981 and established himself as the club's first-choice goalkeeper for the 1983–84 season. | - | (He, moved on to, Everton) → (neville_southall.Q436650, member_of_sports_team.P54, everton_f.c..Q5794) |
| Olga_Semenova_Tyan-Shanskaya | She died in Leningrad . | (She, died, Leningrad) → (olga_semenova_tyan-shanskaya.Q2392240, place_of_death.P20, saint_petersburg.Q656) | - |
| Parvathy_Ratheesh | Parvathy Ratheesh made her film acting debut in 2015 similar to that of her younger brother , Padmaraj Ratheesh who also made his acting debut in the same year through " Fireman ". | (Her Film, that of her younger brother, Padmaraj Ratheesh) → (parvathy_ratheesh.Q19895785, sibling.P3373, padmaraj_ratheesh.Q19895782) | - |
| Peter_Dalla_Riva | In 1993, he was inducted into the Canadian Football Hall of Fame. | - | (He, was inducted into the, Fame) → (peter_dalla_riva.Q7173528, award_received.P166, canadian_football_hall_of_fame.Q3517653) |
| Pierre_Duhem | Unlike many former historians , who denigrated the Middle Ages, he endeavored to show that the Roman Catholic Church had helped foster Western science in one of its most fruitful periods. | - | (The Roman Catholic Church, had, Foster Western Science) → (pierre_duhem.Q314172, field_of_work.P101, philosophy_of_science.Q59115) |
| Rafael_Arcadio_Bernal_Supelano | He then served as bishop of the Roman Catholic Diocese of Arauca, Colombia, from 1990 to 2003 and as bishop of the Roman Catholic Diocese of Líbano–Honda, Colombia, from 2003 to 2004. | - | (The Roman Catholic Diocese, ,, Colombia) → (rafael_arcadio_bernal_supelano.Q2126904, country_of_citizenship.P27, colombia.Q739) |
| Rajeshwar_Dayal | The Government of India awarded him the second highest civilian award of the Padma Vibhushan, in 1969. | - | (Him, award, The Padma Vibhushan) → (rajeshwar_dayal.Q15448539, award_received.P166, padma_vibhushan.Q672392) |
| Ray_Johnson,_(American_Football) | Ray Johnson Raymond Robert Johnson was an American football defensive back who played three seasons in the National Football League with the Cleveland Rams and Chicago Cardinals. | (The National Football League, back who played, Raymond Robert Johnson) → (ray_johnson_(american_football).Q21077889, sport.P641, american_football.Q41323) | - |
| Sean_Hanish | Sean Hanish Sean Hanish is an American film writer , producer and director best known for " Saint Judy " , " Return to Zero " and " Sister Cities ". | (Sean Hanish, is, An American Film Writer) → (sean_hanish.Q19867622, country_of_citizenship.P27, united_states.Q30); (Sean Hanish, is, An American Film Writer) → (sean_hanish.Q19867622, occupation.P106, screenwriter.Q28389) | - |
| Seth_Plum | Born in Edmonton, Plum played professionally for Charlton Athletic, Chelsea and Southend United. | - | (Plum, played professionally for, Charlton Athletic) → (seth_plum.Q7456581, member_of_sports_team.P54, charlton_athletic_f.c..Q19462) |
| Seth_Plum | Plum received his only cap for England at age 23 while playing for Charlton Athletic , starting and playing the full 90 minutes in a 4 – 1 win over France on 10 May 1923 | (His Only Cap, playing for, Charlton Athletic) → (seth_plum.Q7456581, member_of_sports_team.P54, charlton_athletic_f.c..Q19462) | - |
| Stanislav_Nedkov | A two - time national champion in freestyle wrestling , he has also had first - place finishes in international tournaments in Russia , Moldova , Turkey , and Bulgaria . | (Turkey, , and, Bulgaria) → (stanislav_nedkov.Q4844375, country_of_citizenship.P27, bulgaria.Q219) | - |
| Stiliyan_Petrov | In addition he is Bulgaria's all-time most-capped player with 105 appearances for the side. | - | (He, is, Bulgaria's All-Time Most-Capped Player) → (stiliyan_petrov.Q166263, country_of_citizenship.P27, bulgaria.Q219) |
| Stiliyan_Petrov | On 14 January 2010 , it was announced that Petrov had come second in Bulgaria ' s Player of the Year . | (Petrov, in, Bulgaria'S Player) → (stiliyan_petrov.Q166263, country_of_citizenship.P27, bulgaria.Q219) | - |
| Thomas_A._Dunn | He graduated from DePaul University College of Law in 1971. | - | (He, graduated from, Depaul University College) → (thomas_a_dunn.Q33082990, educated_at.P69, depaul_university_college_of_law.Q5244034) |
| Thomas_Scott_(Commentator) | In 1803 , Scott left the Lock Hospital to become Rector of Aston Sandford in Buckinghamshire where he remained until his death in 1821 . | (His Death, he remained until, Aston Sandford) → (thomas_scott_(commentator).Q7793831, place_of_death.P20, aston_sandford.Q3088093) | - |
| Thor_Heyerdahl | Thor Heyerdahl Thor Heyerdahl was a Norwegian adventurer and ethnographer with a background in zoology, botany and geography. | - | (Thor Heyerdahl, was, A Norwegian Adventurer) → (thor_heyerdahl.Q133622, country_of_citizenship.P27, norway.Q20) |
| Venkatesh_Kulkarni | His first novel , " Naked in Deccan " , won the 1984 American Book Award of the Before Columbus Foundation and was listed among the top ten novels of the decade by the " Chicago Tribune ". | (His First Novel, ,, won, The 1984 American Book Award) → (venkatesh_kulkarni.Q7920091, award_received.P166, american_book_awards.Q463606) | - |
| Vitor_Castro_De_Souza | Vitor Castro De Souza Vitor Castro de Souza , or simply Vitor Castro, is a Brazilian striker. | - | (Vitor Castro De Souza, is, A Brazilian Striker) → (vitor_castro_de_souza.Q5604876, position_played_on_team_/_speciality.P413, forward_(association_football).Q280658) |
| Vyacheslav_Tsaryov | He made his debut in the Soviet Top League in 1990 for FC Dynamo Moscow . | (He, for, Fc Dynamo Moscow) → (vyacheslav_tsaryov.Q4503307, member_of_sports_team.P54, fc_dynamo_moscow.Q17497) | - |
| Zare_Markovski | Zare Markovski as player played in KK Rabotnički and MZT Skopje . | (Zare Markovski, played in, Kk Rabotnički) → (zare_markovski.Q4023877, member_of_sports_team.P54, kk_rabotnički.Q2603345) | - |

Figure 12: Mapped facts: MAMA-BERT$_{\text{LARGE}}$ vs. MAMA-GPT-2$_{\text{XL}}$ on Wikidata.

| ID | Sentence | Candidate facts to unmapped facts |
|---|---|---|
| Nick_Aldis | He was also a co - presenter of " Britain ' s Strongest Man " on Challenge TV in the United Kingdom . | (Challenge Tv, in, The United Kingdom) → (Challenge TV, in, united_kingdom.Q145) |
| Douglas_Bader | At the age of two , Bader joined his parents in India for a year . | (His Parents, in, India) → (douglas_bader.Q348780, in, india.Q668) |
| Douglas_Bader | He remained in France after the war , where , having attained the rank of major , he died in 1922 of complications from those wounds in a hospital in Saint - Omer , the same area where Bader would bail out and be captured in 1941 . | (He, in, France) → (douglas_bader.Q348780, in, france.Q142) |
| Douglas_Bader | On 2 July 1941 he was awarded the bar to his DSO. | (He, was awarded the bar, his DSO) → (douglas_bader.Q348780, was awarded the bar, distinguished_service_order.Q615838) |
| Helen_Storrow | His father , also named James Jackson Storrow , was a prominent attorney , whose clients included Alexander Graham Bell and the government of Venezuela ; his mother , Ann Maria Perry , was the grandchild of naval hero Commodore Oliver Hazard Perry , and a distant cousin of President Thomas Jefferson . | (His Father, was, A Prominent Attorney) → (His Father, occupation.P106, lawyer.Q40348) |
| Helen_Storrow | Helen Storrow Helen Osborne Storrow was a prominent American philanthropist , early Girl Scout leader , and chair of the World Committee of the World Association of Girl Guides and Girl Scouts for eight years . | (Helen Osborne Storrow, was, A Prominent American Philanthropist) → (helen_storrow.Q5703224, place_of_birth.P19, united_states.Q30) |
| Jacob_Van_Ruisdael | His cousin Jacob was a registered Mennonite in Amsterdam . | (His Cousin, was, A Registered Mennonite) → (His Cousin, was, A Registered Mennonite) |
| Jacob_Van_Ruisdael | He appears to have been strongly influenced by other contemporary local Haarlem landscapes , most notably Cornelis Vroom and Allaert van Everdingen . | (He, have been strongly influenced by, landscape painting) → (jacob_van_ruisdael.Q213612, have been strongly influenced by, landscape_art.Q191163) |
| Liaquat_Ali_Khan | Liaquat Ali Khan was educated at the Aligarh Muslim University in India , and then at Oxford University in the United Kingdom . | (Liaquat Ali Khan, in, India) → (liaquat_ali_khan.Q295713, in, india.Q668) |
| Pauline_Baynes | Baynes began her education at a convent school . | (Her Education, at, A Convent School) → (pauline_baynes.Q101951, at, catholic_school.Q1138671) |
| Pauline_Baynes | Her most vivid recollection of their New Year ' s Eve lunch at Magdalen College was of his gleefully picking nuts out of a bowl of Brussels sprouts . | (Their New Year'S Eve Lunch, at, Magdalen College) → (pauline_baynes.Q101951, at, magdalen_college._oxford.Q81162) |
| Pauline_Baynes | By the time that she left , she had already formed the ambition of becoming an illustrator . | (She, formed the ambition of becoming, An Illustrator) → (pauline_baynes.Q101951, formed the ambition of becoming, illustrator.Q644687) |
| Thor_Heyerdahl | He was an atheist . | (He, was, An Atheist) → (thor_heyerdahl.Q133622, was, An Atheist) |
| Thor_Heyerdahl | He is buried in the garden of the family home in Colla Micheri . | (He, is buried, Colla Micheri) → (thor_heyerdahl.Q133622, is buried, colla_micheri.Q3647771) |
| Neville_Southall | He moved on to Cheshire County League club Winsford United at the age of 20 . | (He, to, Cheshire County League Club Winsford United) → (neville_southall.Q436650, to, winsford_united_f.c..Q5213266) |
| Neville_Southall | He later became a player - coach at York City , Rhyl , Shrewsbury Town and Dagenham & Redbridge . | (He, became, A Player-Coach) → (neville_southall.Q436650, occupation.P106, a Player-Coach) |
| John_Maynard_Keynes | After the war , Winston Churchill attempted to check the rise of Keynesian policy - making in the United Kingdom and used rhetoric critical of the mixed economy in his 1945 election campaign . | (Keynesian Policy-Making, in, The United Kingdom) → (john_maynard_keynes.Q9317, in, united_kingdom.P145) |
| John_Maynard_Keynes | Apart from Great Britain , Keynes ' s attention was primarily focused on the United States . | (Keynes'S Attention, was, The United States) → (john_maynard_keynes.Q9317, was, united_states.Q30) |
| John_Maynard_Keynes | John Maynard Keynes John Maynard Keynes , 1st Baron Keynes , was a British economist , whose ideas fundamentally changed the theory and practice of macroeconomics and the economic policies of governments . | (John Maynard Keynes, was, A British Economist) → (john_maynard_keynes.Q9317, was, united_kingdom.P145) |
| Ernst_Haeckel | Ernst Haeckel Ernst Heinrich Philipp August Haeckel was a German zoologist , naturalist , philosopher , physician , professor , marine biologist , and artist who discovered , described and named thousands of new species , mapped a genealogical tree relating all life forms , and coined many terms in biology , including " ecology " , " phylum " , " phylogeny " , and " Protista . " | (August Haeckel, was, A German Zoologist) → (ernst_haeckel.Q48246, was, germany.Q183) |
| Ernst_Haeckel | Haeckel was awarded the title of Excellency by Kaiser Wilhelm II in 1907 and the Linnean Society of London ' s prestigious Darwin - Wallace Medal in 1908 . | (Haeckel, was awarded the, Excellency) → (ernst_haeckel.Q48246, award_received.P166, Excellency) |
| Ernst_Haeckel | He was also a pacifist until the First World War , when he wrote propaganda in favor of the war . | (He, was, A Pacifist) → (ernst_haeckel.Q48246, was, A Pacifist) |

Figure 13: Unmapped facts: MAMA-BERT$_{\text{LARGE}}$ on Wikidata.

| ID | Sentence | Candidate facts to unmapped facts |
|---|---|---|
| Douglas_Bader | He left St Edward's in early 1928, aged 18. | (He, left, St Edward) → (douglas_bader.Q348780, left, st_edward's_school._oxford.Q128260) |
| Douglas_Bader | Group Captain Sir Douglas Robert Steuart Bader, was a Royal Air Force flying ace during the Second World War. | (Douglas Bader Group Captain Sir Douglas Robert Steuart Bader, was, A Royal Air Force Flying Ace) → (douglas_bader.Q348780, was, flying_ace.Q222982) |
| Douglas_Bader | On 2 July 1941 he was awarded the bar to his DSO. | (He, was awarded the bar to, his DSO) → (douglas_bader.Q348780, was awarded the bar to, distinguished_service_order.Q615838) |
| Helen_Storrow | His father, also named James Jackson Storrow, was a prominent attorney, whose clients included Alexander Graham Bell and the government of Venezuela; his mother, Ann Maria Perry, was the grandchild of naval hero Commodore Oliver Hazard Perry, and a distant cousin of President Thomas Jefferson. | (His Father, was, A Prominent Attorney) → (His Father, occupation.P106, lawyer.Q40348) |
| Helen_Storrow | Helen Osborne Storrow was a prominent American philanthropist, early Girl Scout leader, and chair of the World Committee of the World Association of Girl Guides and Girl Scouts for eight years. | (Helen Osborne Storrow, was, A Prominent American Philanthropist) → (helen_storrow.Q5703224, was, united_states.Q30) |
| Jacob_Van_Ruisdael | Ruisdael lived and worked in Amsterdam for the rest of his life. | (Ruisdael, lived, Amsterdam) → (jacob_van_ruisdael.Q213612, lived, amsterdam.Q727); (Ruisdael, lived and worked in, Amsterdam) → (jacob_van_ruisdael.Q213612, lived and worked in, amsterdam.Q727) |
| Liaquat_Ali_Khan | Liaquat Ali Khan was educated at the Aligarh Muslim University in India, and then at Oxford University in the United Kingdom. | (Liaquat Ali Khan, was educated at, The Aligarh Muslim University) → (liaquat_ali_khan.Q295713, was educated at, aligarh_muslim_university.Q196544) |
| Liaquat_Ali_Khan | After the death of his father in 1919, Ali Khan, with British Government awarding the grants and scholarship, went to England, attending Oxford University's Exeter College to pursue his higher education. | (His Higher Education, pursue, Oxford University'S Exeter College) → (liaquat_ali_khan.Q295713, pursue, university_of_oxford.Q34433) |
| Liaquat_Ali_Khan | In 1940, Khan was made the deputy leader of the Muslim League Parliamentary party. | (Khan, was made the deputy leader of, The Muslim League Parliamentary Party) → (liaquat_ali_khan.Q295713, deputy leader, all-india_muslim_league.Q223898) |
| Pauline_Baynes | When Baynes's father retired, he left India and returned to England, settling with Baynes's mother in a house close to Baynes's own near Farnham in southwest Surrey. | (He, left, India) → (pauline_baynes.Q101951, left, india.Q668) |
| Pauline_Baynes | Baynes began her education at a convent school. | (Her Education, at, A Convent School) → (pauline_baynes.Q101951, at, catholic_school.Q1138671) |
| Pauline_Baynes | Her most vivid recollection of their New Year's Eve lunch at Magdalen College was of his gleefully picking nuts out of a bowl of Brussels sprouts. | (Their New Year'S Eve Lunch, at, Magdalen College) → (pauline_baynes.Q101951, at, magdalen_college._oxford.Q81162) |
| Pauline_Baynes | She was not a diligent student, frittering away her time on coffee and parties; she left the Slade without a qualification, But she did achieve the distinction, one shared with her sister, of exhibiting at the Royal Academy of Arts in 1939. In 1940, a year into World War II, both Baynes sisters joined the Women's Voluntary Service. | (She, without a qualification, The Slade) → (pauline_baynes.Q101951, without a qualification, slade_school_of_fine_art.Q1399299) |
| Thor_Heyerdahl | His theories rarely won any scientific acceptance, whereas Heyerdahl himself rejected all scientific criticism and concentrated on publishing his theories in popular books aimed at the general public. | (His Theories, at, The General Public) → (thor_heyerdahl.Q133622, at, general_public) |
| Thor_Heyerdahl | He was an atheist. | (He, was, An Atheist) → (thor_heyerdahl.Q133622, was, An Atheist) |
| Neville_Southall | As a teenager, Southall had unsuccessful trials at Wrexham, Crewe Alexandra and Bolton Wanderers. | (Southall, at, Wrexham) → (neville_southall.Q436650, at, wrexham_a.f.c..Q18529); (Southall, at, Crewe Alexandra) → (neville_southall.Q436650, at, crewe_alexandra_f.c..Q19587) |
| John_Maynard_Keynes | John Maynard Keynes, 1st Baron Keynes, was a British economist, whose ideas fundamentally changed the theory and practice of macroeconomics and the economic policies of governments. | (John Maynard Keynes, was, A British Economist) → (john_maynard_keynes.Q9317, was, united_kingdom.Q145) |
| John_Maynard_Keynes | In January 1889 at the age of five and a half, Keynes started at the kindergarten of the Perse School for Girls for five mornings a week. | (Keynes, at, The Perse School) → (john_maynard_keynes.Q9317, at, the_perse_school.Q7756751) |
| John_Maynard_Keynes | In January 1892, at eight and a half, he started as a day pupil at St Faith's preparatory school. | (He, at, St Faith'S Preparatory School) → (john_maynard_keynes.Q9317, at, st_faith's_school.Q7593056) |
| John_Maynard_Keynes | Keynes was a lifelong member of the Liberal Party, which until the 1920s had been one of the two main political parties in the United Kingdom, and as late as 1916 had often been the dominant power in government. | (Keynes, was a lifelong member of, The liberal party) → (john_maynard_keynes.Q9317, was a lifelong member of, liberal_party_(uk).Q622441) |
| Ernst_Haeckel | Haeckel was awarded the title of Excellency by Kaiser Wilhelm II in 1907 and the Linnean Society of London's prestigious Darwin-Wallace Medal in 1908. | (Haeckel, was awarded the, Excellency) → (ernst_haeckel.Q48246, award_received.P166, Excellency) |
| Ernst_Haeckel | He was also a pacifist until the First World War, when he wrote propaganda in favor of the war. | (He, was, A Pacifist) → (ernst_haeckel.Q48246, was, A Pacifist) |
| Ernst_Haeckel | He was also a social Darwinist who believed that "survival of the fittest" was a natural law, and that struggle led to improvement of the race. | (He, was, A Social Darwinist) → (ernst_haeckel.Q48246, was, A Social Darwinist) |
| Ernst_Haeckel | Ernst Haeckel was born on 16 February 1834, in Potsdam. | (Haeckel, was born on 17 February 1834, in, Potsdam) → (ernst_haeckel.Q48246, was born on 17 February 1834, in, potsdam.Q1711) |

Figure 14: Unmapped facts: MAMA-GPT-2$_{\text{XL}}$ on Wikidata.

| ID | Sentence | Candidate facts to mapped facts | |
|---|---|---|---|
| | | BERT$_{\text{LARGE}}$ | GPT-2$_{\text{XL}}$ |
| Nick_Aldis | He was also a co - presenter of " Britain ' s Strongest Man " on Challenge TV in the United Kingdom . | (Challenge Tv, in, The United Kingdom) → (Challenge TV, in, united_kingdom.Q145) | - |
| Douglas_Bader | Group Captain Sir Douglas Robert Steuart Bader, was a Royal Air Force flying ace during the Second World War. | - | (Douglas Bader Group Captain Sir Douglas Robert Steuart Bader, was, A Royal Air Force Flying Ace) → (douglas_bader.Q348780, was, flying_ace.Q222982) |
| Douglas_Bader | At the age of two , Bader joined his parents in India for a year . | (His Parents, in, India) → (douglas_bader.Q348780, in, india.Q668) | - |
| Douglas_Bader | On 2 July 1941 he was awarded the bar to his DSO. | (He, was awarded the bar, his DSO) → (douglas_bader.Q348780, was awarded the bar, distinguished_service_order.Q615838) | (He, was awarded the bar to, his DSO) → (douglas_bader.Q348780, was awarded the bar to, distinguished_service_order.Q615838) |
| Helen_Storrow | His father , also named James Jackson Storrow , was a prominent attorney , whose clients included Alexander Graham Bell and the government of Venezuela ; his mother , Ann Maria Perry , was the grandchild of naval hero Commodore Oliver Hazard Perry , and a distant cousin of President Thomas Jefferson . | (His Father, was, A Prominent Attorney) → (His Father, occupation.P106, lawyer.Q40348) | (His Father, was, A Prominent Attorney) → (His Father, occupation.P106, lawyer.Q40348) |
| Helen_Storrow | Helen Osborne Storrow was a prominent American philanthropist, early Girl Scout leader, and chair of the World Committee of the World Association of Girl Guides and Girl Scouts for eight years. | (Helen Osborne Storrow, was, A Prominent American Philanthropist) → (helen_storrow.Q5703224, was, united_states.Q30) | (Helen Osborne Storrow, was, A Prominent American Philanthropist) → (helen_storrow.Q5703224, was, united_states.Q30) |
| Jacob_Van_Ruisdael | Ruisdael lived and worked in Amsterdam for the rest of his life. | - | (Ruisdael, lived, Amsterdam) → (jacob_van_ruisdael.Q213612, lived, amsterdam.Q727); (Ruisdael, lived and worked in, Amsterdam) → (jacob_van_ruisdael.Q213612, lived and worked in, amsterdam.Q727) |
| Jacob_Van_Ruisdael | His cousin Jacob was a registered Mennonite in Amsterdam . | (His Cousin, was, A Registered Mennonite) → (His Cousin, was, A Registered Mennonite) | - |
| Jacob_Van_Ruisdael | He appears to have been strongly influenced by other contemporary local Haarlem landscapists , most notably Cornelis Vroom and Allaert van Everdingen . | (He, have been strongly influenced by, landscape painting) → (jacob_van_ruisdael.Q213612, have been strongly influenced by, landscape_art.Q191163) | - |
| Liaquat_Ali_Khan | Liaquat Ali Khan was educated at the Aligarh Muslim University in India, and then at Oxford University in the United Kingdom. | (Liaquat Ali Khan, in, India) → (liaquat_ali_khan.Q295713, in, india.Q668) | (Liaquat Ali Khan, was educated at, The Aligarh Muslim University) → (liaquat_ali_khan.Q295713, was educated at, aligarh_muslim_university.Q196544) |
| Liaquat_Ali_Khan | After the death of his father in 1919, Ali Khan, with British Government awarding the grants and scholarship, went to England, attending Oxford University's Exeter College to pursue his higher education. | - | (His Higher Education, pursue, Oxford University'S Exeter College) → (liaquat_ali_khan.Q295713, pursue, university_of_oxford.Q34433) |
| Liaquat_Ali_Khan | In 1940, Khan was made the deputy leader of the Muslim League Parliamentary party. | - | (Khan, was made the deputy leader of, The Muslim League Parliamentary Party) → (liaquat_ali_khan.Q295713, deputy leader, all-india_muslim_league.Q223898) |
| Pauline_Baynes | When Baynes's father retired, he left India and returned to England, settling with Baynes's mother in a house close to Baynes's own near Farnham in southwest Surrey. | - | (He, left, India) → (pauline_baynes.Q101951, left, india.Q668) |
| Pauline_Baynes | Baynes began her education at a convent school. | (Her Education, at, A Convent School) → (pauline_baynes.Q101951, at, catholic_school.Q1138671) | (Her Education, at, A Convent School) → (pauline_baynes.Q101951, at, catholic_school.Q1138671) |
| Pauline_Baynes | Her most vivid recollection of their New Year's Eve lunch at Magdalen College was of his gleefully picking nuts out of a bowl of Brussels sprouts. | (Their New Year'S Eve Lunch, at, Magdalen College) → (pauline_baynes.Q101951, at, magdalen_college_oxford.Q81162) | (Their New Year'S Eve Lunch, at, Magdalen College) → (pauline_baynes.Q101951, at, magdalen_college_oxford.Q81162) |
| Pauline_Baynes | She was not a diligent student, frittering away her time on coffee and parties; and she left the Slade without a qualification, But she did achieve the distinction, one shared with her sister, of exhibiting at the Royal Academy of Arts in 1939, In 1940, a year into World War II, both Baynes sisters joined the Women's Voluntary Service. | - | (She, without a qualification, The Slade) → (pauline_baynes.Q101951, without a qualification, slade_school_of_fine_art.Q1399299) |
| Pauline_Baynes | By the time that she left , she had already formed the ambition of becoming an illustrator . | (She, formed the ambition of becoming, An Illustrator) → (pauline_baynes.Q101951, formed the ambition of becoming, illustrator.Q646487) | - |
| Thor_Heyerdahl | His theories rarely won any scientific acceptance, whereas Heyerdahl himself rejected all scientific criticism and concentrated on publishing his theories in popular books aimed at the general public. | - | (His Theories, at, The General Public) → (thor_heyerdahl.Q133622, at, general_public) |
| Thor_Heyerdahl | He was an atheist . | (He, was, An Atheist) → (thor_heyerdahl.Q133622, was, An Atheist) | (He, was, An Atheist) → (thor_heyerdahl.Q133622, was, An Atheist) |
| Thor_Heyerdahl | He is buried in the garden of the family home in Colla Micheri . | (He, is buried, Colla Micheri) → (thor_heyerdahl.Q133622, is buried, colla_micheri.Q3647771) | - |
| Neville_Southall | As a teenager, Southall had unsuccessful trials at Wrexham, Crewe Alexandra and Bolton Wanderers. | - | (Southall, at, Wrexham) → (neville_southall.Q436650, at, wrexham_a.f.c..Q18529); (Southall, at, Crewe Alexandra) → (neville_southall.Q436650, at, crewe_alexandra_f.c..Q19587) |
| Neville_Southall | He moved on to Cheshire County League club Winsford United at the age of 20 . | (He, to, Cheshire County League Club Winsford United) → (neville_southall.Q436650, to, winsford_united_f.c..Q5213266) | - |
| Neville_Southall | He later became a player - coach at York City , Rhyl , Shrewsbury Town and Dagenham & Redbridge . | (He, became, A Player-Coach) → (neville_southall.Q436650, occupation.P106, A Player-Coach) | - |
| John_Maynard_Keynes | In January 1889 at the age of five and a half, Keynes started at the kindergarten of the Perse School for Girls five mornings a week. | - | (Keynes, at, The Perse School) → (john_maynard_keynes.Q9317, at, the_perse_school.Q7756751) |
| John_Maynard_Keynes | In January 1892, at eight and a half, he started as a day pupil at St Faith's preparatory school. | - | (He, at, St Faith'S Preparatory School) → (john_maynard_keynes.Q9317, at, st_faith's_school.Q7593056) |
| John_Maynard_Keynes | After the war , Winston Churchill attempted to check the rise of Keynesian policy - making in the United Kingdom and used rhetoric critical of the mixed economy in his 1945 election campaign . | (Keynesian Policy-Making, in, The United Kingdom) → (john_maynard_keynes.Q9317, in, united_kingdom.P145) | - |
| John_Maynard_Keynes | Keynes was a lifelong member of the Liberal Party, which until the 1920s had been one of the two main political parties in the United Kingdom, and as late as 1916 had often been the dominant power in government. | - | (Keynes, was a lifelong member of, The Liberal Party) → (john_maynard_keynes.Q9317, was a lifelong member of, liberal_party_(uk).Q622441) |
| Ernst_Haeckel | Ernst Haeckel Ernst Heinrich Philipp August Haeckel was a German zoologist , naturalist , philosopher , physician , professor , marine biologist , and artist who discovered , described and named thousands of new species , mapped a genealogical tree relating all life forms , and coined many terms in biology , including " ecology " , " phylum " , " phylogeny " , and " Protista . " | (August Haeckel, was, A German Zoologist) → (ernst_haeckel.Q48246, was, germany.Q183) | - |
| Ernst_Haeckel | Ernst Haeckel was born on 16 February 1834, in Potsdam. | - | (Haeckel, was born on 17 February 1834, in, Potsdam) → (ernst_haeckel.Q48246, was born on 17 February 1834, in, potsdam.Q1711) |
| Ernst_Haeckel | Haeckel was awarded the title of Excellency by Kaiser Wilhelm II in 1907 and the Linnean Society of London's prestigious Darwin-Wallace Medal in 1908. | (Haeckel, was awarded the, Excellency) → (ernst_haeckel.Q48246, award_received.P166, Excellency) | (Haeckel, was awarded the, Excellency) → (ernst_haeckel.Q48246, award_received.P166, Excellency) |
| Ernst_Haeckel | He was also a pacifist until the First World War, when he wrote propaganda in favor of the war. | (He, was, A Pacifist) → (ernst_haeckel.Q48246, was, A Pacifist) | (He, was, A Pacifist) → (ernst_haeckel.Q48246, was, A Pacifist) |

Figure 15: Unmapped facts: MAMA-BERT$_{\text{LARGE}}$ vs. MAMA-GPT-2$_{\text{XL}}$ on Wikidata.

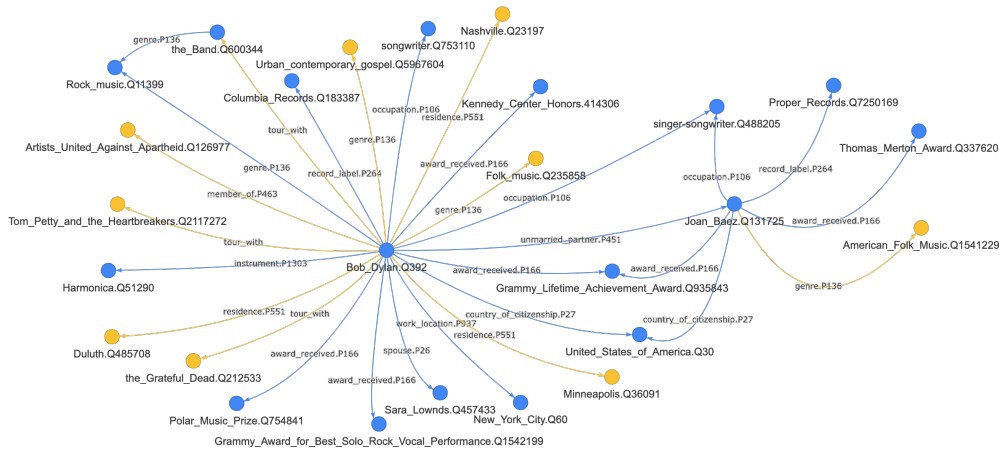

Figure 16: A snapshot subgraph of the open KG generated by MAMA using BERT$_{\text{LARGE}}$ from Wikipedia pages neighboring "Bob_Dylan". The blue node and arrow represent the mapped facts in the Wikidata schema, while the yellow node and arrow denote the unmapped facts in the open schema. We also visualize the correct facts that are new in Wikidata in yellow.

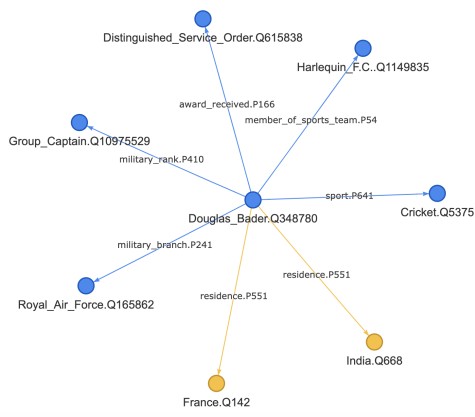

Figure 17: A snapshot subgraph of the open KG generated by MAMA-BERT_LARGE from the Wikipedia page "Douglas_Bader".

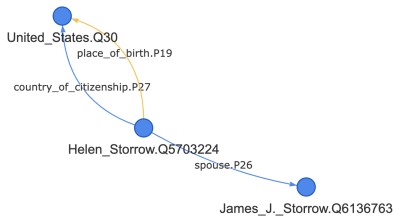

Figure 18: A snapshot subgraph of the open KG generated by MAMA-BERT_LARGE from the Wikipedia page "Helen_Storrow".

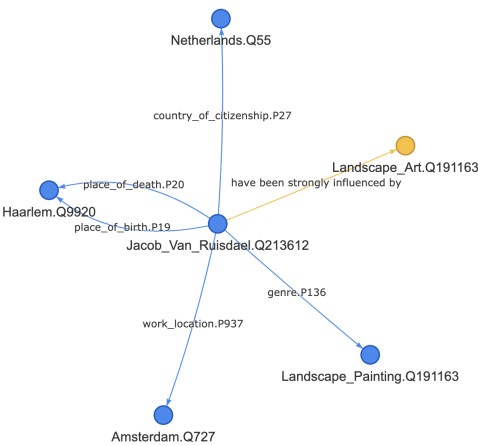

Figure 19: A snapshot subgraph of the open KG generated by MAMA-BERT_LARGE from the Wikipedia page "Jacob_van_Ruisdael".

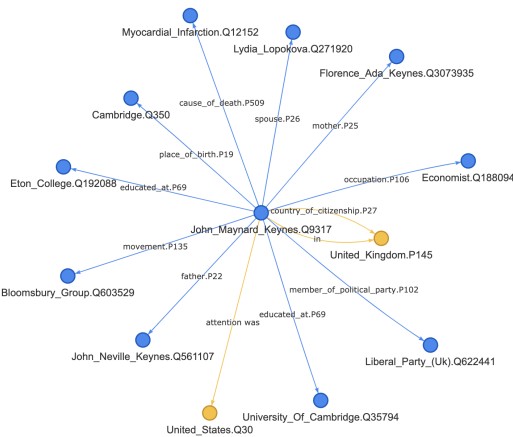

Figure 20: A snapshot subgraph of the open KG generated by MAMA-BERT_{LARGE} from the Wikipedia page "John_Maynard_Keynes".

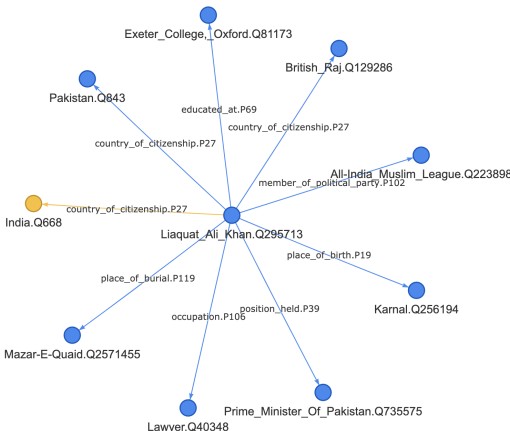

Figure 21: A snapshot subgraph of the open KG generated by MAMA-BERT_{LARGE} from the Wikipedia page "Liaquat_Ali_Khan".

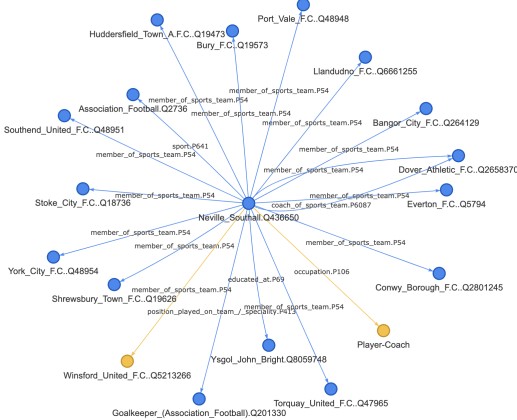

Figure 22: A snapshot subgraph of the open KG generated by MAMA-BERT_{LARGE} from the Wikipedia page "Neville_Southall".

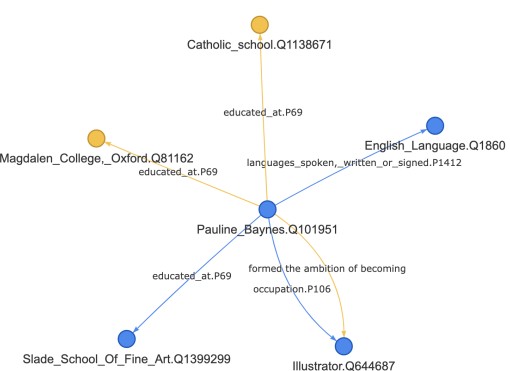

Figure 23: A snapshot subgraph of the open KG generated by MAMA-BERT$_{\text{LARGE}}$ from the Wikipedia page "Pauline_Baynes".

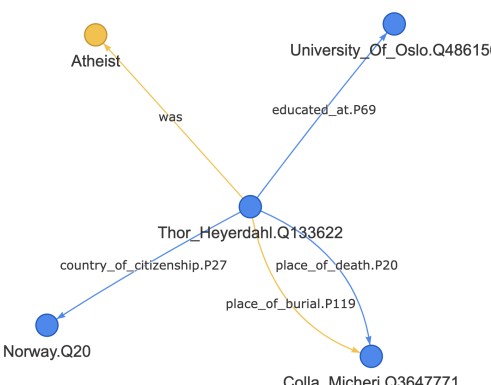

Figure 24: A snapshot subgraph of the open KG generated by MAMA-BERT$_{\text{LARGE}}$ from the Wikipedia page "Thor_Heyerdahl'.

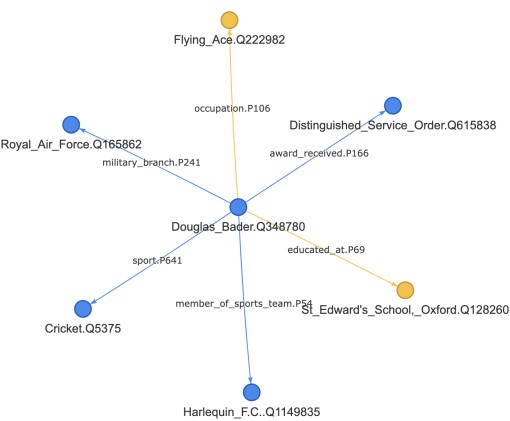

Figure 25: A snapshot subgraph of the open KG generated by MAMA-GPT-2$_{\text{XL}}$ from the Wikipedia page "Douglas_Bader".

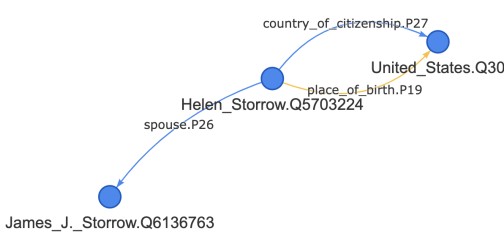

Figure 26: A snapshot subgraph of the open KG generated by MAMA-GPT-2_XL from the Wikipedia page "Helen_Storrow".

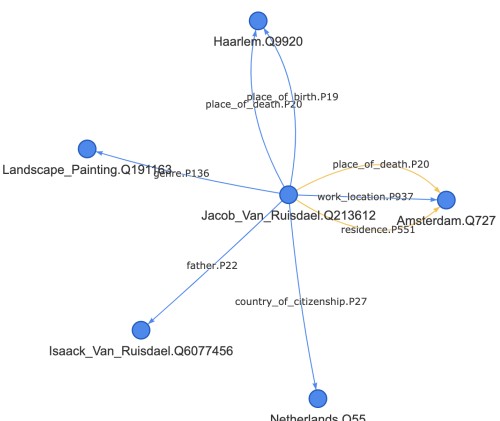

Figure 27: A snapshot subgraph of the open KG generated by MAMA-GPT-2_XL from the Wikipedia page "Jacob_van_Ruisdael".

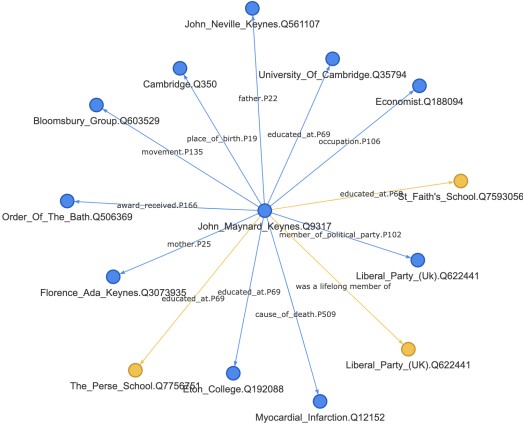

Figure 28: A snapshot subgraph of the open KG generated by MAMA-GPT-2_XL from the Wikipedia page "John_Maynard_Keynes".

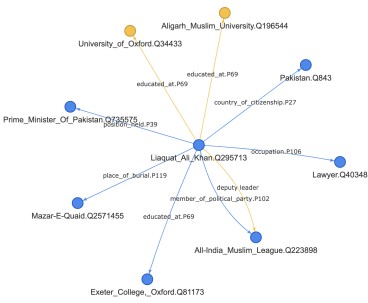

Figure 29: A snapshot subgraph of the open KG generated by MAMA-GPT-2$_{XL}$ from the Wikipedia page "Liaquat_Ali_Khan".

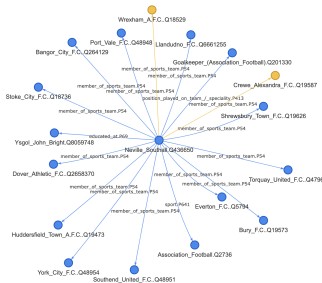

Figure 30: A snapshot subgraph of the open KG generated by MAMA-GPT-2$_{XL}$ from the Wikipedia page "Neville_Southall".

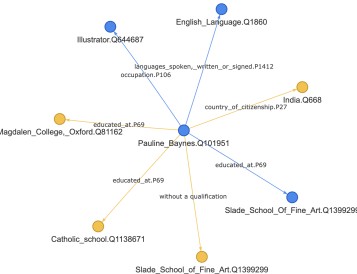

Figure 31: A snapshot subgraph of the open KG generated by MAMA-GPT-2$_{XL}$ from the Wikipedia page "Pauline_Baynes".

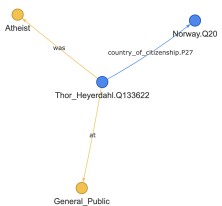

Figure 32: A snapshot subgraph of the open KG generated by MAMA-GPT-2$_{XL}$ from the Wikipedia page "Thor_Heyerdahl".

# Language Models are Open Knowledge Graphs

## Supplementary Materials

# Problem

- Knowledge graph construction requires human supervision

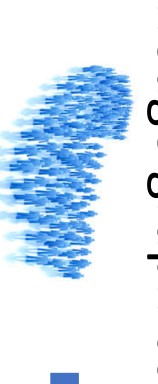

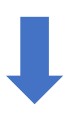

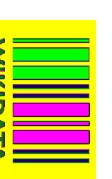

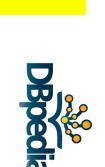

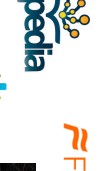

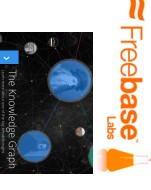

- Language models store knowledge

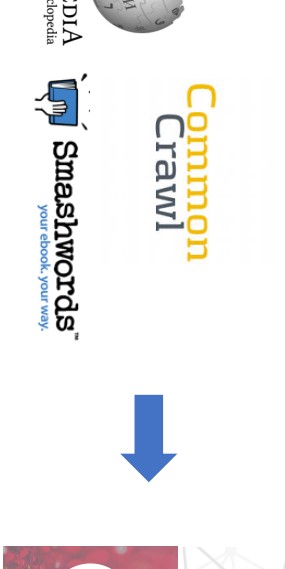

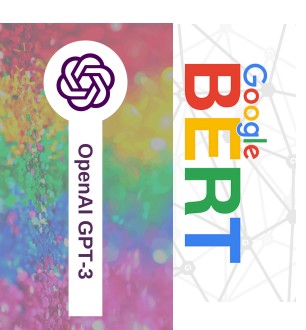

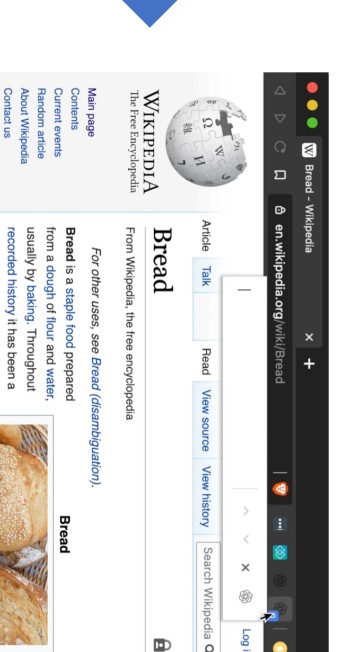

# Problem

- How to use **language models** to construct **knowledge graphs**?

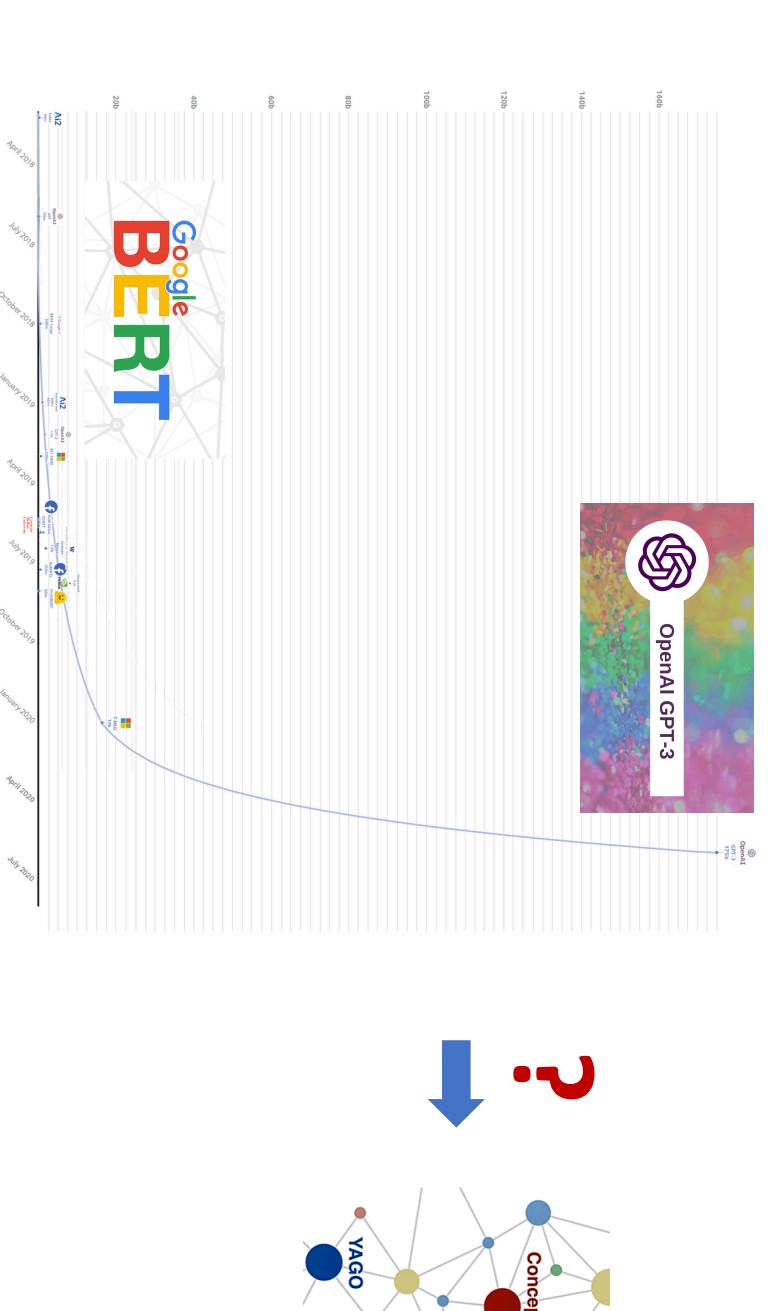

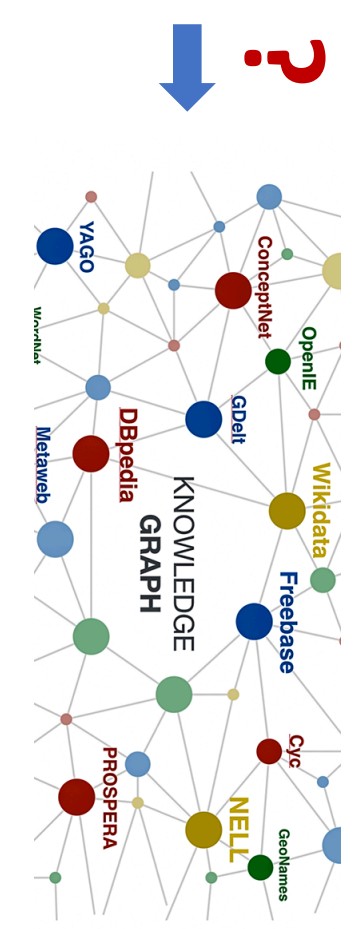

# Challenges

Language model 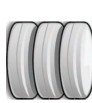

**General knowledge**

↓

Target corpus 

**Domain specific knowledge**

↓

Knowledge graph 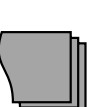

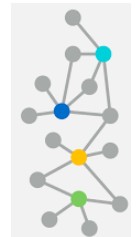

**Structured representation**

# Proposed Approach

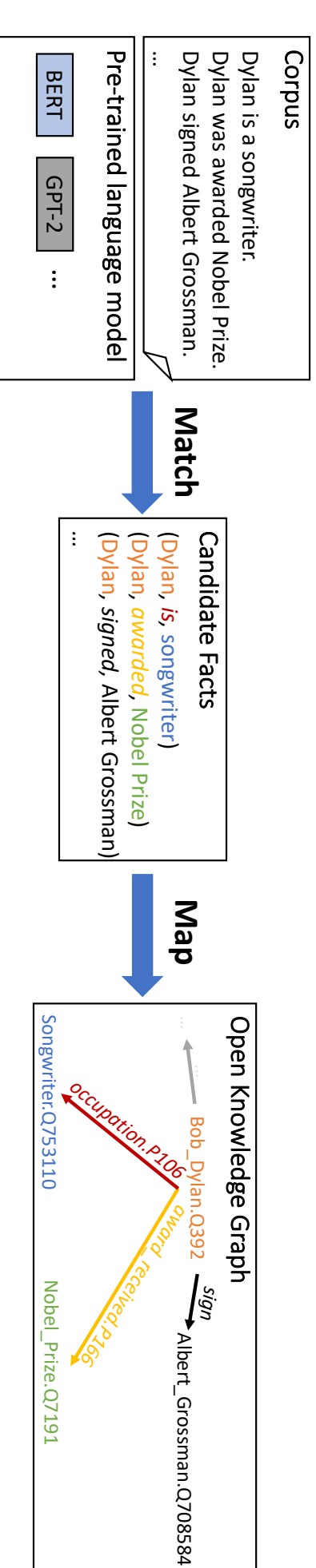

**Corpus**

Dylan is a songwriter.
Dylan was awarded Nobel Prize.
Dylan signed Albert Grossman.
...

**Pre-trained language model**

BERT    GPT-2    ...

**Match**

**Candidate Facts**

(Dylan, *is*, songwriter)
(Dylan, *awarded*, Nobel Prize)
(Dylan, *signed*, Albert Grossman)
...

**Map**

**Open Knowledge Graph**

Bob_Dylan.Q392

*occupation.P106*

Songwriter.Q753110

*award_received.P166*

Nobel_Prize.Q7191

*sign*

Albert_Grossman.Q708584

- MAMA constructs an open knowledge graph with **a single forward pass** of the language model (without fine-tuning) over the corpus
  - **Match**: generates a set of candidate facts from a textual corpus
  - **Map**: produces an open knowledge graph from the matched candidates

# Results on TAC KBP

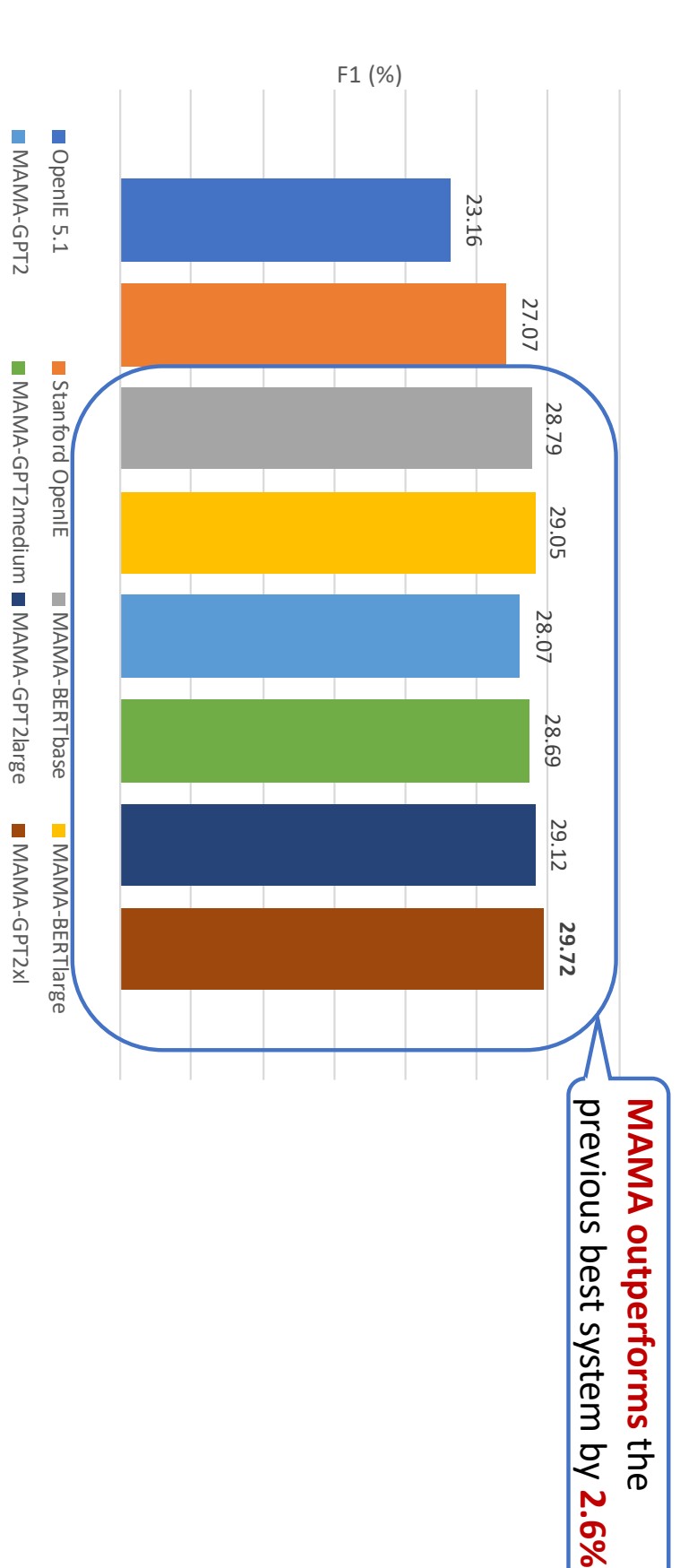

F1 (%)

- OpenIE 5.1
- Stanford OpenIE
- MAMA-GPT2medium
- MAMA-BERTbase
- MAMA-GPT2
- MAMA-GPT2large
- MAMA-BERTlarge
- MAMA-GPT2xl

23.16
27.07
28.79
29.05
28.07
28.69
29.12
**29.72**

**MAMA outperforms** the previous best system by **2.6%**

# Results on TAC KBP

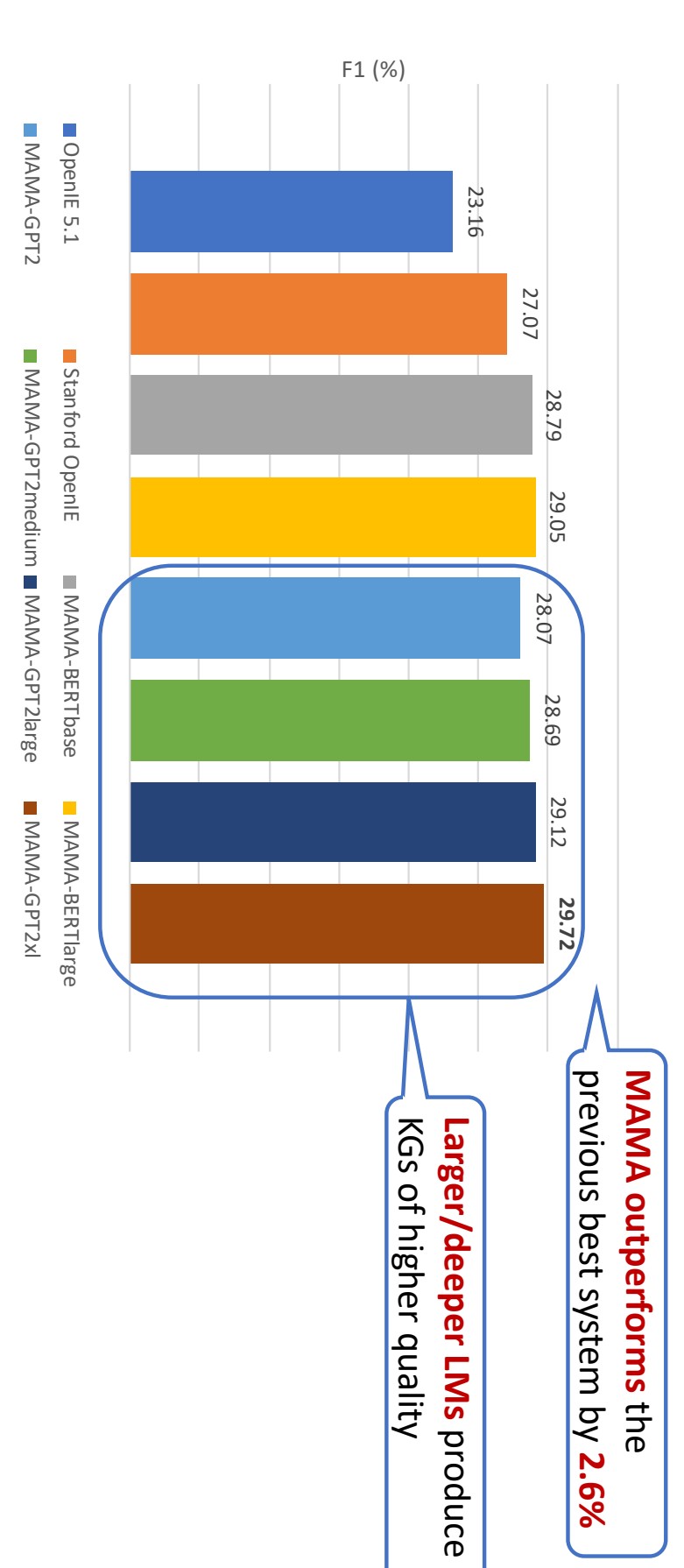

F1 (%)

- OpenIE 5.1 — 23.16
- MAMA-GPT2 — 27.07
- Stanford OpenIE — 28.79
- MAMA-GPT2medium — 29.05
- MAMA-GPT2large — 28.07
- MAMA-BERTbase — 28.69
- MAMA-BERTlarge — 29.12
- MAMA-GPT2xl — **29.72**

**MAMA outperforms** the previous best system by **2.6%**

**Larger/deeper LMs** produce KGs of higher quality

# Results on TAC KBP

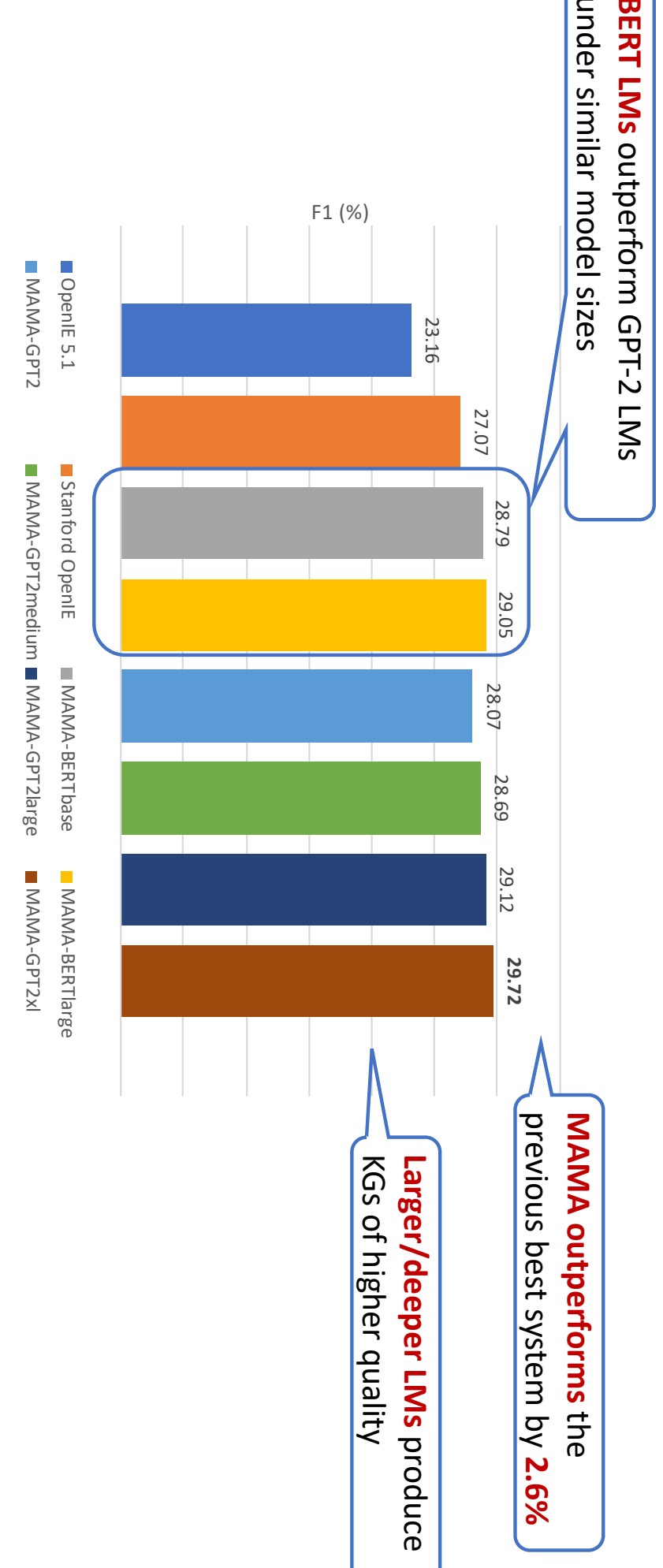

**BERT LMs** outperform GPT-2 LMs under similar model sizes

**MAMA outperforms** the previous best system by **2.6%**

**Larger/deeper LMs** produce KGs of higher quality

F1 (%)

- OpenIE 5.1 — 23.16
- MAMA-GPT2 — 27.07
- Stanford OpenIE — 28.79
- MAMA-GPT2medium — 29.05
- MAMA-GPT2large — 28.07
- MAMA-BERTbase — 28.69
- MAMA-BERTlarge — 29.12
- MAMA-GPT2xl — **29.72**

# Results on Wikidata

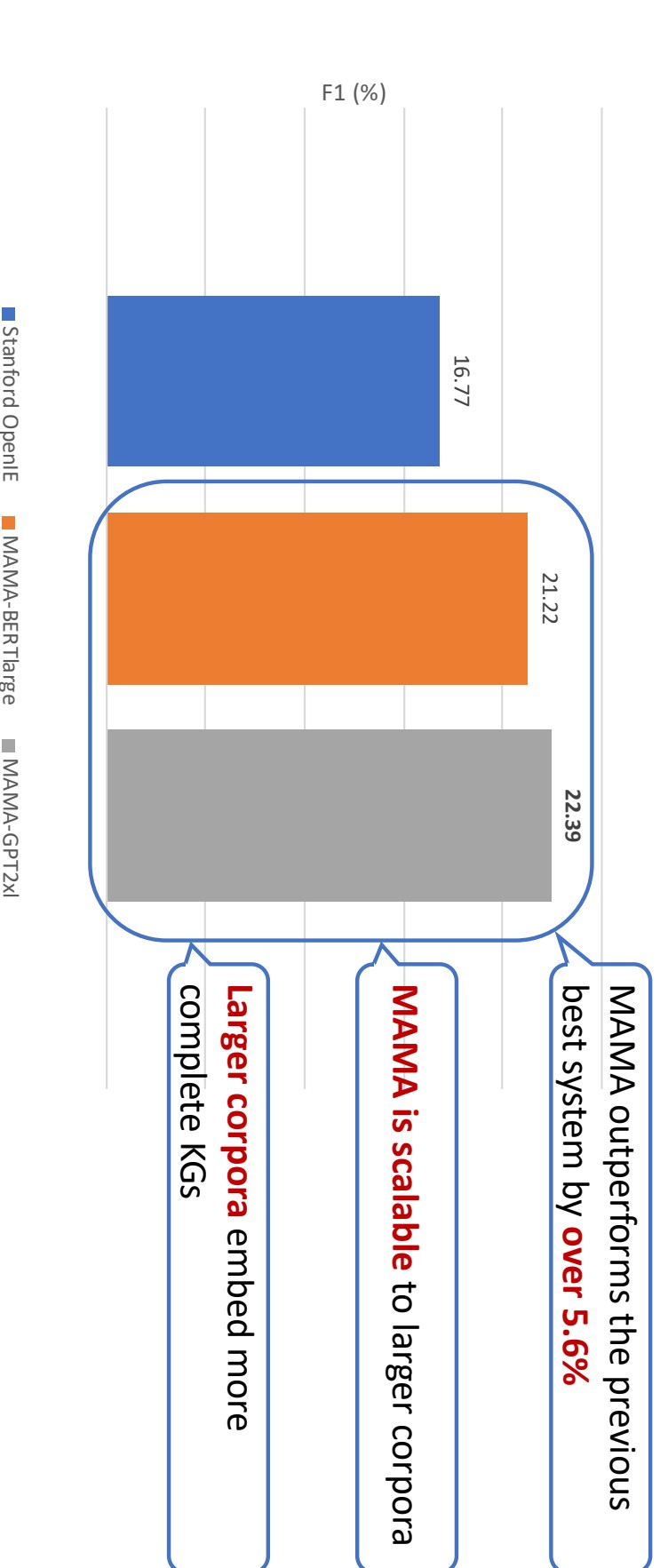

F1 (%)

- Stanford OpenIE
- MAMA-BERTlarge
- MAMA-GPT2xl

16.77

21.22

**22.39**

MAMA outperforms the previous best system by **over 5.6%**

**MAMA is scalable** to larger corpora

**Larger corpora** embed more complete KGs

# An Open KG Example

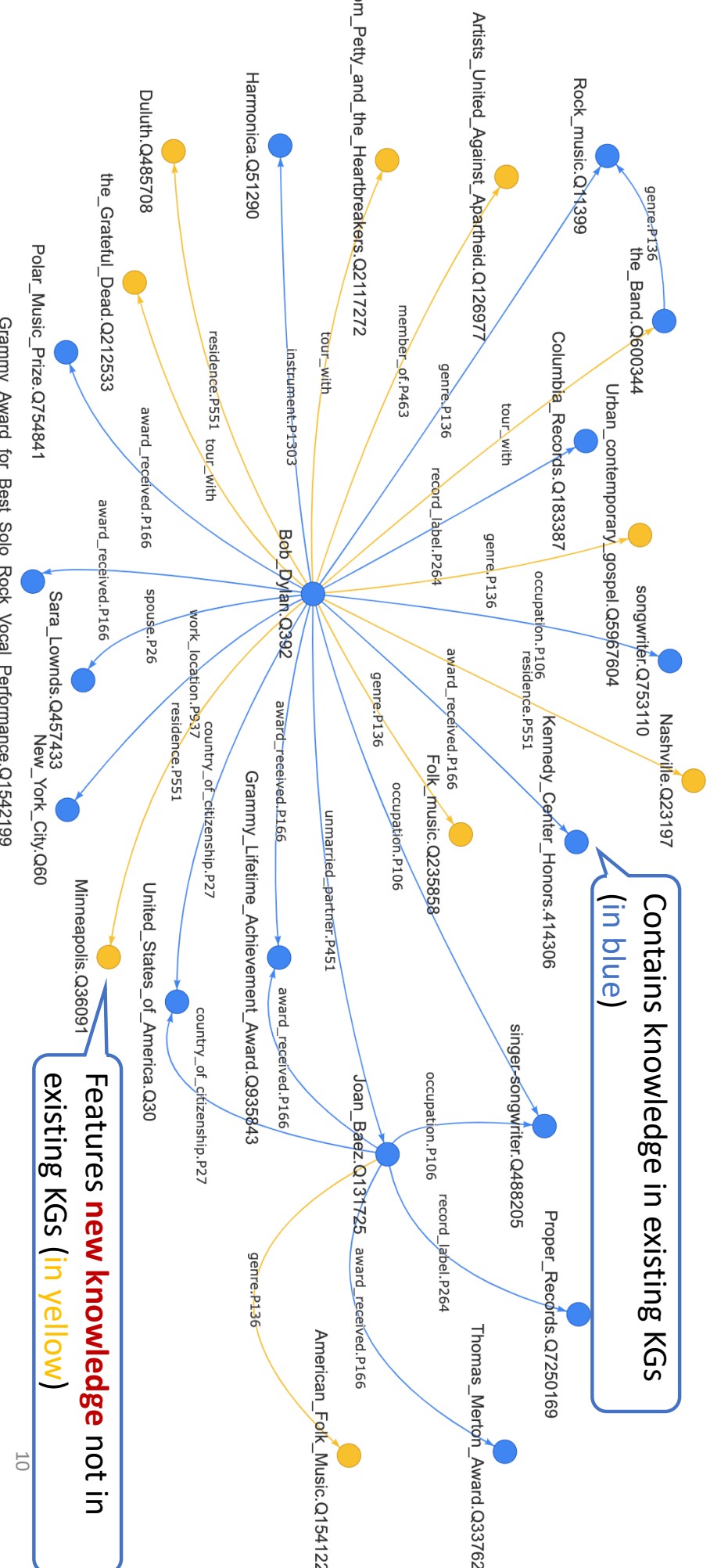

Contains knowledge in existing KGs (in blue)

Features **new knowledge** not in existing KGs (in yellow)

# An Open Question

- **Assumption**: Language models and knowledge graphs just encode the same world knowledge in two different formats, thus the two formats should be equal. But,

## Is the assumption TRUE?

Language Models ▇▇ Knowledge Graphs

# Contributions

- **Problem**: How to construct knowledge graphs from pre-trained language models.

- **Approach:** An unsupervised two-stage approach that constructs knowledge graphs with a single forward pass of the pre-trained language models without fine-tuning over the textual corpora (outperforming compared methods by over 5.6% in F1 on Wikidata).

- **Result:** Open knowledge graphs not only cover the knowledge already in existing knowledge graphs (e.g., Wikidata), but also features open factual knowledge that is new.

