# OpenReview forum: "Language Models are Open Knowledge Graphs"
_ICLR.cc/2021/Conference — Reject_

### Official Review · AnonReviewer1 · 2020-10-27
**Conceptually interesting paper but with some limitations and missing related work that may need to be addressed**

**Rating:** 4
**Confidence:** 4

**Review:**

Update:  Thanks for the detailed response.  I appreciate the additional figures and other results that you have provided.  However, it seems like there's still some open questions about the types of improvements being made and what this implies about the LM's attention mechanism.  It seems like more quantitative analysis would be needed to determine how much the LM's attention is correlating empirically to factual knowledge or if there are other factors that are affecting the downstream improvements.  Additionally, there's some limitations to the way the language model is being leveraged and the types of knowledge it can extract.

I also wanted to mention that I appreciate the addition of the suggested related work, but I would still suggest that the authors consider looking into more detailed means of comparison in the future (especially to the Petroni work), since this seemed to be a concern in multiple reviews.

----------------------------------------------
Original:
This paper is aimed at using pre-trained language models to create open-ended knowledge graphs.  They introduce an unsupervised approach (MAMA) to construct a knowledge graph in two phases in which they take a target corpus and output a knowledge graph.  In the first phase (match), they extract knowledge tuples from each sentence in a corpus using a beam search over the self-attention within a pre-trained language model. In the second phase (match), they ground facts to a knowledge graph schema by using combinations of entity linking and relation matching techniques from previous work.

While this is a promising direction for future research, I tend to lean towards rejection for a few reasons.  The main limitations seem to be: (1) the proposed method is a bit limited in that it can only be used with a corpus in which the target head, relation, and tail spans need to be directly mentioned in a single sentence (2) it’s not clear whether the quantitative improvements are due to factual knowledge in the pretrained model or the syntactic/semantic relationships encoded in the self-attention.  There's also some missing related work in extracting knowledge from pretrained models that should probably be discussed.

Contributions:
- A new algorithm for unsupervised knowledge graph creation from a target corpus
- Demonstrating the utility of large pre-trained language models towards knowledge graph creation (though, there are other works in this area that should probably be discussed more.  I will list a few in the limitations section.)
- Improvement over previous state-of-the-art models.

Strengths:
- The algorithms presented here are relatively straightforward but surprisingly effective.  They improve computationally over previous work.
- Most of the paper is clearly written.  The experiments seem easily reproduceable.
- The algorithm presented here is able to be used in an unsupervised way and can work with both open-ended and more structured knowledge graph schema.

Limitations:
- There are some limitations to the type of knowledge that can be extracted with the proposed approach: the only tuples that can be selected must be described within the target corpus in which the head and tail must appear within the same sentence.
- There is some missing prior work in creating knowledge graphs from pre-trained language models. To name a few:
  -“Language Models as Knowledge Bases?” EMNLP 2019
  -“Commonsense Knowledge Mining from Pretrained Models.” EMNLP 2019
  -“Comet: Commonsense Transformers for Automatic Knowledge Graph Construction.” ACL 2019
- Although the proposed model achieves quantitative improvements over Angeli et al. (2015), it seems unclear whether these improvements are due to the factual knowledge encoded in the pre-trained LM, as claimed.  It seems possible that these improvements could instead be due to the high quality semantic/syntactic relations encoded in the attention mechanism.
- The title of the paper is a bit strongly worded and may be over-claiming what is shown quantitatively in this paper.

Questions for the authors:
- How the parameter study was conducted? Was this using the development set, and if so for which dataset?

---

> ### Author Response · Authors · 2020-11-18
> **Response to AnonReviewer1 (Part 1)**
>
> We thank the reviewer for the valuable feedback. The concerns are addressed in the rebuttal revision. In particular, based on your suggestion, we have added more related work in Sec. 4, more analysis of the improvements and more detailed parameter study in both the main body of the paper and the newly added 30-page appendix. Additionally, Figure 4 to Figure 32 have been added to provide more insights into the proposed approach. We have also attached some slides to summarize the key idea and findings of the paper as supplementary materials at the end of the rebuttal revision for your reference. Please find our responses below.
>
> First of all, we are so glad that you generally like the paper.
>
> TYPE OF KNOWLEDGE: This is a very interesting point! As you have already identified, the work would be a promising direction for future research. As a generally new research topic, in this paper, we mainly focus on illustrating the effectiveness of the direction, and also want to enable follow-up studies from the whole community to explore more to further improve the quality of the generated knowledge graphs, e.g., exploring higher-order knowledgeable facts from multiple sentences. We also plan to leverage graph neural networks to generate more accurate relation phrases by leveraging the structural information, and reason over the generated knowledge graphs to generate high-order facts. The main focus of the paper is to demonstrate the knowledge is stored in the pre-trained language models, and show the key findings to the larger community. We hope you could understand our current position and future plan. Thanks again for the great point.
>
> RELATED WORK: We have followed your feedback to add more related work in Sec. 4. In particular, we highlight some of the key differences below. "Language Models as Knowledge Bases?" uses language models in Cloze-style statement, which is not relevant to knowledge graph construction. "Commonsense Knowledge Mining from Pretrained Models" uses language models to score the existing triplets for knowledge base completion (the triplets have been provided in the problem setting, where we generate the triplets from scratch), and targets commonsense knowledge, which is not directly associated with general-purpose knowledge graph construction (general knowledge vs. commonsense). "Comet: Commonsense Transformers for Automatic Knowledge Graph Construction" targets commonsense knowledge, and derives a supervised approach, while our approach targets general-purpose knowledge and the proposed approach is unsupervised. In general, the key difference is that our approach explicitly extracts knowledgeable facts from the language models. More broadly, as you can see, the work tries to help researchers explicitly understand what the language models learn, bridging the deep language model and knowledge graph communities through enhanced model transparency. Besides, the reach of the result is broad and has downstream utility for knowledge graph construction, deep neural network interpretation, and information extraction.
>
> IMPROVEMENT ANALYSIS: This is a great point! Based on your feedback, we have provided more mapped and unmapped results from Figure 4 to Figure 32 in the appendix for your reference. The extensive results have shown the effectiveness of the approach in recovering factual knowledge from the pre-trained language models. The results have shown that the knowledgeable triplets are constructed through the proposed approach from sentences with the complicated linguistic structure or long dependency. In fact, at the beginning of the project, we found there were intermediate facts that contain semantic/syntactic relations, the observation actually motivates us to design constraints in Sec. 2.1.2 to filter out the triplets with less knowledgeable information. As mentioned by Reviewer \#3, the work is the first evidence that the attention maps of pre-trained transformers contain paths that capture relational knowledge. The key motivation for inventing the unsupervised approach is based on the above evidence. Besides, the goal of the paper is to demonstrate the factual knowledge is contained within language models. A knowledge graph is also a container of knowledge. It is less possible for an approach that can not produce knowledge to perform competitively well in the knowledge graph construction task. The extensive experiments have successfully shown the effectiveness of the proposed approach. The additional results in the appendix from Figure 4 to Figure 32 are aiming to provide more evidence and insights for you. Thanks again for the valuable feedback.

---

> > ### Author Response · Authors · 2020-11-18
> > **Response to AnonReviewer1 (Part 2)**
> >
> > TITLE: We are open to a better title for the paper in the final copy. Please feel free to share your suggestion. The results of the paper show that knowledge graphs can be constructed with a single forward pass of the language models without supervision. The motivation of the current title is based on the results: we find the language models and knowledge graphs just encode the same world knowledge in two different formats, thus the two formats should be equal. Thanks for your valuable feedback.
> >
> > PARAMETER STUDY: We have added details about the parameter study in Sec. A.3 based on your feedback. The parameter study is conducted on a hold-out dataset, which is 20\% of oracle query entities in the TAC KBP dataset. Thanks for pointing this out.

---

### Official Review · AnonReviewer3 · 2020-10-28
**Interesting approach but unconvincing evaluation and a lack of details**

**Rating:** 4
**Confidence:** 4

**Review:**

This paper presents an unsupervised approach for extracting OpenIE style triples from a corpus. The approach leverages the internal attention maps of pretrained transformers to identify paths which correspond to relations between a head entity and a tail entity. The extracted open triples are then mapped, wherever possible, to an existing KG to create what is referred to as an Open KG.

Strengths:
- The unsupervised matching approach for extracting triples is quite novel and, to my knowledge, the first evidence that the attention maps of pretrained transformers contain paths which capture relational knowledge.

Weaknesses:
- The main evaluation in the paper is done on the grounded facts after the mapping stage. While the details are hazy, based on Section 2.2.1, it seems the mapping stage relies on a pre-existing KB aligned with a corpus to do the entity linking and relation linking steps. In this case another strong baseline would be to learn distantly supervised entity and relation extraction methods. Of course, this approach would not be able to produce the ungrounded facts, but when evaluating only the grounded facts it would be nice to see how the proposed approach compares to this more traditional setting.

- Related to the above, important details are missing about the interaction between the KG used for the entity and relation linkers and the KG used for evaluation. Specifically, what is the size of the KG available for entity / relation linkers, and were there any overlapping facts between this and the facts used for evaluation?

- Several recent papers have looked at probing LMs for knowledge facts (e.g. "Language Models as Knowledge Bases" Petroni et al, EMNLP 2019, and follow up papers). These are very relevant, as another approach for constructing KGs from pretrained LMs is to use natural language templates. But there is no discussion of these works in the paper.

- Selecting the threshold for the matching degree seems to be an important step influencing the quality of the final KG, but it seems to have been done in an ad-hoc manner here.

Other comments:
- Section 2.1 states that "self-attention weight matrix ... is the main container of the knowledge information in pre-trained LMs". This is clearly not true as a large amount of knowledge is also stored in the word embedding table.

- Section 3.2 has some confusing terminology where the facts supported by the KG schema are also referred to as "ungrounded" which contradicts the definitions in section 2.2.2.

---

> ### Author Response · Authors · 2020-11-18
> **Response to AnonReviewer3 (Part 1)**
>
> We thank the reviewer for the valuable feedback. The concerns are addressed in the rebuttal revision. In particular, we have followed your suggestion to add extensive details and results of the proposed approach (especially on the Map/Ground stage) in the main body of the paper as well as the newly added 30-page appendix. We have also attached some slides to summarize the key idea and findings of the paper as supplementary materials at the end of the rebuttal revision for your reference. Please find our responses below.
>
> First of all, we are so glad that you like the novelty of the paper.
>
> EVALUATION: The mapping/grounding stage is based on the entity linking and relation mapping, which are executed on the fly without relying on pre-existing alignment between the knowledge graph and corpus. The entity linker is a standalone module, which enriches the previous largest mention-to-entity dictionary (21 million entries in Spitkovsky & Chang (2012)) to 26 million entries by incorporating the new Wikipedia anchors. The relation mapping is also adapted to the task, which first leverages co-occurrence probability to rank the possible relation mapping pairs, then manually checks the correctness of the pairs. Next, the entity linker and relation mapping are used in the Map stage to project the candidate facts from a corpus to the facts in a reference knowledge graph. Finally, the quantitative evaluation is conducted for the mapped/grounded facts (shown in Table 2 and Table 3). The above details have been added in Sec A.1. The main focus of the paper is to demonstrate the knowledge is stored in the pre-trained language models. For this purpose, an unsupervised approach (no training/fine-tuning is involved) is desired. We successfully design the unsupervised approach to recover the knowledge using a single forward pass of the pre-trained language models without fine-tuning/training. We will incorporate the distantly supervised methods to enhance the mapping stage in the future. Thanks for the suggestion.
>
> DETAILS of MAPPING STAGE: Based on your suggestion, we have added the details of the mapping or grounding stage in the paper. On a high level, the goal of the Map stage is to ground candidate facts to a knowledge graph schema, and the Map stage is decoupled from the evaluation. For Map stage on TAC KBP, we link to the oracle annotation of the entities or spans in the TAC KBP corpus. On Wikidata, the entity linking method described in Sec. 2.2.1 is first leveraged to link entities in the candidate facts to Wikipedia anchors. We build an enhanced mention-to-entity dictionary based on Spitkovsky & Chang (2012). In particular, we add new Wikipedia anchors to the dictionary which results in 26 million entries comparing to 21 million entries in Spitkovsky & Chang (2012). Then a Wikipedia anchor to the Wikidata item dictionary is constructed and used to further link the entities to Wikidata. If the head or tail is a pronoun, we further use neuralcoref for coreference resolution. We use GloVe embedding for disambiguation. The relation mapping is constructed offline for TAC KBP and Wikidata respectively using the method in Sec. 2.2.1. For oracle facts in Wikidata, we only preserve those facts describing relations between entities that could be linked to a corresponding Wikipedia anchor. We rule out facts of attributes about entities and facts of auxiliary relations (such as topic’s main category.P901) and finally result in 27,368,562 oracle facts. The above details have been added in Sec. A.1. The statistics of the reference KGs are in Table 1. In addition, we have provided more mapped and unmapped results from Map/Ground stage in Figure 4 to Figure 32 in the appendix for your reference.

---

> > ### Author Response · Authors · 2020-11-18
> > **Response to AnonReviewer3 (Part 2)**
> >
> > RELATED WORK DISCUSSION: We have followed your suggestion to add the discussion in Sec. 4. The main difference between Petroni et al., 2019 (Petroni) and the paper is mainly three-fold: (1) Petroni aims to complete Cloze-style statement, e.g., given "Dylan is a \_", Petroni predicts which words/phrases should fill the blank "\_", which has no direct connection to knowledge graphs. While the proposed approach in this paper aims to solve a reasoning problem, e.g., given a passage, the proposed approach directly generates a triplet (Dylan, is, songwriter) at the first step, then maps the triplet to produce a knowledge graph. The proposed approach tries to solve a more challenging problem. (2) The benchmark datasets used in the paper are larger compared to that in Petroni, e.g., Wikidata is 3 orders of magnitude larger compared to the largest dataset in Petroni. (3) More importantly, compared to Petroni, the paper helps researchers explicitly understand what the language models learn, bridging the deep language model and knowledge graph communities through enhanced model transparency. Besides, the reach of the result is broad and has downstream utility for knowledge graph construction, deep neural network interpretation, and information extraction. The research would be a promising direction for future research as endorsed by Reviewer \#1.
> >
> > MATCHING DEGREE EFFECTIVENESS: Detailed analysis of the matching degree has been added in Sec. A.3 based on your feedback. We observe significant changes in the performance regarding the matching degree threshold based on Figure 3(b). We conclude the matching degree threshold is effective, which is mainly because of the knowledge contained in the self-attention matrix. The score in the attention matrix is representing the chance of the facts to be the true facts based on the stored knowledge. The main focus of the paper is to demonstrate that knowledge is stored in the pre-trained language models. For this purpose, an unsupervised method (no training/fine-tuning is involved) is desired. We plan to explore supervised/semi-supervised approaches to learn to rank the facts or classify the facts into true facts or negative facts to further improve the performance. Thanks for your valuable feedback.
> >
> > OTHERS: (a) Thanks for your feedback. We have changed to ``self-attention weight matrix ... is one of the main containers''. As mentioned in the paper, another advantage of using self-attention is that the relational connections or dependencies within a sentence have been pre-built, so we do not need to construct the connections between different parts of the sentence from scratch. (b) We have added additional detailed analysis of the unmapped facts, and fixed the terminology as shown in Sec. 3.2. Thanks for pointing this out.

---

> > ### Comment · AnonReviewer3 · 2020-11-24
> > **Response**
> >
> > Thanks to the authors for the clarifying responses. Unfortunately, the revisions do not address some of my concerns:
> >
> > - Re: evaluation and mapping stage details: My main concern about the evaluation was whether the facts extracted from the unstructured text are already present in the reference KG used for relation mapping (in the case of Wikidata). This is still not clear to me -- at least there is no mention in the paper of explicitly constructing the test document collections to avoid this. Since our goal is always to extract new facts from unseen document collections, this is important to clarify.
> >
> > - Re: related work: I disagree with the limitations pointed about Petroni et al. That approach can be easily used to populate new facts in a KG, and it can also be extended to extract facts from a passage by providing that as a context. While I agree that by relying solely on attention maps, this paper may provide additional interpretability of where the extracted fact comes from, it is important to understand, at least on the subset of facts used by Petroni et al, how does the current proposed approach compare to using natural language templates constructed from the relations.

---

> > > ### Author Response · Authors · 2020-11-25
> > > **Response to AnonReviewer3 Response**
> > >
> > > Thanks a lot for your valuable feedback. And thanks for clarifying your concerns and helping to further improve the work. We have addressed them in the new version. Please find the corresponding replies below.
> > >
> > > RE: Re: evaluation and mapping stage details: We have added additional details based on your feedback in Sec. A.1. We randomly sampled a hold-out dataset including 2,000 documents from the TAC KBP corpus and English Wikipedia for the relation mapping on TAC KBP and Wikidata respectively. The hold-out dataset is not used in the evaluation. The evaluation datasets in Sec.3 are considered as test datasets only. Conceptually, the mapping stage aims to map candidate facts to the reference knowledge graph schema, not the actual knowledge graphs. We also notice that, compared to the size of evaluation/test datasets as shown in Table 1 (3 million and 6 million), the hold-out is small. The incomplete relation mapping due to the size of the hold-out dataset is actually one of the major reasons leading to the errors (around 5\%) in the resulting facts as analyzed in both Sec. 3.2 and Sec. A.2. We plan to construct improved relation mapping by leveraging more sophisticated methods, such as life-long learning, which is also mentioned in the new version. Yes, we agree that we should always target new facts. In this paper, we first evaluate the quality of the generated facts with oracle knowledge graphs (Sec. 3.1). We then evaluate the quality of the new facts in Sec. 3.2, and find approximately 35\% of the new facts are true. We will continue to improve the quality and the coverage of the new facts in the future as mentioned in the paper. Thanks again for pointing this out.
> > >
> > > RE: Re: related work: We have added more discussion on Petroni et al. in Sec. 4 based on your feedback. While we agree with you that the approach in Petroni et al. can be extended to extract facts from a passage, we also notice that there are many fundamental differences between Petroni et al. and our approach. (1) Petroni et al. does not construct knowledge graphs, instead, it aims to answer questions. We instead construct knowledge graphs. (2) There are also two major limitations of using Petroni et al. to construct knowledge graphs: (2.1) Additional queries must be constructed first. A pre-defined head entity and relation are required for Petroni et al. to construct the queries, then Petroni et al. can be used to produce candidates. However, it requires additional effort in producing the queries. And it is not trivial to automatically generate entities and relations for Petroni et al. While our approach does not require constructing additional queries by relying solely on the input corpus; (2.2) The answers for the queries must be linked to knowledge graphs. It is non-trivial to map the candidates from Petroni et al. based on the queries to knowledge graph schema, since probably a similar procedure with our mapping stage is desired, but it is not clear how to construct the entity and relation mapping. We will study how to adapt Petroni et al. to the knowledge construction task as a future direction. Thanks again for the detailed and insightful review.

---

### Official Review · AnonReviewer4 · 2020-10-29
**An interesting question but the methodology needs improvement.**

**Rating:** 4
**Confidence:** 4

**Review:**

This paper targets an ambitious question -- constructing a knowledge base from scratch using pre-trained language models. The proposed approach deals two problems: (a) constructing facts from raw corpora and (b) disambiguating the entities in the fact by linking them to WikiData entities.

First, the insight that knowledge base is embedded in the world knowledge, which is captured by the pretrained embeddings, is not new. [1] studies the joint embedding of knowledge base and text and thus performs knowledge base construction in a semi-supervised fashion, which seems to be more promising than reconstructing the knowledge base from scratch in an unsupervised fashion. I'm curious to see the performance of the power of existing knowledge bases, the signals there can be used to initiate the training.

Second, baseline comparisons seem to be missing in the experiment section. Even though the task sounds difficult, the numerical performance on recall and F1 are not satisfactory. It's hard to understand the current performance without a reasonable baseline. Apart from that, it would be nice to see the error analysis -- what's the common pattern where MaMa fails?

In addition to the above major comments, I have the following questions:
(a) In the map phase where MaMa converts a sentence into facts, do we have an underlying assumption that the input sentence is a fact passage. What will happen if not?
(b) What if the input passage contains more than one fact? E.g., "Bob Dylan (born Robert Allen Zimmerman; May 24, 1941) is an American singer-songwriter, author, and visual artist." (the first sentence from Wikipedia) contains facts: (bob dylan, is, song writer), (bob dylan, is author), (bob dylan, is visual artist), (bob dylan, also named, Robert Allen Zimmerman), (bob dylan, date of birth, May 24, 1941).

[1] Wang, Zhen, et al. "Knowledge graph and text jointly embedding." Proceedings of the 2014 conference on empirical methods in natural language processing (EMNLP). 2014.

---

> ### Author Response · Authors · 2020-11-18
> **Response to AnonReviewer4**
>
> We thank the reviewer for the valuable feedback. The concerns are addressed in the rebuttal revision. In particular, based on your suggestion, we have clarified the novelty of the paper, and added extensive error analysis in both the main body and the newly added appendix (30 pages) including examples of the facts from Figure 4 to Figure 32. We have also attached some slides to summarize the key idea and findings of the paper as supplementary materials at the end of the rebuttal revision. Please find our responses below.
>
> NOVELTY: The paper shows that the knowledge has been stored in the deep language models through pre-training on large corpora. The proposed approach is able to cast the stored knowledge into a structured knowledge graph in an unsupervised fashion. The main focus of the paper is to demonstrate the knowledge is stored in the pre-trained language models. For this purpose, an unsupervised approach (no training or joint training or fine-tuning is involved) is desired. The problem and proposed approach are new. Both Reviewer \#1 and Reviewer \#3 have endorsed the novelty of the approach, e.g., ``the work is the first evidence of relational knowledge is stored in the pre-trained language models'' based on Reviewer \#3. Compare to the joint training with the knowledge base to improve shallow word embedding [1], we find that the knowledge is already stored in the pre-trained deep language models. We have cited the paper and added the discussion in Sec. 4. In the future, we plan to incorporate certain knowledge into language models for specific domains if the domain knowledge is missing in the pre-trained language models. Thanks for the valuable suggestion.
>
> COMPARISON AND ERROR ANALYSIS: Following your suggestion, except for comparing with Stanford OpenIE, we have also added a new compared system called OpenIE 5.1 (one of the strongest open IE systems). Our method significantly outperforms OpenIE 5.1 by over 6.5\% in F1 on TAC KBP. We have also added results of two larger pre-trained models, GPT-2_LARGE and GPT-2_XL with model sizes 774M and 1558M, which are approximately 2x and 5x larger than the largest model in the previous version. In total, we have experimented with six pre-trained language models. From results in Table 2 and Table 3, we find that all the proposed methods significantly outperform compared methods on both benchmark datasets, including the state-of-the-art Stanford OpenIE on TAC KBP. On Wikidata, our approach outperforms the best compared system by over 5.6\% in F1. On TAC KBP, our approach outperforms the best compared system by over 2.6\% in F1. Following your suggestion, we have also added a more detailed error analysis. We find that the main cause of the moderate recall is the incorrect entities caused by the third-party spaCy noun chunk as summarized in Sec. A.2. To further improve the recall, we are investigating enhancing the entity detection with more accurate entity recognition, entity linker, relation mapping, and relation generation, which have been extensively discussed and incorporated in Sec. 3.2 and A.2. In addition, a detailed error analysis on ummapped facts has been incorporated in Sec. 3.2. The detailed error analysis on mapped facts has been shown in Sec. A.2. In both cases, we find at least one-third of the errors are caused by the third-party spaCy noun chunk. The examples of mapped and unmapped facts from the Map stage have been shown from Figure 4 to Figure 32 to provide more insights for you.
>
> ADDITIONAL QUESTIONS: Thanks for the insightful questions. For question (a): We do not assume the input passage containing facts. Whether there is any fact generated from a passage is totally up to the proposed approach. In general, there are two situations when no fact is generated. (1) No candidate facts are generated from Match stage: this could due to many factors. For example, there are less than two entities identified by the spaCy noun chunk (in some cases, there is just one or even no entities. This is actually part of the reasons leading to the moderate recall as mentioned in the above response). This could also due to the candidate is filtered out by any of the constraints defined in Sec. 2.1.2. For example, if the matching degree of a candidate fact is below the threshold (Constraint \#1), the candidate will be filtered out, thus will not appear in the final facts set. (2) No mapped facts from Map stage: for facts that are mapped to KG schema (Sec. 2.2.1), those are final facts. In this case, we still preserve unmapped facts. For unmapped facts (Sec. 2.2.2), some of them are true facts (35\% based on the analysis in Sec. 3.2), some of them are not. For question (b): Theoretically, all the valid facts should be generated. But similar to the response to question (a), it is totally up to the proposed approach, i.e., Match and Map stages. For the particular example about ``Bob Dylan'', all the facts in the review are produced.

---

### Official Review · AnonReviewer2 · 2020-10-29
**Interesting unsupervised IE approach but not 100% convinced**

**Rating:** 5
**Confidence:** 4

**Review:**

Paper summary: The paper introduces an unsupervised method that utilizes an off-the-shelf BERT (without any fine-tuning) to create an information extraction system without any training data. It first creates ungrounded triples (a.k.a. OpenIE) from raw text by looking into the attention weights between words and finding a sequence of words through a beam search that has high attention weights between every consecutive words. When such sequence is created, the first and the last word become the head and the tail entities and the words between them becomes the relation. The next step is grounding each triplet to a known Knowledge Graph by utilizing off-the-shelf entity and relation grounding mechanisms. The proposed method shows 1-2% F1 score advantage over Stanford OpenIE on TAC KBP and Wikidata.

Strengths:
- The proposed method is unsupervised (doesn't require any training process or data).
- The proposed method only requires a single application of BERT per paragraph (instead of, for instance, applying BET for every possible triplet), which is relatively efficient.
- The idea of looking into the pooled attention weights for linking entities and relations is novel.


Weaknesses:
- The head and the tail entities are always single words, whereas many entities such as names have two or more words.
- The comparison is made only against Stanford OpenIE, which was proposed 5 years ago.
- The paper highly depends on the grounding techniques from many years go, namely Spitkovsky & Chang (2012) and Stanford OpenIE (2015). Given that the proposed model and Stanford OpenIE have similar performance, I think it is necessary to perform ablation study and show how much is the dependency. In the worst case, I am worried that most of the work is being done in this grounding stage (i.e. ungrounded triplets are not so good)
- If all possible head-tail pairs are enumerated in a sentence, my expectation is that there will be a lot of garbage triplets. It is amazing yet hard to believe that such simple thresholding techniques make it work and get rid of all of the bad triplets. More analysis will be helpful.

Overall: The approach is interesting in that it is completely unsupervised, and is able to achieve a meaningful performance (compared to Stanford OpenIE). However, I am not 100% convinced how this could work so more analysis would be helpful, and given that Stanford OpenIE is pre-BERT from 5 years ago, the gain seems to be not so exciting.

---

> ### Author Response · Authors · 2020-11-18
> **Response to AnonReviewer2**
>
> We thank the reviewer for the valuable feedback. The following new changes have been incorporated in the rebuttal revision based on your feedback. Please find our responses below. In particular, following your feedback, we have added one of the most recent approaches for comparison (our method still significantly outperforms the compared approach by more than 6.5\% in F1), as well as experimented with two larger pre-trained models, which results in outperforming the best compared system by over 5.6\% and 2.6\% in F1 on two datasets respectively. We have also added more details and results of the grounding/mapping stage both in the main body of the paper and the newly added 30-page appendix based on your feedback. Additional results of the Map/Ground stage have been provided from Figure 4 to Figure 32. We have also attached some slides to summarize the key idea and findings of the paper as supplementary materials at the end of the rebuttal revision.
>
> First of all, we are very happy that you like the problem and the proposed method.
>
> PRE-TRAINED LANGUAGE MODELS: We have experimented with six pre-trained language models (not only BERT), namely, BERT_BASE (109M), BERT_LARGE (335M), GPT-2 (117M), GPT-2_MEDIUM (345M), GPT-2_LARGE (774M), and GPT-2_XL (1558M).
>
> SUBWORD HANDLING: We convert the subwords from both BERT tokenizer and GPT-2 tokenizer to the corresponding full words. This has been incorporated in Sec. 2.1.1. Thanks for pointing this out!
>
> COMPARISON: Following your suggestion, we have added a new compared method called OpenIE 5.1 from the year 2018 (one of the most recent and strongest open IE systems). Our method significantly outperforms OpenIE 5.1 by more than 6.5\% in F1 on TAC KBP. We have also added results of two larger pre-trained models, GPT-2_LARGE and GPT-2_XL with model sizes 774M and 1558M, which are approximately 2x and 5x larger than the largest model in the previous version. In total, we have experimented with six pre-trained language models. We have achieved more than 4\% F1 improvements both on TAC KBP and Wikidata compared to the best results of our approach in the previous version. From results in Table 2 and Table 3, we find that all the proposed methods significantly outperform compared methods on both datasets, including the state-of-the-art Stanford OpenIE on TAC KBP. On Wikidata, our best approach outperforms the best compared system by over 5.6\% in F1. On TAC KBP, our best approach outperforms the best compared system by over 2.6\% in F1.
>
> GROUNDING STRATEGY: To ensure a fair comparison, the same grounding/mapping stage is shared across compared methods and our method, including the same entity linking dictionary and relation map, which is detailed in Sec. 3.1.2. As stated above, our proposed system outperforms the best compared system by over 5.6\% and 2.6\% in F1 on two datasets respectively. The results are based on the ablation study, and the improvements are brought only by the Match stage. For the unmapped/ungrounded facts, we find 35\% of the unmapped facts are true based on human annotation, which again shows the effectiveness of the Match stage. Besides, we have provided more mapped and unmapped results from the Map/Ground stage from Figure 4 to Figure 32 in the appendix for your reference. We can see that both the Match stage and Map stage contribute to the final results. The details of the grounding strategy are incorporated in Sec. A.1. As described in Sec. A.1, we have enhanced both entity linking and relation mapping. For entity linking, we add new Wikipedia anchors to the mention-to-entity dictionary resulting in a total of 26 million entries compared to the 21 million entries in the Spitkovsky & Chang paper, and construct an additional dictionary to convert Wikipedia anchors to Wikidata items. For relation mapping, compared to the relation mapping in Stanford OpenIE, we manually check whether the top relation phrases are true as described in Sec. 2.2.1. Both the entity linker and relation map will be released. The entity linking and relation map are not perfect mainly due to the third-party noun chunk based entity detection, we are looking into stronger entity detection, entity linker, and relation mapping approaches as discussed in Sec. 3.2 and Sec. A.2.
>
> MORE ANALYSIS: More detailed analysis has been added in Sec. A.3. We tune the parameters such as the matching degree threshold on a hold-out dataset, and observe significant changes in the performance. The matching degree threshold is effective, which is mainly because of the knowledge contained in the self-attention matrix. The score in the attention matrix is representing the chance of the facts to be the true facts based on the stored knowledge. However, we plan to explore supervised approaches to learn to rank the facts or classify the facts into true facts or negative facts to further improve the performance. Thanks for your valuable suggestion.

---

### Decision · Program_Chairs · 2021-01-07
**Final Decision**

**Decision:**

Reject

**Comment:**

We want to acknowledge that there has been a tremendous amount of work done during the discussion period on this paper for clarifying multiple points, adding multiple new comparison methods and new analysis. This is a very different draft than what was submitted and the reviewers acknowledged that. The draft is much closer from acceptance at ICLR than it was at submission time. However, despite all those additions, we do not support a publication at ICLR.

The main issues with the current draft are its positioning and motivations.  Right now the draft is in-between a paper about Knowledge Base Construction (KBC) and a paper analyzing the knowledge contained within a large language model. This in-between came up in multiple places during the discussion and is what causes the biggest confusion around this work. And, since there is no clear choice, the draft has limitation on either side.
* If the main point is around KBC to build general-purpose KBs, then one would expect experiments on downstream tasks powered by a KB, language understanding tasks for instance. Indeed, KBs are just a means to an end and the latest advances in very large language models have shown that KBs were not essential to be state of the art in language understanding tasks (GLUE, QA datasets, etc). So we would like to see whether these enhanced KBs could be beneficial. Or the KBs are studied as a way to encode commonsense like in (Bosselut et al., 2019) or (Davison et al., 2019), but this is not the point of the current draft.
* If the main impact of the draft is around what the language models learn, bridging the deep language model and knowledge graph communities through enhanced model transparency, as it has been said in the discussion, then the discussion with (Petroni et al. 19) should be more prominent and the introduction, motivation and experiments of the draft should reflect that.

That's why, even if this work is of solid quality, the current draft can not be accepted.